# Development of on-site self-calibration and retrieval methods for sky-radiometer observations of precipitable water vapor

Masahiro Momoi[1,2], Rei Kudo[3], Kazuma Aoki[4], Tatsuhiro Mori[5], Kazuhiko Miura[5], Hiroshi Okamoto[1], Hitoshi Irie[1], Yoshinori Shoji[3], Akihiro Uchiyama[6], Osamu Ijima[7], Matsumi Takano[8,7], and Teruyuki Nakajima[9]

[1]Center for Environmental Remote Sensing, Chiba University, Chiba, 263-8522, Japan
[2]Graduate School of Science, Tokyo University of Science, Tokyo, 162-8601, Japan
[3]Meteorological Research Institute, Japan Meteorological Agency, Tsukuba, 305-0052, Japan
[4]Graduate School of Science and Engineering, University of Toyama, Toyama, 930-8555, Japan
[5]Faculty of Science Division I, Tokyo University of Science, Tokyo, 162-8601, Japan
[6]Narional Institute for Environmental Studies, Tsukuba, 305-0053, Japan
[7]Aerological Observatory, Japan Meteorological Agency, Tsukuba, 305-0052, Japan
[8]Osaka Regional Headquarters, Japan Meteorological Agency, Osaka, 540-0008, Japan
[9]Earth Observation Research Center, Japan Aerospace Exploration Agency, Tsukuba, 305-8505, Japan
*Correspondence to*: Masahiro Momoi (1217641@ed.tus.ac.jp)

**Abstract.**

The Prede sky-radiometer measures direct solar irradiance and the angular distribution of diffuse radiances at the ultraviolet, visible, and near-infrared wavelengths. These data are utilized for remote sensing of aerosols, water vapor, ozone, and clouds, but the calibration constant which is the sensor output current of the extraterrestrial solar irradiance at the mean distance between the Earth and the sun, is needed. The aerosol channels, which are the weak gas absorption wavelengths of 340, 380, 400, 500, 675, 870, and 1020 nm, can be calibrated by an on-site self-calibration method, the Improved Langley method. This on-site self-calibration method is useful for the continuous long-term observation of aerosol properties. However, the continuous long-term observation of precipitable water vapor (PWV) by the sky-radiometer remains challenging, because calibrating the water vapor absorption channel of 940 nm generally relies on the standard Langley method (SL) at limited observation sites (*e.g.*, the Mauna Loa Observatory) and the transfer of the calibration constant by side-by-side comparison with the reference sky-radiometer calibrated by the SL method. In this study, we developed the SKYMAP algorithm, a new on-site method of self-calibrating the water vapor channel of the sky-radiometer using diffuse radiances normalized by direct solar irradiance (normalized radiances). Because the sky-radiometer measures direct solar irradiance and diffuse radiance using the same sensor, the normalization cancels the calibration constant included in the measurements. The SKYMAP algorithm consists of three steps. First, aerosol optical and microphysical properties are retrieved using direct solar irradiances and normalized radiances at aerosol channels. The aerosol optical properties at the water vapor channel are interpolated from those at aerosol channels. Second, PWV is retrieved using the angular distribution of the normalized radiances at the water vapor channel. Third, the calibration constant at the water vapor channel is estimated from the transmittance of PWV and aerosol optical properties. Intensive sensitivity tests of the SKYMAP algorithm using simulated data of the sky-radiometer showed that the calibration constant is retrieved reasonably well for PWV < 2 cm, which indicates that the SKYMAP algorithm can calibrate the water vapor channel on-site in dry conditions. Next, the SKYMAP algorithm was applied to actual measurements under the clear-sky and low PWV (< 2 cm) conditions at two sites, Tsukuba and Chiba, Japan, and the annual mean calibration constants at the two sites were determined. The SKYMAP-derived calibration constants were 10.1% and 3.2% lower, respectively, than those determined by side-by-side comparison with the reference sky-radiometer. After determining the calibration constant, we obtained PWV from the direct solar irradiances in both the dry and wet seasons. The retrieved PWV values corresponded well to those derived from a Global Navigation Satellite System/Global Positioning System receiver, a microwave radiometer, and an AERONET sun-sky radiometer at both sites. The correlation coefficients were greater than 0.96. We calculated the bias errors and the root mean square errors by comparing PWV between the DSRAD algorithm and other instruments. The magnitude of the bias error and the root mean square error were < 0.163 cm and < 0.251 cm for PWV < 3 cm, respectively. However, our

method tended to underestimate PWV in the wet conditions, and the magnitude of the bias error and the

root mean square error became large, < 0.594 cm and < 0.722 cm for PWV > 3 cm, respectively. This problem was mainly due to the overestimation of the aerosol optical thickness before the retrieval of PWV. These results show that the SKYMAP algorithm enables us to observe PWV over the long term, based on its unique on-site self-calibration method.

# 1 Introduction

The highly variable spatiotemporal distributions of aerosols, clouds, and gases (*e.g.,* water vapor and ozone) still include large uncertainties for the quantitative understanding of the Earth's radiation budget at various spatial and temporal scales. Water vapor is specified as an essential climate variable (ECV) by the World Meteorological Organization (WMO), a critical key parameter that contributes to characterizing Earth's climate and changes in atmospheric temperature (Schmidt *et al.*, 2010). Water vapor absorbs visible radiation and absorbs and emits infrared radiation to heat and cool the Earth and its atmosphere. Atmospheric heating drives the evaporation of sea water, causing an increase in temperature as positive feedback (IPCC, 2013). In addition, the distribution of water vapor controls precipitation amounts and aerosol-cloud interactions (Twomey, 1990). To understand these effects quantitatively, many previous studies have measured precipitable water vapor using a radiosonde, Global Navigation Satellite System (GNSS)/Global Positioning System (GPS) receiver (Bevis *et al.*, 1992), or spectroradiometer (*e.g.,* Fowle, 1912, 1915).

Precipitable water vapor (PWV), which is the total atmospheric water vapor contained in a vertical column, has been estimated from the measurement of direct solar irradiance at the water vapor absorption bands. One of the strong water vapor absorption bands is around 940 nm and can be measured by sun photometer (Fowle, 1912, 1915; Bruegge *et al.,* 1992; Schmid *et al.,* 1996, 2001; Halthore *et al.,* 1997), SKYNET sky-radiometer (Campanelli *et al.,* 2014, 2018; Uchiyama *et al.,* 2014, 2018a), and AERONET sun-sky photometer (Holben *et al.,* 1998). Previous studies of SKYNET and AERONET derived PWV from the observed transmittance of water vapor ($\bar{T}_{H2O}$), assuming $\bar{T}_{H2O} = e^{-a(m \cdot w)^b}$ (Bruegge *et al.*, 1992), where $a$ and $b$ are adjustment parameters, $m$ is the optical air mass, and $w$ is PWV. However, there is a known noticeable uncertainty in the estimate of PWV because the adjustment parameters depend on the spectral sensitivity of the spectroradiometer as well as the vertical profiles of water vapor and temperature. Therefore, the adjustment parameters should be determined for each observation site. Campanelli *et al.* (2014, 2018) developed a practical method for determining the adjustment parameters based on PWV retrieved by a GNSS/GPS receiver or by surface humidity observations.

To estimate PWV using a spectroradiometer, it is necessary to calibrate the water vapor channel. The calibration constant, which is the sensor output current of the extraterrestrial solar irradiance at the mean distance between the Earth and the sun, at the water vapor channel can be determined by the Langley method. For example, Uchiyama *et al.* (2014) calibrated the water vapor channel of a sky-radiometer with high accuracy using observations from the Mauna Loa Observatory (3400 m a.s.l.). In the AERONET led by NASA, the field instrument of the AERONET sun-sky radiometer is calibrated every year by lamp calibration and side-by-side comparison with a reference spectroradiometer (Holben

*et al.*, 1998). Dedicated effort and expenses are required to maintain accurate long-term calibrations using these methods.

The sky-radiometer models POM-01 and POM-02 (Prede, Tokyo, Japan), which are deployed in the international radiation observation network SKYNET, measure solar direct irradiances and diffuse irradiances at the ultraviolet, visible, and near-infrared wavelengths. These measurements are used for the remote sensing of aerosol, cloud, water vapor, and ozone (Table 1; Takamura and Nakajima, 2004; Nakajima *et al.*, 2007). Table 1 shows the relationship between the wavelengths and the main target of the remote sensing. The aerosol channels are 340, 380, 400, 500, 675, 870, and 1020 nm; the water vapor channel is 940 nm; the ozone channel is 315 nm; and the cloud channels are 1225, 1627, and 2200 nm. Through on-site self-calibration of the aerosol channels by the Improved Langley (IL) method (Tanaka *et al.,* 1986; Nakajima *et al.,* 1996; Campanelli *et al.*, 2004, 2007), the SKYNET system is capable of long-term and continuous aerosol observation. The IL method works not only in clean atmospheric conditions, but also in turbid atmospheric conditions. However, no improved calibration method has replaced the standard (Uchiyama *et al.*, 2014) or modified (Campanelli *et al.*, 2014, 2018) Langley methods for the water vapor channel. In this study, we developed a new method of retrieving PWV using the PWV dependency of the normalized radiance, defined as the ratio of diffuse radiance to direct solar irradiance at the water vapor channel. This method enables us to estimate PWV without the calibration constant, and to perform on-site self-calibration of the water vapor channel. We developed two algorithms, SKYMAP and DSRAD. The SKYMAP algorithm is a new on-site method for self-calibrating the water vapor channel. It retrieves PWV ($PWV_{SKYMAP}$) from the angular distribution of the normalized radiance at the water vapor channel and calibrates the water vapor channel. The DSRAD algorithm estimates PWV ($PWV_{DSRAD}$) from the calibrated direct solar irradiance at the water vapor channel. This method does not require adjustment parameters and explicitly uses the filter response function and the vertical profiles of water vapor, temperature, and pressure. The SKYMAP and DSRAD algorithms are described in Section 2. We discuss the results of sensitivity tests of the SKYMAP algorithm using simulation data in Section 3 and apply two algorithms to observational data at two SKYNET sites in Section 4. At these two sites, PWV is observed by the GNSS/GPS receiver, MWR, or AERONET sun-sky radiometer other than the sky-radiometer. The retrieval accuracy of our method is evaluated by comparison to these established methods.

## 2 Methods

In this study, PWV is retrieved using angular distributions of the normalized radiance, which does not require the calibration constant of the sky-radiometer. Section 2.1 shows the normalized radiances and dependencies of the normalized radiance on PWV. Next, we describe two algorithms, the flow and relationships of which are shown in Fig. 1. The SKYMAP algorithm retrieves aerosol optical and

microphysical properties and calibrates the water vapor channel by retrieving PWV from the angular distribution of the normalized radiance (Section 2.2). The DSRAD algorithm retrieves PWV from the transmittance derived from the direct solar irradiance at the water vapor channel (Section 2.3).

**2.1 Sky-radiometer measurements and the relationship between normalized radiances and PWV**

We explain the normalized radiance (Nakajima *et al.*, 1996) in Section 2.1.1 and the theoretical relationship between the normalized radiance and PWV in Section 2.1.2.

**2.1.1 Sky-radiometer measurements**

The direct solar irradiance ($F$) and angular distribution of the diffuse irradiance ($V$) are measured at seven wavelengths by the model POM-01 or eleven wavelengths by the model POM-02 (Table 1). $V$ is measured in the almucantar and principal planes (Fig. 2). The angular distribution of $V$ is measured at scattering angles $\Theta = 2°, 3°, 4°, 5°, 7°, 10°, 15°, 20°, 25°, 30°, 40°, 50°, 60°, 70°, 80°, 90°, 100°, 110°, 120°, 130°, 140°, 150°,$ and $160°$ in the almucantar and principal planes every 10 min. The aerosol channels are calibrated with the IL method using the normalized radiance at $\Theta < 30°$. $F$ and $V(\Theta \geqq 4°)$ at the aerosol and water vapor channels are used in this study.

In the plane-parallel non-refractive atmosphere, $F$ at the bottom of the atmosphere (BOA) at the solar zenith angle (SZA) $\theta_0$ and the solar azimuth angle $\phi_0$ is derived from

$$F(\lambda) = \frac{F_0}{d^2} \exp(-m_0 \tau(\lambda)), \quad (1)$$

where $F_0$ is the calibration constant; $d$ is the distance between Earth and the sun (AU); $\lambda$ is the wavelength; $\tau$ is the total optical thickness; and $m_0$ is optical air mass, represented as $m_0 = 1/\cos\theta_0$. In clear-sky conditions, the total optical thickness is the integrated value of aerosol scattering + absorption, Rayleigh scattering, and gas absorption coefficients in the column. Assuming a narrow spectral band filter response function, the normalized radiance ($R$), which is the ratio of $V$ to $F$ at the zenith angle ($\theta$) and the azimuth angle ($\phi$), is obtained from the radiative transfer equation:

$$R(\Theta, \lambda) = \frac{V(\Theta, \lambda)}{F(\lambda) m_0 \Delta\Omega} = \int_0^{\tau(\lambda)} \exp\left[(\tau - \tau')\left(\frac{1}{\mu_0} - \frac{1}{\mu}\right)\right] \omega'(\lambda, \tau') P'(\Theta, \lambda, \tau') d\tau' + Q(\Theta, \lambda), \quad (2)$$

where $P'(\Theta, \lambda, \tau')$ and $\omega'(\lambda, \tau')$ are, the total phase function and the total single scattering albedo, respectively, at the altitude $\tau = \tau'$, $\Delta\Omega$ is the solid view angle (or field of view); $Q$ is the multiple scattering contribution; and

$$\cos\Theta = \cos\theta \cos\theta_0 + \sin\theta \sin\theta_0 \cos(\phi - \phi_0), \quad (3)$$

$$\mu = \cos\theta \, ; \mu_0 = \cos\theta_0$$

Note that $F_0$ is cancelled by the normalization. In the second term of Eq. (2), the solid view angle of each wavelength can be retrieved from the angular distribution around the solar disk (Nakajima *et al.*, 1996; Boi *et al.*, 1999; Uchiyama *et al.*, 2018b). Eq. (2) can be simplified in the almucantar plane due to $\theta = \theta_0$:

$$R(\Theta, \lambda) = \int_0^{\tau(\lambda)} \omega'(\lambda, \tau') P'(\Theta, \lambda, \tau') d\tau' + Q(\Theta, \lambda) = \omega(\lambda)\tau(\lambda)P(\Theta, \lambda) + Q(\Theta, \lambda) \,, (4)$$

where $P(\Theta, \lambda)$ and $\omega$ are the total phase function and the total single scattering albedo, respectively. In contrast, $R$ in the principal plane can be described simply, similar to Eq. (4), if we assume that the atmosphere is a single layer:

$$R(\Theta, \lambda) = \frac{\mu_0^2}{\mu_0 - \mu} \omega(\lambda)P(\Theta, \lambda) \left[ 1 - \exp\left(\frac{\tau(\lambda)}{\mu_0} - \frac{\tau(\lambda)}{\mu}\right) \right] + Q(\Theta, \lambda) \,. \quad (5)$$

### 2.1.2 The relationship between normalized radiances at the water vapor channel and PWV

We examined the sensitivity of $R$ at 940 nm in the two observation planes to PWV, aerosol optical properties, and aerosol vertical profiles by simulating $R$ using the radiative transfer model RSTAR (Nakajima and Tanaka, 1986, 1988). The simulation was conducted with two aerosol types based on those used by Kudo *et al.* (2016): the continental average, and the continental average + transported dust in the upper atmosphere (Table 2). The continental average consisted of water-soluble particles, soot particles, and insoluble particles (Hess *et al.*, 1999). Transported dust was defined as the mineral-transported component from Hess *et al.* (1999). Figure 3 shows the dependencies of $R$ in the almucantar plane on PWV for continental average aerosol with aerosol optical thicknesses of 0.02 and 0.20 at 940 nm. The simulations were conducted for the SZA of 70°. $R$ decreases with increasing PWV regardless of the aerosol optical thickness. This suggests that PWV can be estimated from the normalized angular distribution, which is the angular distribution of $R$, without the calibration constant. The dependencies of $R$ on PWV cannot be observed in the radiative transfer using single scattering approximation in the almucantar plane. The first term of Eq. (4) is the normalized single scattering contribution and includes only the influences of aerosol and Rayleigh scattering. Note that this is true only for $R$, and not for $V$, because total optical thickness contributes to the single scattering approximation of $V$. However, the second term for the multiple scattering includes the influence of water vapor absorption and creates the dependencies of $R$ on PWV. Figure 3 shows that the dependency of $R$ on PWV at the forward scattering

angles is not strong, but $R$ at the backward scattering angles between 90° and 120° changes with PWV. The range of the scattering angle for $R$ is an important factor.

Figure 4 illustrates the dependency of $R$ on PWV for different observation planes. The simulation was conducted for transported dust aerosol (Table 2) with an aerosol optical thickness of 0.06 at 940 nm at an SZA of 70° in the almucantar and principal planes. The transported dust aerosol is composed of coarse particles, which have larger impacts on the angular distribution of $R$ at the near-infrared wavelength than fine particles. The dependency of $R$ in the almucantar plane on PWV is the same as in Fig. 3. The dependency of $R$ on PWV is also found in the principal plane. $R$ increases with increasing PWV at $\theta \ll \theta_0$ and decreases with increasing PWV at $\theta \gg \theta_0$. Although the dependency of $R$ on PWV in the almucantar plane is strong at the backward scattering angles, that in the principal plane is strong at scattering angles between 60° and 90°. $R$ in the principal plane is more sensitive to PWV than $R$ in the almucantar plane because the normalized single scattering contribution in Eq. (5) includes not only Rayleigh and aerosol scattering but also gas absorption.

In theory, the maximum scattering angle of the principal plane is $\theta_0 + 90°$ and that of the almucantar plane is $2\theta_0$. When the SZA is small, the principal plane has a broader scattering angle range than the almucantar plane. Therefore, the principal plane is more advantageous for PWV retrieval. Figure 5 is the same as Fig. 4 but for an SZA of 30°. Because the maximum scattering angle of the principal plane is obviously larger than that of the almucantar plane, PWV retrieval using the principal plane is more effective compared to that using the almucantar plane.

$R$ in the principal plane is affected by the aerosol vertical profile, but this influence can be ignored for $R$ in the almucantar plane (Torres *et al*., 2014). Figure 6 shows the normalized angular distribution in the two observation planes for the different heights of the transported dust layer. It is obvious that the normalized angular distribution in the principal plane is sensitive to the aerosol vertical profile. Consequently, the principal plane is useful for retrieving PWV when the aerosol vertical profile is known, but the almucantar plane is better when the aerosol vertical profile is not known. In this study, we used the normalized angular distribution in the almucantar plane because the aerosol vertical profile was not known. The influence of SZA on the retrieval of PWV is examined in Section 3.

**2.2 SKYMAP algorithm**

The SKYMAP algorithm consists of three steps (Fig. 7). First, aerosol optical and microphysical properties are retrieved from $F$ and normalized angular distributions at aerosol channels. Second, aerosol optical properties at the water vapor channel are interpolated from those at aerosol channels. PWV is retrieved from the normalized angular distribution at the water vapor channel. Third, the calibration constant at the water vapor channel is estimated from PWV and the aerosol optical properties.

### 2.2.1 Step 1: Retrieval of aerosol optical and microphysical properties

Aerosol optical and microphysical properties are estimated from sky-radiometer measurements at aerosol channels using normalized angular distributions and transmittance $T = \frac{Fd^2}{F_0}$ with an optimal estimation method similar to the AERONET and SKYNET retrievals (Dubovik and King, 2000; Dubovik *et al.,* 2006; Kobayashi *et al.*, 2006; Hashimoto *et al.*, 2012; Kudo *et al.*, 2016). Estimated optical and microphysical properties are the real and imaginary parts of the refractive index at aerosol channels (Table 1), the volume size distribution, and the volume ratio of non-spherical particles to total particles in coarse mode. Hereafter, these are referred to as aerosol parameters.

In step 1, we construct the forward model to calculate the sky-radiometer measurements from the aerosol parameters. We assume that the aerosol volume size distribution in the radius range from 0.02 to 20.0 μm consists of 20-modal lognormal size distributions as illustrated in Fig. 8:

$$\frac{dV(r)}{d\ln r} = \sum_{i=1}^{20} C_i \exp\left[-\frac{1}{2}\left(\frac{\ln r - \ln r_i}{s}\right)^2\right], \qquad (6)$$

$$\ln r_i = \ln(0.02\mu m) + \frac{2i-1}{2}\ln \Delta r, \qquad (7)$$

$$s \equiv \frac{\ln \Delta r}{\eta}, \qquad (8)$$

$$\ln \Delta r \equiv \frac{1}{20}\left(\ln(20\mu m) - \ln(0.02\mu m)\right) = \frac{3}{20}\ln 10, \qquad (9)$$

where $C_i$, $r_i$, and $s$ are the volume, radius, and width of each lognormal function, respectively. $\eta$ is the parameter to determine the width and is given by a fixed value (Appendix A). We can separate the size distribution into fine and coarse modes by giving the boundary radius $r_b$, which is obtained as the local minimum. Furthermore, we separate coarse mode into spherical and non-spherical particles:

$$\frac{dV(r)}{d\ln r} = \frac{dV_f(r)}{d\ln r} + (1-\delta)\frac{dV_c(r)}{d\ln r} + \delta\frac{dV_c(r)}{d\ln r}, \qquad (10)$$

where $\frac{dV_f(r)}{d\ln r}$ is fine mode, $\frac{dV_c(r)}{d\ln r}$ is coarse mode, and $\delta$ is the fraction of the non-spherical particles in coarse mode (Fig. 8). The aerosol optical properties are calculated from the size distribution and refractive index, similar to the methods of Kudo *et al.* (2016) and Dubovik *et al.* (2006), as follows:

$$\tau_{\text{ext/sca}}(\lambda) = \sum_k \frac{dV_{\text{f}}(r_k)}{d\ln r} K^{\text{S}}_{\text{ext/sca}}(\lambda, n, k, r_k) + \sum_k (1-\delta) \frac{dV_{\text{c}}(r_k)}{d\ln r} K^{\text{S}}_{\text{ext/sca}}(\lambda, n, k, r_k) +$$

$$\sum_k \delta \frac{dV_{\text{c}}(r_k)}{d\ln r} K^{\text{NS}}_{\text{ext/sca}}(\lambda, n, k, r_k), \qquad (11)$$

$$\tau_{\text{sca}}(\lambda) P_{ii}(\Theta, \lambda) = \sum_k \frac{dV_{\text{f}}(r_k)}{d\ln r} K^{\text{S}}_{ii}(\Theta, \lambda, n, k, r_k) + \sum_k (1-\delta) \frac{dV_{\text{c}}(r_k)}{d\ln r} K^{\text{S}}_{ii}(\Theta, \lambda, n, k, r_k) +$$

$$\sum_k \delta \frac{dV_{\text{c}}(r_k)}{d\ln r} K^{\text{NS}}_{ii}(\Theta, \lambda, n, k, r_k), \qquad (12)$$

where $\tau_{\text{ext/sca}}(\lambda)$ denotes the optical thickness of extinction and scattering, and $\tau_{\text{sca}}(\lambda) P_{ii}(\Theta, \lambda)$ denotes the directional scattering corresponding to the scattering matrix elements $P_{ii}(\Theta, \lambda)$. $K^{\text{S}}$ and $K^{\text{NS}}$ are the kernels of extinction and scattering properties for spherical and non-spherical particles, respectively. $n$ and $k$ are the real and imaginary parts of the refractive index, respectively. We use randomly oriented spheroids as non-spherical particles and use the kernels developed by Dubovik *et al.* (2006).

We compute normalized angular distributions and transmittances of the extinction, using the radiative transfer model RSTAR (Nakajima and Tanaka, 1986, 1988). The model atmosphere is divided by 18 altitudes of 0, 1, 2, 3, 4, 5, 6, 7, 8, 9, 10, 15, 20, 30, 40, 50, 70, and 120 km. Atmospheric vertical profiles of temperature and pressure are obtained from the NCEP/NCAR Reanalysis 1 data. The absorption coefficients of $H_2O$, $CO_2$, $O_3$, $N_2O$, $CO$, $CH_4$, and $O_2$ are calculated by the correlated *k*-distribution method from the data table of Sekiguchi and Nakajima (2008).

The aerosol parameters for the best fit to all measurements (normalized angular distributions and transmittances at aerosol channels) and *a priori* information are obtained by minimizing the following cost function,

$$f(\boldsymbol{x}) = \frac{1}{2}\big(\boldsymbol{y}^{\text{meas}} - \boldsymbol{y}(\boldsymbol{x})\big)^T (\boldsymbol{W}^2)^{-1}\big(\boldsymbol{y}^{\text{meas}} - \boldsymbol{y}(\boldsymbol{x})\big) + \frac{1}{2}\big(\boldsymbol{y}_a(\boldsymbol{x})\big)^T (\boldsymbol{W}_a^2)^{-1}\big(\boldsymbol{y}_a(\boldsymbol{x})\big), \quad (13)$$

where vector $\boldsymbol{y}^{\text{meas}}$ describes the measurements (normalized radiances $R^{\text{meas}}$ and transmittances of total extinction $T^{\text{meas}}$) at the aerosol channels, vector $\boldsymbol{x}$ describes the aforementioned aerosol parameters — $n(\lambda)$, $k(\lambda)$, $C_i$, and $\delta$ — to be estimated, vector $\boldsymbol{y}(\boldsymbol{x})$ comprises the values corresponding to $\boldsymbol{y}^{\text{meas}}$ calculated from $\boldsymbol{x}$ by the forward model ($R^{\text{ret}}$ and $T^{\text{ret}}$), and matrix $\boldsymbol{W}^2$ is the covariance matrix of $\boldsymbol{y}$ and is assumed to be diagonal. The diagonal elements of $\boldsymbol{W}$ are standard errors in the measurements. We set their values at 0.02 for $T^{\text{meas}}$, and 10% for $R^{\text{meas}}$.

To reduce the effects of observational error on retrieval and to conduct stable analyses, Dubovik and King (2000) considered restricting the spectral variability of the volume size distribution and limiting the length of the refractive index derivative with respect to the wavelength. They considered this *a priori* smoothness constraint as being of the same nature as a measurement and incorporated the smoothness constraint into their retrieval scheme. We also consider the smoothness constraints in this

study. The second term of Eq. (13) consists of *a priori* information on the wavelength dependencies of the refractive index, aerosol optical thickness, and smoothness of the volume spectrum, which is described as

$$\boldsymbol{y}_a(x) = \left(\boldsymbol{y}_a^{\mathrm{Re}}, \boldsymbol{y}_a^{\mathrm{Im}}, \boldsymbol{y}_a^{\mathrm{Sca}}, \boldsymbol{y}_a^{\mathrm{Abs}}, \boldsymbol{y}_a^{\mathrm{Vol}}\right)^T, \qquad (14)$$

where vectors $\boldsymbol{y}_a^{\mathrm{Re}}$, $\boldsymbol{y}_a^{\mathrm{Im}}$, $\boldsymbol{y}_a^{\mathrm{Sca}}$, $\boldsymbol{y}_a^{\mathrm{Abs}}$, and $\boldsymbol{y}_a^{\mathrm{Vol}}$ are *a priori* information on the wavelength dependencies of the refractive index (real and imaginary parts), aerosol optical thickness (scattering and absorption parts), and smoothness of the volume spectrum, respectively. The matrix $\boldsymbol{W}_a^2$ in Eq. (13) is the covariance matrix for determining the strengths of the constraints.

We adapt the smoothness constraints of the second derivative for the real and imaginary parts of the refractive index. The second derivatives are defined as

$$y_a^{\mathrm{Re}(i)}(\boldsymbol{x}) = \left(\frac{\ln (\lambda_i) - \ln n(\lambda_{i+1})}{\ln \lambda_i - \ln \lambda_{i+1}} - \frac{\ln n(\lambda_{i+1}) - \ln n(\lambda_{i+2})}{\ln_{i+1} - \ln \lambda_{i+2}}\right), \quad (15)$$

$$y_a^{\mathrm{Im}(i)}(\boldsymbol{x}) = \left(\frac{\ln k(\lambda_i) - \ln (\lambda_{i+1})}{\ln \lambda_i - \ln \lambda_{i+1}} - \frac{\ln k(\lambda_{i+1}) - \ln k(\lambda_{i+2})}{\ln \lambda_{i+1} - \ln_{i+2}}\right), \quad (16)$$

$$(i = 1, \cdots, N_w - 2),$$

where $y_a^{\mathrm{Re}(i)}$ and $y_a^{\mathrm{Im}(i)}$ are the *i*-th elements of the vectors $\boldsymbol{y}_a^{\mathrm{Re}}$ and $\boldsymbol{y}_a^{\mathrm{Im}}$, respectively. $N_{\mathrm{w}}$ is the number of wavelengths. The values entered into the weight matrix $\boldsymbol{W}_a$ are 0.2 for the real part and 1.25 for the imaginary part. These values are adopted from Dubovik and King (2000). Furthermore, we introduce the smoothness constraints to the spectral distributions of the scattering and absorption parts of the aerosol optical thickness by

$$y_a^{\mathrm{Sca}(i)}(\boldsymbol{x}) = \left(\frac{\ln_{sca}(\lambda_i) - \ln \tau_{sca}(\lambda_{i+1})}{\ln_i - \ln \lambda_{i+1}} - \frac{\ln_{sca}(\lambda_{i+1}) - \ln \tau_{sca}(\lambda_{i+2})}{\ln_{i+1} - \ln \lambda_{i+2}}\right), \qquad (17)$$

$$y_a^{\mathrm{Abs}(i)}(\boldsymbol{x}) = \left(\frac{\ln \tau_{abs}(\lambda_i) - \ln \tau_{abs}(\lambda_{i+1})}{\ln_i - \ln \lambda_{i+1}} - \frac{\ln_{abs}(\lambda_{i+1}) - \ln \tau_{abs}(\lambda_{i+2})}{\ln_{i+1} - \ln \lambda_{i+2}}\right), \qquad (18)$$

$$(i = 1, \cdots, N_w - 2),$$

where $y_a^{\mathrm{Sca}(i)}$ and $y_a^{\mathrm{Abs}(i)}$ are the *i*-th elements of the vectors $\boldsymbol{y}_a^{\mathrm{Sca}}$ and $\boldsymbol{y}_a^{\mathrm{Abs}}$, respectively. The value entered in the weight matrix $\boldsymbol{W}_a$ is 2.5 for both the scattering and absorption parts of the aerosol optical thickness. To stabilize the estimation of the volume size distribution, we introduce the smoothness constraint for the adjacent volume size spectrum $C_i$, as:

$$y_a^{\text{Vol}(i)}(\boldsymbol{x}) = (\ln C_{i-1} - \ln C_i) - (\ln C_i - \ln C_{i+1}), \quad (19)$$

$$(i = 1,\cdots,20),$$

$$C_0 = 0.01 \times \min\{C_i | i = 1,\cdots,20\}, C_{21} = 0.01 \times \min\{C_i | r_i > r_b, i = 1,\cdots,20\}.$$

where $y_a^{\text{Vol}(i)}$ is the $i$-th element of the vector $\boldsymbol{y}_a^{\text{Vol}}$. The small values of $C_0$ and $C_{21}$ at $r_0$ and $r_{21}$ are given to prevent both ends of the size distribution ($C_1$ and $C_{20}$) from being abnormal values because $F$ and $V$ do not have sufficient information to estimate the size distribution of both small ($r < 0.1$ μm) and large particles ($r > 7$ μm; Dubovik *et al.*, 2000). Note that $r_0$ and $r_{21}$ satisfy Eq. (7). The value entered in the weight matrix $\boldsymbol{W}_a$ is 1.6 for the smoothness constraint of the size distribution.

We minimize $f(\boldsymbol{x})$ of Eq. (13) using the algorithm developed in Kudo *et al.* (2016), which is based on the Gauss-Newton method and the logarithmic transformations of $\boldsymbol{x}$ and $\boldsymbol{y}$. Finally, the aerosol optical properties from aerosol channels are obtained from $\boldsymbol{x}$ using Eqs. (11) and (12).

### 2.2.2 Step 2: Retrieval of PWV

We estimate PWV by the following procedure. The aerosol volume size distribution is obtained from step 1, and the refractive index at 940 nm is calculated from those at 870 and 1020 nm by linear interpolation in the log-log plane. Using the size distribution and the interpolated refractive index, we can compute the aerosol optical properties and the normalized angular distribution at the water vapor channel using the forward model described in Section 2.2.1. We retrieve PWV by minimizing the following cost function:

$$f(\boldsymbol{x}) = \frac{1}{2}\left(\boldsymbol{y}^{\text{meas}} - \boldsymbol{y}(\boldsymbol{x})\right)^T (\boldsymbol{W}^2)^{-1}\left(\boldsymbol{y}^{\text{meas}} - \boldsymbol{y}(\boldsymbol{x})\right), (20)$$

where the component of vector $\boldsymbol{x}$ is PWV, vectors $\boldsymbol{y}^{\text{meas}}$ and $\boldsymbol{y}(\boldsymbol{x})$ are the normalized angular distribution in the range from 4° to 160°, matrix $\boldsymbol{W}^2$ is assumed to be diagonal, and the values of the diagonal matrix $\boldsymbol{W}$ are assumed to be 10%. The cost function is minimized by the Gauss-Newton method. Note that this process does not require the calibration constant of the sky-radiometer, because we use the normalized angular distribution (Eq. [4]) to obtain PWV instead of using the direct solar irradiance (Eq. [1]).

### 2.2.3 Step 3: Retrieval of the calibration constant of the water vapor channel

$F_0$ at the water vapor channel can be obtained from the observed $F$ and the band average transmittance $\bar{T}_{\text{H2O}}$ converted from PWV in step 2 as follows:

$$F_0 = \frac{F d^2 e^{m \cdot (\tau_R + \tau_a)}}{\bar{T}_{H2O}}, \quad (21)$$

where $\tau_R$ and $\tau_a$ are Rayleigh scattering and aerosol optical thicknesses, respectively. The band average transmittance can be written as

$$\bar{T}_{H2O} = \frac{\int_{\Delta\lambda} \Phi(\lambda) T_{H2O}(\lambda) d\lambda}{\int_{\Delta\lambda} \Phi(\lambda) d\lambda} = \frac{\int_{\Delta\lambda} \Phi(\lambda) \exp\left(-m_{H2O}(\theta) \int_0^z \alpha_{H2O}(g_w(z), K(z), \lambda) dz\right) d\lambda}{\int_{\Delta\lambda} \Phi(\lambda) d\lambda}, \quad (22)$$

$$w = \int_0^z g_w(z) dz, \quad (23)$$

where $\Phi(\lambda)$ is the filter response function, $\Delta\lambda$ is the bandwidth of the filter response function, $T_{H2O}$ is the transmittance of water vapor at wavelength $\lambda$, $m_{H2O}(\theta)$ is the optical air mass, $g_w$ is the mass mixing ratio, $K$ is temperature, $\alpha_{H2O}$ is the absorption coefficient at altitude $z$, and $w$ is PWV. Eq. (22) is discretized by

$$\bar{T}_{H2O} = \frac{\sum_i^{N_s} \Phi_i \int_{\Delta\lambda_i} \exp\left(-m_{H2O}(\theta) \int_0^z \alpha_{H2O}(g_w(z), K(z), \lambda) dz\right) d\lambda}{\sum_i^{N_s} \Phi_i \Delta\lambda_i}, \quad (24)$$

where $\Phi_i$ is the stepwise filter response function, $\Delta\lambda_i$ is the sub-bandwidth of the filter response function, and $N_s$ is the number of sub-bands. We calculate the absorption coefficients at each wavelength by the correlated $k$-distribution (Sekiguchi and Nakajima, 2008) using the vertical profiles of temperature, pressure, and specific humidity in the NCEP/NCAR Reanalysis 1 data.

We can calculate a value for $F_0$ from a data set of the normalized angular distribution. Therefore, for example, a time series of $F_0$ in a day is obtained from the daily measurements of the sky-radiometer. The mean value of the calibration constant at the water vapor channel is determined by the robust statistical and iterative method with Huber's M-estimation:

$$\ln \bar{F}_0 = \sum_i \beta_H(t_i) \cdot \ln F_0(t_i), \quad (25)$$

$$\beta_H(t_i) = \begin{cases} 1 & (|\ln \bar{F}_0 - \ln F_0(t_i)| \le 0.03) \\ \frac{0.03}{|\ln \bar{F}_0 - \ln {}_0(t_i)|} & (|\ln \bar{F}_0 - \ln F_0(t_i)| > 0.03) \end{cases}, \quad (26)$$

where $\bar{F}_0$ is the mean calibration constant and is calculated at each iterative step, $F_0(t_i)$ is the calibration constant at a specific time $t$, and $\beta_H$ is Huber's weight function.

**2.2.4 Cloud screening using the smoothness criteria of the angular distributions (SCAD method)**

The SKYMAP algorithm can only be applied to measurements under clear-sky conditions. We estimated clear-sky conditions from two indexes calculated from sky-radiometer measurements. Index 1 is a value for the normalized radiances near the sun. If clouds pass over the sun, index 1 has large temporal variation. Index 2 is a value for the normalized angular distribution. If clouds are detected on the scanning plane of the sky-radiometer, the normalized angular distribution has large variation. Index 1 is defined as follows. First, the mean normalized radiance near the sun $\bar{R}_{\text{near}}$ is calculated by

$$\bar{R}_{\text{near}}(t) = \frac{1}{N}\sum_{i=1}^{N} R(\Theta_i, t), \Theta \leq 10^\circ, \qquad (27)$$

where $N$ is the number of measurements, and $R$ is the normalized radiance at a time $t$, scattering angle $\Theta$, and wavelength 500 nm. Next, the running mean of the time series of $\bar{R}_{\text{near}}(t)$ with a window of three consecutive data points is calculated as $< \bar{R}_{\text{near}}(t) >$. Index 1 is defined as the deviation $\tilde{R}_{\text{near}}(t)$ of $\bar{R}_{\text{near}}(t)$ from $< \bar{R}_{\text{near}}(t) >$,

$$\tilde{R}_{\text{near}}(t) = |\bar{R}_{\text{near}}(t) - < \bar{R}_{\text{near}}(t) >| / < \bar{R}_{\text{near}}(t) >. \qquad (28)$$

Index 2 is the deviation $\tilde{R}_{\text{far}}$ of normalized angular distributions far from the sun and is defined as

$$\tilde{R}_{\text{far}}(t) = \sigma\left(\frac{R(\Theta,t) - <R_{\text{far}}(\Theta,t)>}{<R_{\text{far}}(\Theta,t)>}\right), \Theta > 10^\circ, \quad (29)$$

where $< R_{\text{far}}(\Theta, t) >$ is the running mean of $R(\Theta_i, t)$ with a window of three consecutive data points, and $\sigma(\mathbf{X})$ is the standard deviation of data set $\mathbf{X}$. Note that the data for calculating $\tilde{R}_{\text{far}}$ varies depending on SZA, which limits available scattering angles. We judged clear-sky conditions when indexes 1 and 2 were both below their respective thresholds (0.1 and 0.2, respectively). We determined the thresholds by comparing the images of the whole-sky camera and the time series of the surface solar radiation observed by the pyranometer. Figure 9 is an example of the results for observations on January 6, 2014, in Tsukuba. Clear-sky conditions continued until 12:30, and then clouds passed over the sky until 15:00. Subsequently, there were clouds near the horizon, but the sky was almost clear. Our algorithm worked well, and cloudy scenes were eliminated. Although the whole-sky camera detected some clouds from 14:00 to 15:00, our algorithm judged the scenes as representative of clear-sky conditions. This may be

because there were no clouds in the line of sight of the sky-radiometer. The decline in the surface solar radiation around 9:00 was due to wiping of the glass dome of the pyranometer to keep the dome clean.

The method was applied to measurements from 2013 to 2014 at the Meteorological Research Institute, Japan Meteorological Agency (MRI, JMA), in Tsukuba. The results were validated using visual observation of the amount of clouds in the Aerological Observatory of the JMA. Figure 10a shows the histograms of index 1 for cases in which the sun was and was not covered by clouds. Index 1 had a low value when there were no clouds shading the sun but had a wide range of values when clouds were shading the sun. Fig. 10b shows the histograms of index 2 when cloud cover was and was not < 20%. The peak shifted to the right when cloud cover was $\geq$ 20%, but the effect was not significant. Table 3 shows the validation results of this method. We defined "best condition" as cloud cover < 20% and "poor condition" as cloud cover $\geq$ 20%. In less than 17% of cases a "poor condition" was judged as a "best condition". The sky-radiometer observes only a part of the whole sky, but our algorithm showed good results.

## 2.3 Estimation of PWV from direct solar irradiance (DSRAD algorithm)

The sky-radiometer observes the angular distribution of $V$ every 10 min but observes the direct solar irradiance every 1 min. Once the calibration constant is determined by the SKYMAP algorithm, we can estimate PWV from the direct solar irradiance. The DSRAD algorithm computes the aerosol optical thickness, and PWV from the direct solar irradiances at the aerosol and water vapor channels. Table 4 shows the references of the DSRAD algorithm. This algorithm consists of two steps. First, aerosol optical thicknesses at aerosol channels are calculated using direct solar irradiances. The aerosol optical thickness at the water vapor channel is interpolated from the aerosol optical thicknesses at 870 and 1020 nm by linear interpolation in the log-log plane. Second, the band mean transmittance of the water vapor, $\overline{T}_{\mathrm{H2O}}^{\mathrm{meas}}$, is calculated from the calibrated direct solar irradiance. PWV is retrieved using the formula,

$$\overline{T}_{\mathrm{H2O}}^{\mathrm{meas}} - \frac{\sum_{i}^{N_s} \Phi_i \int_{\Delta\lambda_i} \exp\left(-m_{\mathrm{H2O}}(\theta)\int_0^Z \alpha_{\mathrm{H2O}}(g_w(z),K(z),\lambda)dz\right)d\lambda}{\sum_{i}^{N_s}\Phi_i\Delta\lambda_i} = 0, \qquad (30)$$

where $m_{\mathrm{H2O}}$ is the optical air mass calculated by Gueymard (2001). Eq. (30) is solved using the Newton–Raphson method.

To ensure the quality of the data and avoid cloud contamination, we adopt the method of Smirnov *et al*. (2000) with two main differences, similar to Estellés *et al*. (2012). First, an aerosol optical thickness at 500 nm > 2 is considered cloud-affected data. Second, the triplet of the aerosol optical thickness in Smirnov *et al*. (2000) is built from the pre/post 1 min data instead of 30 s.

## 3 Sensitivity tests using simulated data

We conducted sensitivity tests using simulated data to evaluate SKYMAP algorithm steps 1 and 2 (Figs. 7a and 7b). The simulation was conducted using the two aerosol types described in Section 2.1.2. The sensitivity test was conducted with sky radiances in the almucantar plane for the wavelengths of 340, 380, 400, 500, 675, 870, 940, and 1020 nm; aerosol optical thicknesses of 0.02, 0.06, and 0.20 at 940 nm; PWV of 0.0, 0.5, 1.0, 1.5, 2.0, 2.5, 3.0, 3.5, 4.0, 4.5, and 5.0 cm; and SZA of 30°, 50°, and 70°.

Figure 11 illustrates the retrieval results from the simulated data for the continental average aerosol with aerosol optical thicknesses of 0.02, 0.06, and 0.20 at 940 nm. The retrievals of the volume size distribution, aerosol optical thickness, and PWV corresponded with their input values ("true" values in Fig. 11) when the input of PWV was <2 cm. This was seen regardless of the magnitude of the aerosol optical thickness. When the input of PWV was >2 cm, the volume size distribution, scattering and absorption optical thickness were retrieved well, but PWV was underestimated. When PWV was >2 cm, the normalized angular distribution was insensitive to PWV (Fig. 3). Figure 12 illustrates the retrieval results from the simulated data for the transported dust aerosol with aerosol optical thicknesses of 0.02, 0.06 and 0.20 at 940 nm. The scattering and absorption optical thicknesses were retrieved well. The volume size distribution of fine mode was slightly overestimated. The retrieval errors of PWV increased with increasing aerosol optical thickness because the near-infrared wavelength was strongly affected by the retrieval of coarse mode particles.

We also conducted sensitivity tests using the simulated data with bias errors to investigate uncertainty in the SKYMAP-derived PWV. The bias errors were ± 5% and ± 10% for $R$. The value of 5% was given by following reasons. The SVA bias errors of the diffuse radiances for the sky-radiometer observations were estimated to be less than 5% (Uchiyama *et al*., 2018b). According to Dubovik *et al*. (2000), the uncertainty of the diffuse radiances for the AERONET measurements is ± 5%. Figures 13 and 14 show the results from the simulated data for the continental average and transported dust aerosols with aerosol optical thicknesses of 0.02, 0.06 and 0.20 at 940 nm. PWV was overestimated when − 5% bias was applied to $R$. This corresponds to the relationship between $R$ and PWV, where $R$ decreases with increasing PWV (Section 2.1.2). The bias errors strongly affected the retrieval of PWV at high PWV (> 2 cm), because the sensitivity of high PWV is lower than that of low PWV. The retrieval error of PWV increased with increasing bias errors. The retrieval error of PWV due to ± 5% and ± 10% errors for $R$ was within 10% for PWV < 2 cm and up to 200% for PWV > 2 cm.

When the input of PWV was < 2 cm, the SKYMAP algorithm retrieved PWV very well, within an error of 10% regardless of the aerosol optical thickness or aerosol type. This was also observed when the bias errors were added for $R$. The scattering and absorption parts of the aerosol optical thickness were also estimated very well within ± 0.01 in all conditions. Present sensitivity tests suggest the design of a sky-radiometer calibration program as follows: to determine the calibration constant of the water

vapor channel in dry days/seasons with PWV <2 cm and to obtain PWV from direct solar irradiance data throughout the year, as illustrated in Fig. 1.

## 4 Application to observational data

We applied our methods to SKYNET sky-radiometer data in Tsukuba and Chiba. The results were compared to PWV observed by well-established instruments and methods other than the sky-radiometer. The aerosol channels of the sky-radiometer were calibrated by the IL method with SKYRAD.pack version 4.2 (Nakajima *et al.*, 1996; Campanelli *et al.*, 2004, 2007), and the solid view angles of all channels were calibrated by the on-site methods (Nakajima *et al.*, 1996; Boi *et al.*, 1999; Uchiyama *et al.*, 2018b).

### 4.1 Observation at Tsukuba

In Tsukuba, the sky-radiometer model POM-02 (S/N PS1202091) is installed at the MRI (36.05°N, 140.12°E). We used data from 2013 to 2014. The water vapor channel of PS1202091 was calibrated each winter by side-by-side comparison with the reference sky-radiometer, which was calibrated by the standard Langley method at the NOAA Mauna Loa Observatory (Uchiyama *et al.*, 2014). PWV was also observed using a GNSS/GPS receiver (Shoji, 2013) at Ami station (No. 0584; 36.03°N, 140.20°E), approximately 7.5 km east-southeast of the MRI.

The calibration constant of the water vapor channel was determined for each month (Figs. 15a and 16a). To obtain the correct value, we used the retrieval results with $PWV_{SKYMAP}$ < 2 cm and sufficiently small cost functions (Eqs. [13] and [20]). The annual mean calibration constants for 2013 and 2014 were $1.886 \times 10^{-4}$ A and $2.212 \times 10^{-4}$ A, respectively. The annual mean calibration constants changed drastically from 2013 to 2014 (+ 17.2%). This is because the lens at the visible and near-infrared wavelengths was replaced in December 2013.The results in 2013 and 2014 were 10.1% and 3.2% lower, respectively, than those determined by the side-by-side comparison with the reference sky-radiometer. The difference in the value of the calibration constant between the SKYMAP algorithm and the side-by-side comparison with the reference sky-radiometer was attributable mainly to the calibration period. The calibration constant of the sky-radiometer has seasonal variation due to the temperature dependency of the sensor output (Uchiyama *et al.*, 2018a). Calibration by side-by-side comparison with the reference sky-radiometer was performed only in the winter. However, the calibration constant of the SKYMAP algorithm was the annual mean.

Figures 15b and 16b show the DSRAD-retrieved PWV, which is denoted by $PWV_{DSRAD+SKYMAP}$, using the monthly calibration constant. $PWV_{DSRAD+SKYMAP}$ of the sky-radiometer agreed well with that of the GNSS/GPS receiver. Note that we did not retrieve PWV using the monthly mean calibration constants for June and July 2014 because their values were obviously small, and because little data were

successfully retrieved due to the wet and cloudy conditions in the summer. In addition, it is possible that the measurements were contaminated by clouds. Although monthly mean calibration constants are best, in theory, they could not be obtained during the wet season or during periods of high aerosol optical thickness due to the transported dust. Thus, we used the annual mean calibration constant from all data in a year to estimate PWV. Figures 15c and 16c illustrate PWV using the annual mean calibration constants. The retrieved PWV agreed well with PWV from the GNSS/GPS receiver (correlation coefficient $\gamma = 0.987$ and 0.987, and slope = 0.919 and 0.934 for 2013 and 2014, respectively; Table 5). We estimated PWV, which is denoted by $PWV_{DSRAD+LM}$, from the DSRAD algorithm using the calibration constant obtained by the side-by-side comparison with the reference sky-radiometer. The comparison of $PWV_{DSRAD+LM}$ and the GNSS/GPS-derived PWV in Figs. 12d and 13d shows good agreement, and the results are similar to those in Figs. 15c and 16c. Then we compared $PWV_{DSRAD+LM}$ and $PWV_{DSRAD+SKYMAP}$ in Figs. 15e and 16e. The difference between $PWV_{DSRAD+LM}$ and $PWV_{DSRAD+SKYMAP}$ was small: 17% in 2013, and 8% in 2014. Our self-calibration method showed comparable results to those based on the standard Langley method (Uchiyama *et al.*, 2014). Table 5 summarizes the results of comparisons of DSRAD-derived PWV and GNSS/GPS-derived PWV. The magnitude of the bias error and root mean square error were small, less than 0.11 cm and less than 0.226 cm, during 2013 to 2014. Table 6 shows the errors of the retrieved PWV with the annual mean calibration constants for the rank of PWV. The bias error was larger for high PWV than it was for low PWV. The magnitude of the bias errors of PWV was less than 0.163 cm for PWV < 3 cm and less than 0.339 cm for PWV > 3 cm.

**4.2 Observation at Chiba**

We used the data from the sky-radiometer model POM-02 (S/N PS2501417) at Chiba University (35.63°N, 140.10°E) in 2017. PWV was also obtained by a Radiometrix MP-1500 microwave radiometer (MWR) and AERONET sun-sky radiometer (Cimel, France) at the same location. The MWR measured in the 22-30 GHz region at 1-min temporal resolution and retrieved $PWV_{MWR}$ using default software. $PWV_{Cimel}$ of the AERONET sun-sky radiometer was retrieved by the direct solar irradiance at 936 nm with adjustment parameters (direct sun algorithm version 3; Holben *et al.*, 1998; Giles *et al.*, 2018) and adopted the cloud screening method (AERONET Level 2.0). The AERONET product comprises three types of data: Level 1.0 data are not screened for cloud-affected or low-quality data, Level 1.5 data are screened but not completely calibrated, and Level 2.0 data are finalized data that have been calibrated and screened. We used PWV for the Level 2.0 data.

Figure 17 shows comparisons of $PWV_{DSRAD+SKYMAP}$ using monthly and annual mean calibration constants, $PWV_{MWR}$, and $PWV_{Cimel}$. $PWV_{DSRAD+SKYMAP}$ using monthly mean calibration constants agreed well (correlation coefficient $\gamma = 0.961$ and slope = 0.964) with those of the MWR (Fig. 17b). $PWV_{DSRAD+SKYMAP}$ using the annual mean calibration constant agreed with $PWV_{MWR}$ (Fig. 17c). The

error of $PWV_{DSRAD+SKYMAP}$ was $-0.041 <$ bias $< 0.024$ cm and RMSE $< 0.212$ cm for low PWV ($<3$ cm) and bias $< -0.356$ cm and RMSE $> 0.465$ cm for high PWV (Table 6). Figure 17d shows that $PWV_{DSRAD+SKYMAP}$ using the annual mean calibration constant also agreed with $PWV_{Cimel}$ for low PWV ($< 3$ cm) but was smaller than $PWV_{Cimel}$ for high PWV ($> 3$ cm). $PWV_{MWR}$ was larger than $PWV_{Cimel}$ (Fig. 17e). $PWV_{DSRAD+SKYMAP}$ using the annual mean calibration constant was 12% and 9.1% smaller than $PWV_{MWR}$ and $PWV_{Cimel}$, respectively (Table 5). These results suggest an underestimation of $PWV_{DSRAD+SKYMAP}$, as the uncertainty of $PWV_{Cimel}$ compared to the GNSS/GPS receiver is expected to be less than 10% (Giles *et al.*, 2018). The underestimation of $PWV_{DSRAD+SKYMAP}$ was due to two factors. The first is the retrieval of PWV by the annual mean calibration constant for the water vapor channel. The calibration constant not only is subject to aging but also undergoes seasonal variation due to temperature dependency (Uchiyama *et al.*, 2018a). Thus, it is possible to underestimate the calibration constant in the wet season. Second, uncertainty regarding the aerosol optical thickness affected PWV retrieval. Figure 18 depicts the differences in PWV and aerosol optical thicknesses at 675, 870, and 1020 nm between the DSRAD algorithm and the AERONET retrieval. In the periods from January to May and from October to November, the differences in PWV and aerosol optical thicknesses were less than 0.1 cm and 0.015, respectively. However, the difference in PWV was greater than 0.1 cm from July to September. This corresponds to the difference in aerosol optical thicknesses at 675, 870, and 1020 nm from July to September, which indicates that the transmittance of water vapor was overestimated by the overestimation of aerosol optical thickness. This led to the underestimation of $PWV_{DSRAD+SKYMAP}$ using the annual mean calibration constant when PWV was $> 3$ cm. In our error estimation, the error of $+ 0.03$ for the aerosol optical thickness at 940 nm resulted in the error of $- 0.214$ cm for PWV (Appendix B).

## 5 Summary

We developed a new on-site self-calibration method, SKYMAP, to retrieve PWV from sky-radiometer data at the water vapor channel. This method first retrieves PWV from the normalized angular distribution without the calibration constant. Then the calibration constant is retrieved from the obtained PWV. Once the calibration constant is determined, PWV can be estimated from the direct solar irradiance. Our DSRAD algorithm retrieves PWV from the direct solar irradiance. This method does not require adjustment parameters used in the empirical methods of previous studies (e.g., Holben *et al.*, 1998; Uchiyama *et al.*, 2014; Campanelli *et al.*, 2014, 2018). Instead, the filter response function and the vertical profiles of water vapor, temperature, and pressure are required as input parameters. Thus, our physics-based algorithm has the potential to be applied to sky-radiometers all over the world. This is the greatest advantage of the present study.

Sensitivity tests using simulated data from sky-radiometer measurements showed that the SKYMAP algorithm retrieved PWV within an error of 10% for cases when PWV was <2 cm. Larger retrieval errors occurred in the cases when PWV was >2 cm because PWV became less sensitive to the normalized angular distribution. Therefore, the SKYMAP algorithm can be applied only to dry conditions.

We applied SKYMAP and DSRAD algorithms to the sky-radiometer measurements at two SKYNET sites (Tsukuba and Chiba, Japan). At Tsukuba, the calibration constant estimated by the SKYMAP algorithm was compared to that obtained by side-by-side comparison with the reference sky-radiometer calibrated by the standard Langley method. The calibration constant calculated by the SKYMAP algorithm was 10.1% lower in 2013 and 3.2% lower in 2014 compared with the calibration constant estimated by side-by-side comparison. Our retrieved PWV data were compared to those obtained by a GNSS/GPS receiver, a microwave radiometer, and an AERONET sun-sky radiometer. The correlation coefficients and slopes were as good as >0.96 and 1.00 ± 0.12, respectively. The magnitude of the bias error and the root mean square error were < 0.163 cm and < 0.251 cm, respectively, for low PWV (< 3 cm). However, our retrieved PWV was underestimated in the wet conditions, and the magnitude of the bias error and the root mean square error were less than 0.594 cm and less than 0.722 cm for high PWV. This was due to seasonal variation in the calibration constant and the overestimation of aerosol optical thickness at 940 nm interpolated from those at 870 and 1020 nm.

These results show that our new on-site self-calibration method is practical. In future work, we plan to compare our method with others in the SKYNET framework (Uchiyama *et al*. 2014; Campanelli *et al*., 2014).

## 6 Data availability

The SKYMAP and DSRAD algorithms are available on request from the first author. The sky-radiometer data are available from the SKYNET website (http://www.skynet-isdc.org/), but the sky-radiometer data in Tsukuba, Japan, are available on request from the first author. The MWR data at Chiba University are available from CEReS, Chiba University (http://atmos3.cr.chiba-u.jp/skynet/). The AERONET sun-sky radiometer data are available from the AERONET website (https://aeronet.gsfc.nasa.gov/).

## Author contributions

This study was designed by MM, RK, KA, TM, KM, and TN. Sky-radiometer measurements at Tsukuba were conducted by RK. Sky-radiometer and MWR measurements at Chiba were conducted by HO and HI. Analyses of both sky-radiometers were performed by MM. The calibration constant of the

sky-radiometer by the Langley method was provided by AU. Analyses of the GPS receiver were
625 conducted by YS. Visual observations at Tsukuba were conducted by OI and MT. The manuscript was
written by MM and RK, and all authors contributed to editing and revision.

**Competing interests**

The authors declare that they have no conflict of interest.

**Acknowledgments**

This work was performed by the joint research programs of CEReS, Chiba University (2018), and the
Environment Research and Technology Development Fund (S-12) of the Environmental Restoration
and Conservation Agency. We are grateful to the OpenCLASTR project (http://157.82.240.167/~clastr/,
last accessed September 2018) for allowing us to use SKYRAD.pack (sky-radiometer analysis package),
RSTAR (System for Transfer of Atmospheric Radiation for Radiance calculations), and PSTAR
(System for Transfer of Atmospheric Radiation for Polarized radiance calculations) in this research. We
acknowledge the AERONET networks for providing retrievals. NCEP reanalysis data were provided by
the NOAA/OAR/ESRL PSD (Boulder, CO, USA) website at http://www.esrl.noaa.gov/psd/ (last
accessed September 2018).

**Appendix A: Width of the volume size distribution**

Because $\frac{dV(r)}{d\ln r}$ is expressed by the superposition of 20-modal lognormal size distributions (Eq. [6]), the

width of $\frac{dV(r)}{d\ln r}$ is larger than that of each lognormal size distribution. The width of the lognormal size

distribution should be small to deal with the complicated and step variations in $\frac{dV(r)}{d\ln r}$. However, $\frac{dV(r)}{d\ln r}$

cannot represent a natural curve if $\eta$ is large and $s$ is small (Fig. A1). Hence, we have to find the

maximum value of $\eta$ for making $\frac{dV(r)}{d\ln r}$ a natural curve. When $C_i$ is constant, such value of $\eta$ minimizes

the roughness of $\frac{dV(r)}{d\ln r}$, and $\frac{dV(r)}{d\ln r}$ approaches to a flat shape. For a simple formulation, we consider the

function $A(x)$ which consists of the multimodal normal distribution function $B_i$ with a constant height.
$A(x)$ and $B_i$ are expressed as

$$A(x) = \sum_{i=-\infty}^{\infty} B_i(x) = \sum_{i=-\infty}^{\infty} \exp\left[-\frac{\eta^2}{2}\left(\frac{x-i\xi}{\xi}\right)^2\right], \quad (A1)$$

where $i\xi$ and $\frac{\xi}{\eta}$ are the mean and standard deviation, respectively. Its differential is written as

$$\frac{dA}{dx} = \sum_{i=-\infty}^{\infty} \frac{dB_i}{dx} = \sum_{i=-\infty}^{\infty} -\eta^2 \left(\frac{x-i\xi}{\xi}\right) \exp\left[-\frac{\eta^2}{2}\left(\frac{x-i\xi}{\xi}\right)^2\right]. \quad (A2)$$

When the shape of $A(x)$ approaches to be flat, the difference between local maximum and minimum values of $A(x)$ is approximately 0. Because $\frac{dB_i}{dx}$ equals 0 at $x = j\xi$ $(j \in \mathbb{Z})$, $A(x)$ has the local maximum and minimum at $x = j\xi$ and $\left(j + \frac{1}{2}\right)\xi$ in $j \leq \frac{x}{\xi} < j + 1$. The difference $\Delta$ between the local maximum and minimum values is obtained as

$$\Delta = 1 - \frac{A\left(\frac{2j+1}{2}\xi\right)}{A(j\xi)}. \quad (A3)$$

Figure A2 shows the relation between $\eta$ and $\Delta$. The value of $\Delta$ increases drastically at around $\eta = 1.5$. in addition, the shape of $\frac{dV(r)}{d\ln r}$ is unnatural when $\eta = 2.0$ (Fig. A1). Therefore, the value of $\eta$ should be selected from the values around $\eta = 1.5$. In this study, we fixed $\eta$ at 1.65. This value represents the natural curve of $\frac{dV(r)}{d\ln r}$ and satisfies that the value of $\Delta$ is small enough, $\Delta = 3.0 \times 10^{-3}$.

**Appendix B: Error propagation from aerosol optical thickness to PWV**

We evaluated the influence of the uncertainty of aerosol optical thickness on PWV using the empirical equation of Bruegge *et al*. (1992). PWV is described using the adjustment parameters as follows

$$w = \frac{1}{m_0}\left(-\frac{\ln \bar{T}_{H2O}}{a}\right)^{\frac{1}{b}} \text{ [cm].} \quad (B1)$$

The uncertainty of PWV $\epsilon_{PWV}$ is given from the partial differentiation of Eq. (B1) with respect to $\ln \bar{T}_{H2O}$ as follows

$$\epsilon_{PWV} = \frac{\partial w}{\partial \ln \bar{T}_{H2O}} \epsilon_{\ln \bar{T}_{H2O}} = \frac{w}{b \ln \bar{T}_{H2O}} \epsilon_{\ln \bar{T}_{H2O}}. \quad (B2)$$

where $\epsilon_{\ln \bar{T}_{H2O}}$ is the uncertainty of $\bar{T}_{H2O}$. Using Eq. (B1) with the adjusting parameters of the sky-radiometer, with $a = 0.620$ and $b = 0.625$ as the coefficient values for the trapezoidal spectral response function (Uchiyama *et al*., 2018a), we write the uncertainty of PWV as

$$\epsilon_{PWV} = -\frac{w}{ab}(m_0 w)^{-b} \epsilon_{\ln \bar{T}_{H2O}} = -\frac{w}{0.388}(m_0 w)^{-0.625} \epsilon_{\ln \bar{T}_{H2O}}. \quad (B3)$$

If the uncertainty of the calibration constant at the water vapor channel is ignored, the uncertainty of $\bar{T}_{\text{H2O}}$ is given from Eq. (21) as follows

$$\epsilon_{\ln \bar{T}_{\text{H2O}}} = m_0 \epsilon_{\text{AOT}}. \qquad \text{(B4)}$$

where $\epsilon_{\text{AOT}}$ is the uncertainty of the aerosol optical thickness at 940 nm. The uncertainty of PWV is written by Eqs. (B3) and (B4) as

$$\epsilon_{\text{PWV}} = -\frac{1}{0.388}(m_0 w)^{0.375} \epsilon_{\text{AOT}} = -0.214 \, [\text{cm}]. \qquad \text{(B5)}$$

where $m_0 = 3.0$, $w = 5.0$ cm, and $\epsilon_{\text{AOT}} = 0.03$.

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

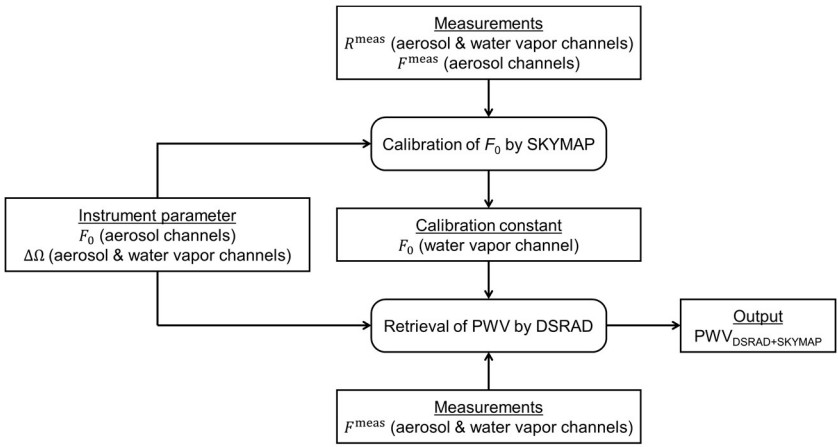

**Figure 1: Diagram of the on-site self-calibration method (SKYMAP) and retrieval of PWV from direct solar irradiances (DSRAD).** Square boxes show the operation of the calculation and input/output parameters and rounded boxes show the operation of the algorithm.

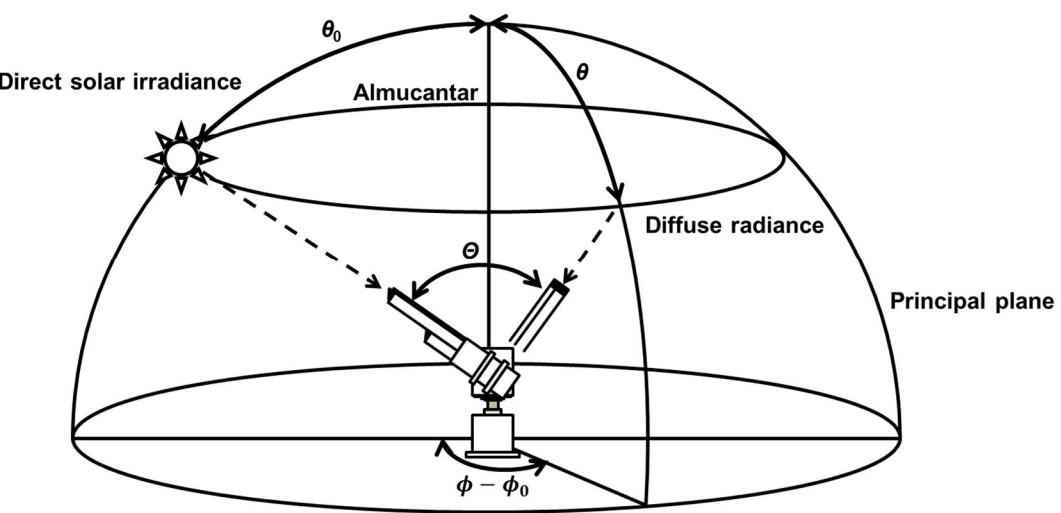

**Figure 2: Observation planes (almucantar and principal planes) of the sky-radiometer.**

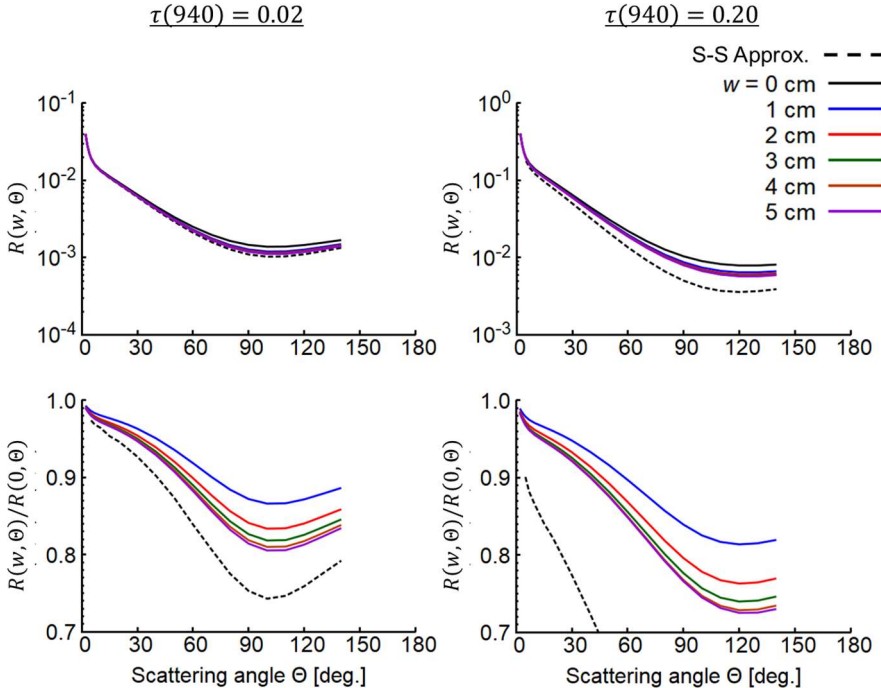

**Figure 3: Normalized angular distributions simulated for continental average aerosol (Table 2) in the almucantar plane with aerosol optical thicknesses of 0.02 and 0.20 at 940 nm. Simulations were conducted for SZA = 70° and PWV (w) = 0, 1, 2, 3, 4, and 5 cm. The top row is the normalized radiance $R(w, \Theta)$, and the bottom row is the ratio of $R(w, \Theta)$ to $R(0, \Theta)$. S-S Approx. is single scattering approximation.**

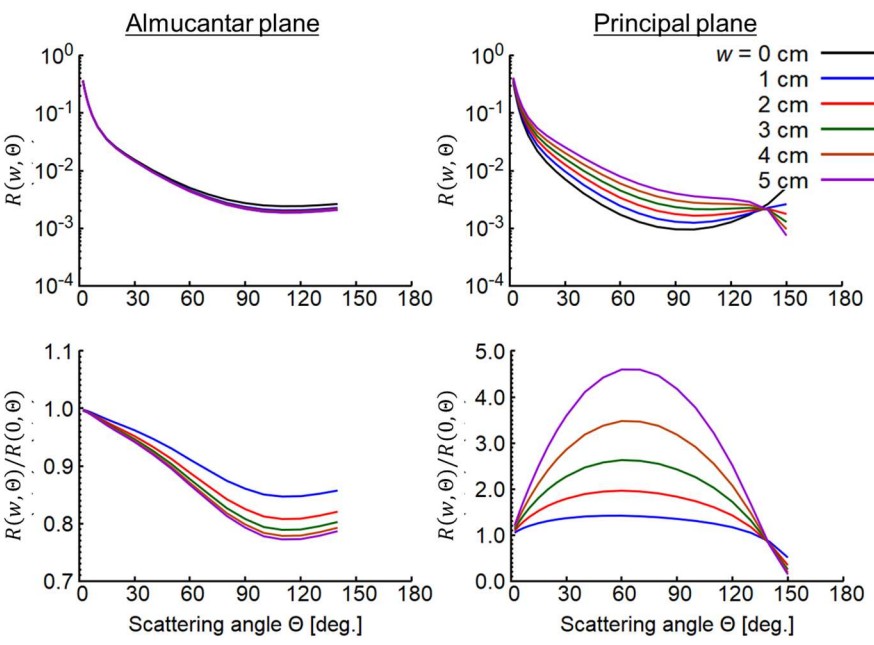

**Figure 4: Normalized angular distributions simulated for transported dust aerosol (Table 2) in the almucantar and principal planes with an aerosol optical thickness of 0.06 at 940 nm. Simulations were conducted for SZA = 70° and PWV ($w$) = 0, 1, 2, 3, 4, and 5 cm. The top row is the normalized radiance $R(w, \Theta)$, and the bottom row is the ratio of $R(w, \Theta)$ to $R(0, \Theta)$.**

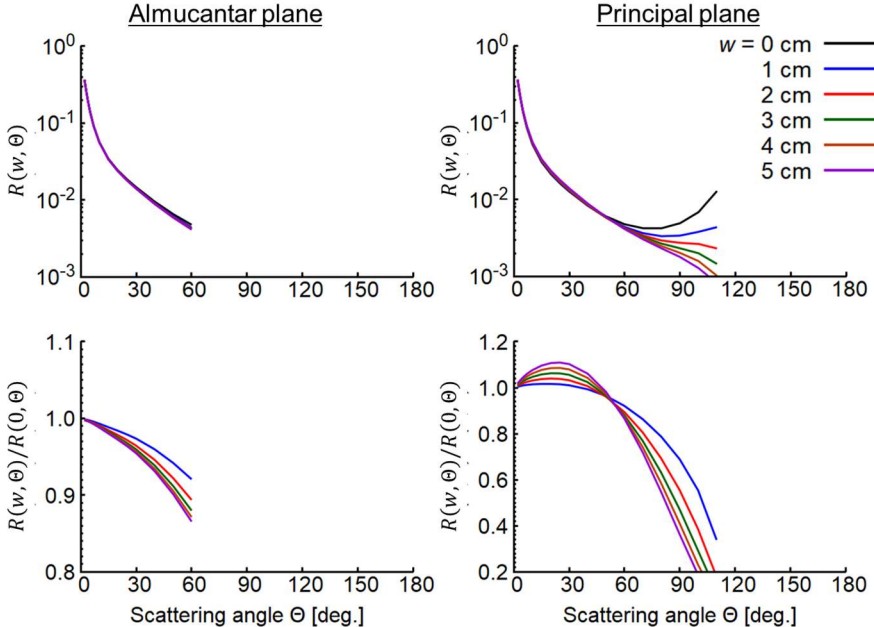

**Figure 5: Similar to Fig. 4 but for SZA = 30°.**

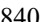840

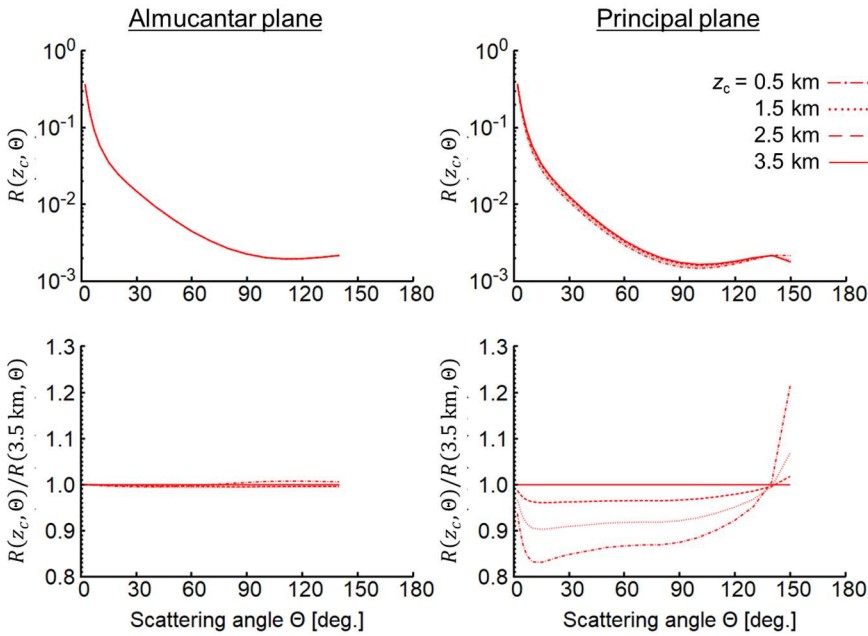

**Figure 6: Normalized angular distributions simulated for transported dust aerosol (Table 2) in the almucantar and principal planes with an aerosol optical thickness of 0.06 at 940 nm. Simulations were conducted for SZA = 70° and PWV = 2 cm. The height of the dust layer ($z_c$) is changed to 0.5, 1.5, 2.5, and 3.5 km. The top row is the normalized radiance $R(z_c, \Theta)$, and the bottom row is the ratio of $R(z_c, \Theta)$ to $R(3.5 \text{ km}, \Theta)$.**

(a) Step 1: Retrieval of aerosol optical and microphysical properties

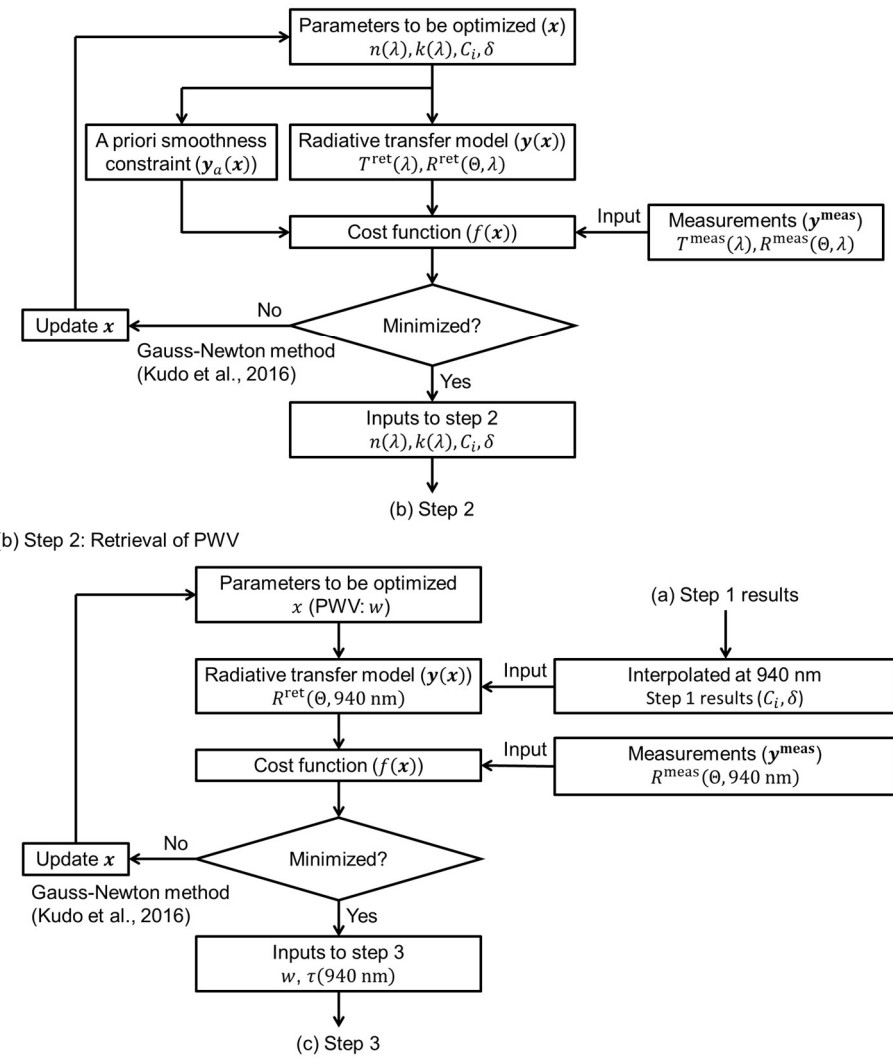

(b) Step 2: Retrieval of PWV

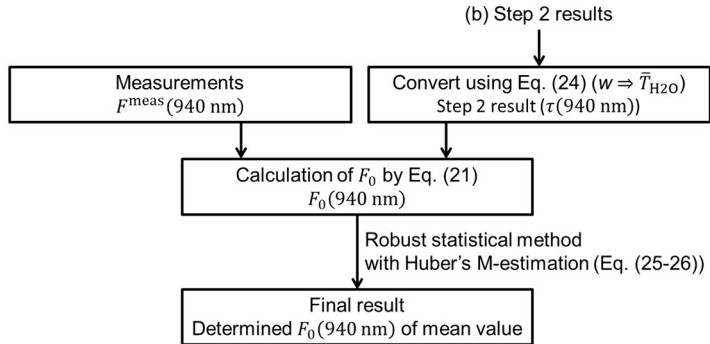

(c) Step 3: Retrieval of the calibration constant of the water vapor channel

**Figure 7: Schematic diagrams of SKYMAP procedures. (a) Step 1. (b) Step 2. (c) Step 3. Square boxes show the calculation and input/output parameters**

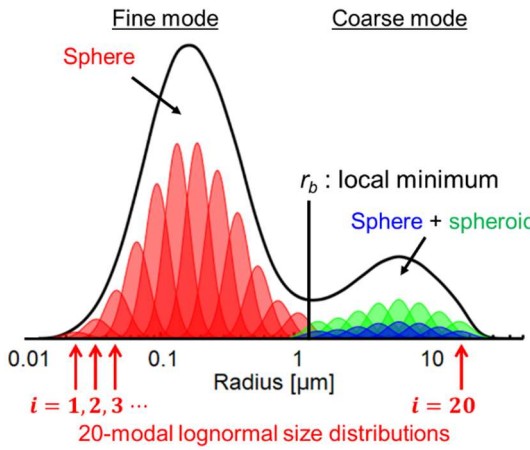

**Figure 8: Assuming volume size distributions in the SKYMAP algorithm. Fine and coarse mode particles are separated at radius $r_b$. Spheroid particles are assumed only in coarse mode. The black line is the volume size distribution, which is computed by the integration of 20-modal lognormal distribution functions (red, blue, and green lines).**

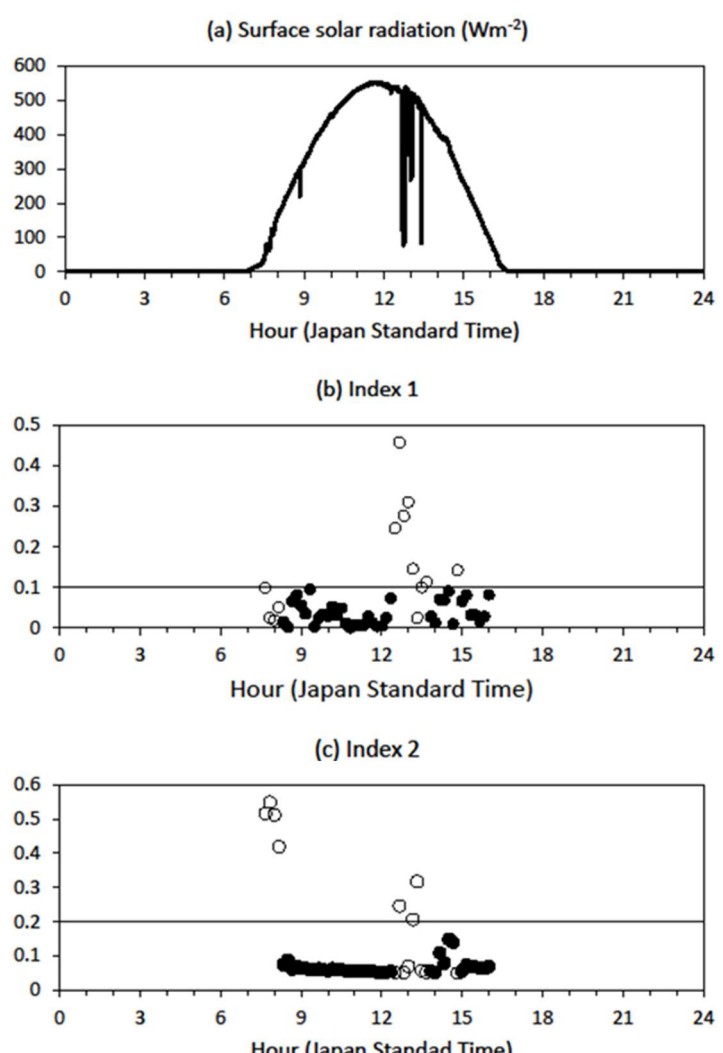

**Figure 9: An example result of the SCAD method on January 6, 2014, in Tsukuba. (a) Surface solar radiation observed by the pyranometer. (b) Index 1. (c) Index 2. The closed circles indicate clear-sky conditions and the open circles indicate cloudy conditions in (b) and (c). The lines at 0.1 in (b) and 0.2 in (c) are thresholds for indexes 1 and 2, respectively.**

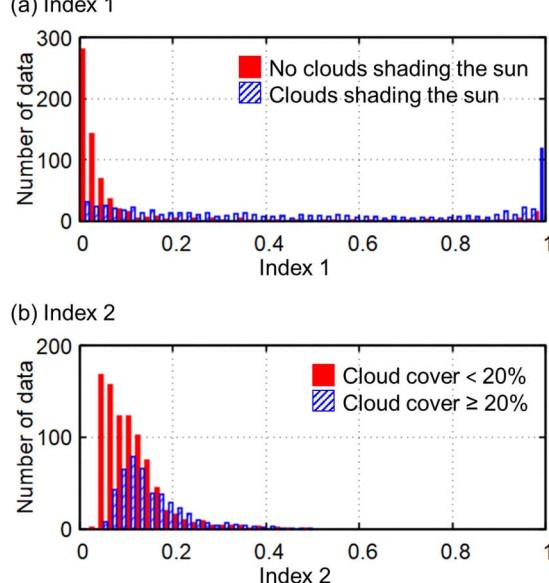

(a) Index 1

(b) Index 2

**Figure 10: Histograms of indexes 1 and 2 of sky-radiometer observations at Tsukuba. (a) Index 1 when the sun is covered by clouds (blue boxes) and not covered by clouds (red boxes). (b) Index 2 when cloud cover is less than to 20% (red boxes) and greater than or equal to 20% (blue boxes).**

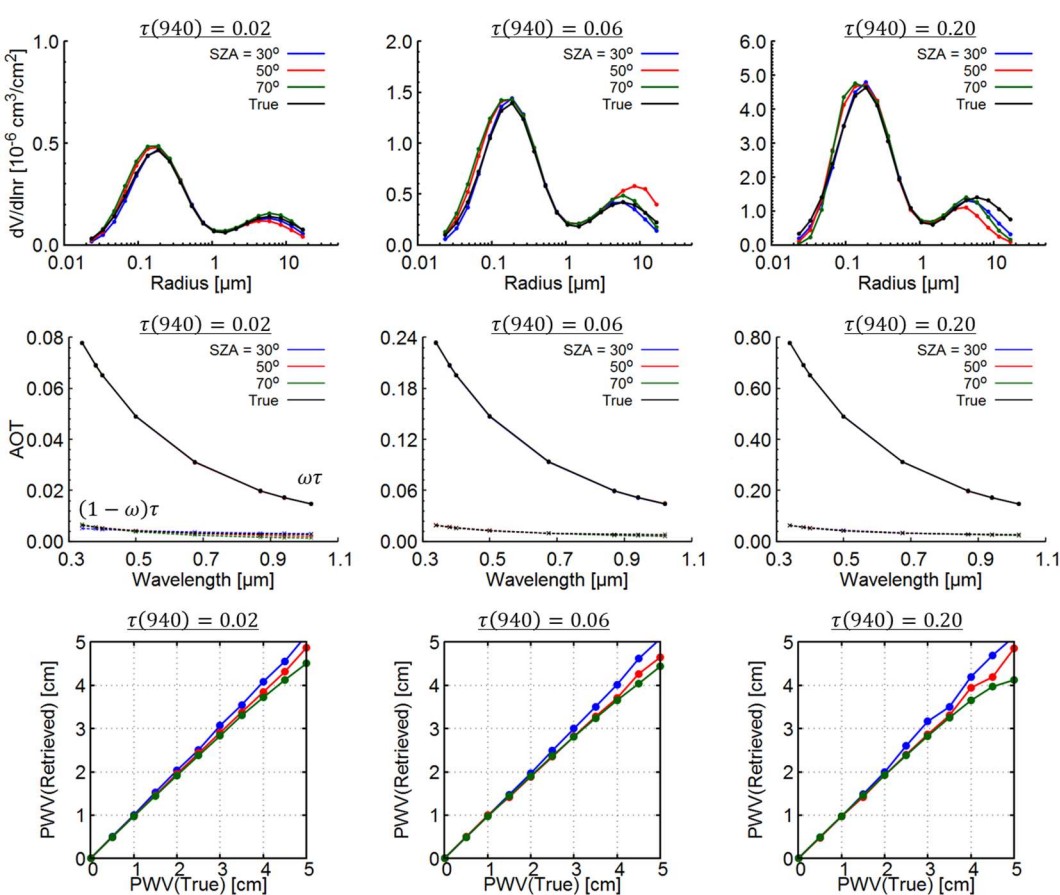

**Figure 11: Retrieval results from simulated data for continental average aerosol. The top row is the volume size distribution, the middle row is the scattering and absorption parts of aerosol optical thickness, and the bottom row is a comparison of the "true" and retrieval values of PWV. Blue, red, and green lines are the retrieval results at SZA = 30°, 50°, and 70°, respectively. The**

**black line is the "true" value. Note that the blue, red, green, and black lines in the middle row overlap.**

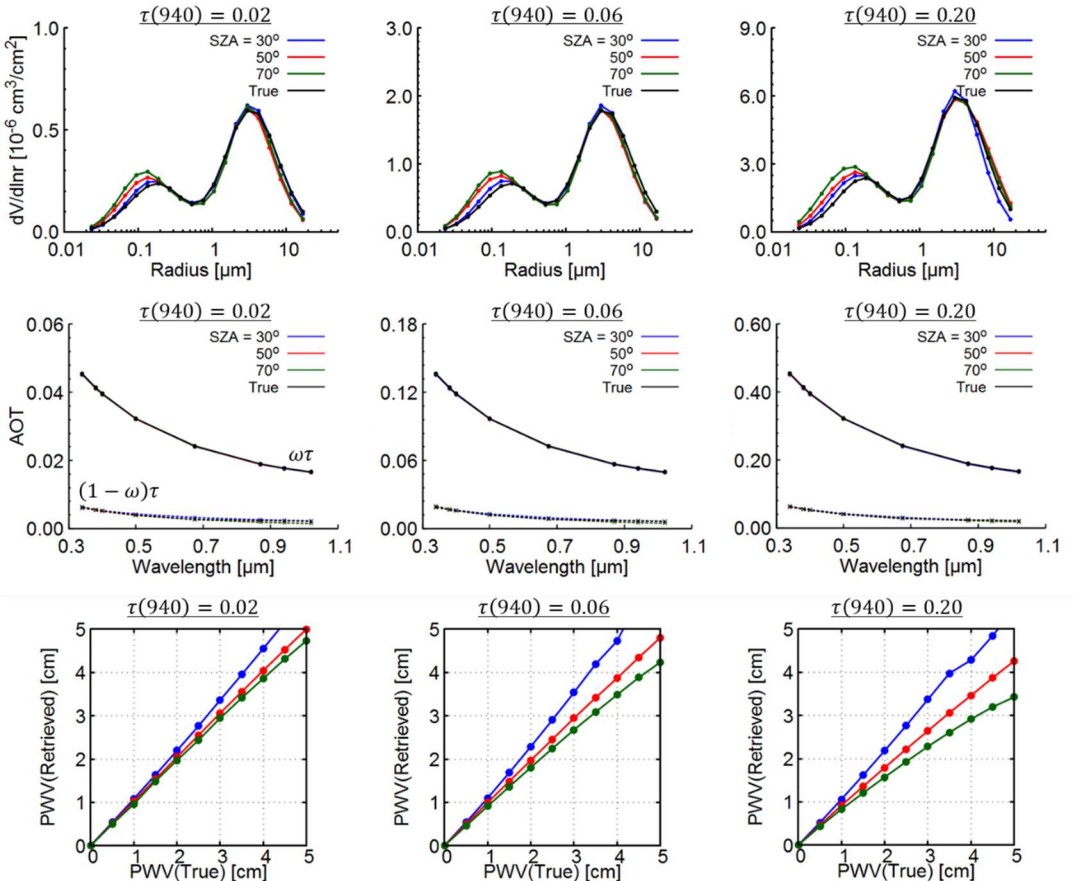

**Figure 12: Similar to Fig. 11 but for transported dust aerosol. Note that the blue, red, green, and black lines in the middle row overlap.**

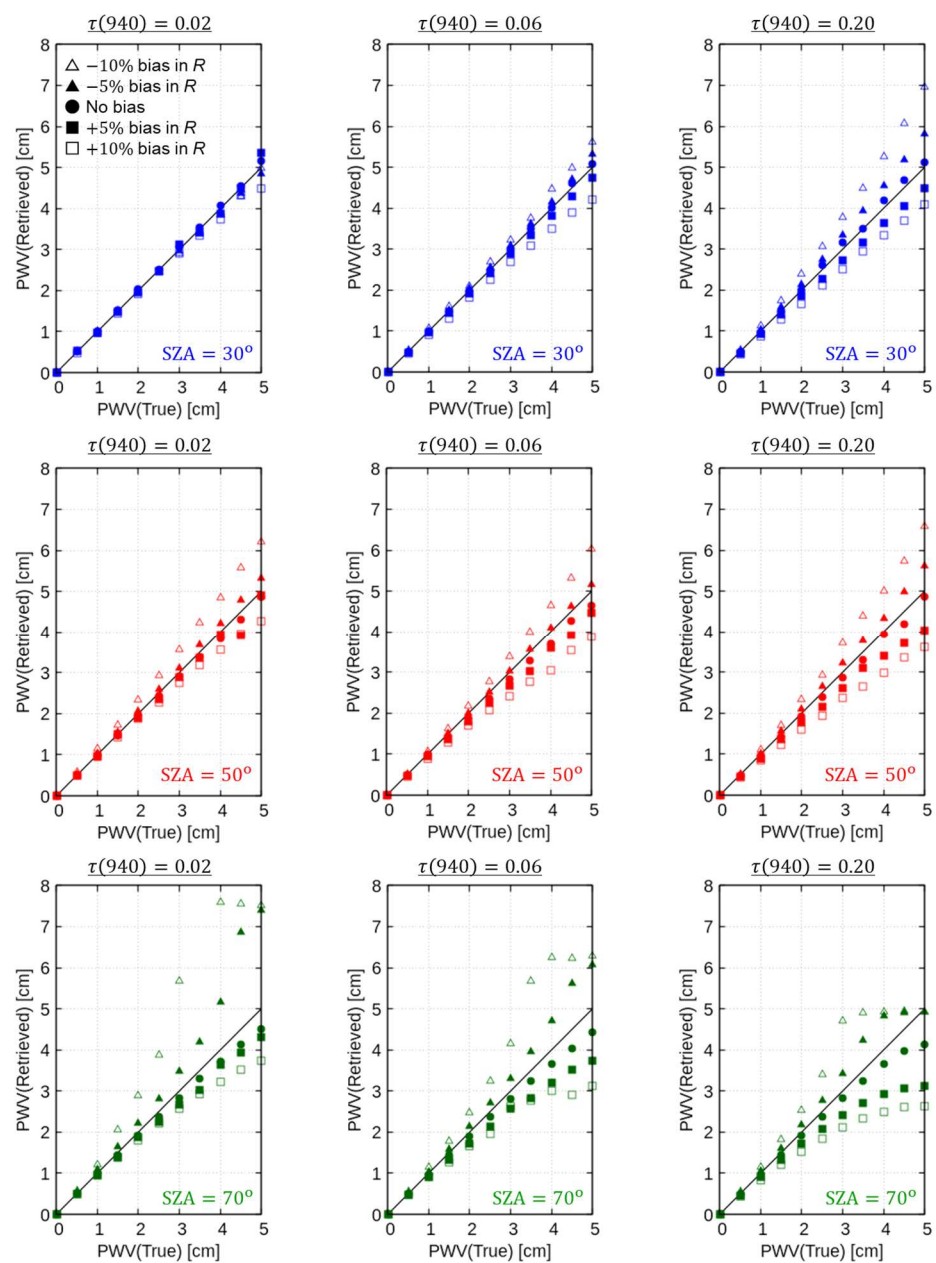

**Figure 13: Comparison of the "true" and retrieval values of PWV from simulated data for continental average aerosol with bias errors. The top, middle, and bottom rows are the retrieval results at SZA = 30°, 50°, and 70°, respectively. Closed circles are the results with no bias errors. Closed squares and closed triangles are the results with bias errors of plus and minus 5% in *R*, respectively. Open squares and open triangles are the results with bias errors of plus and minus 10% in *R*, respectively.**

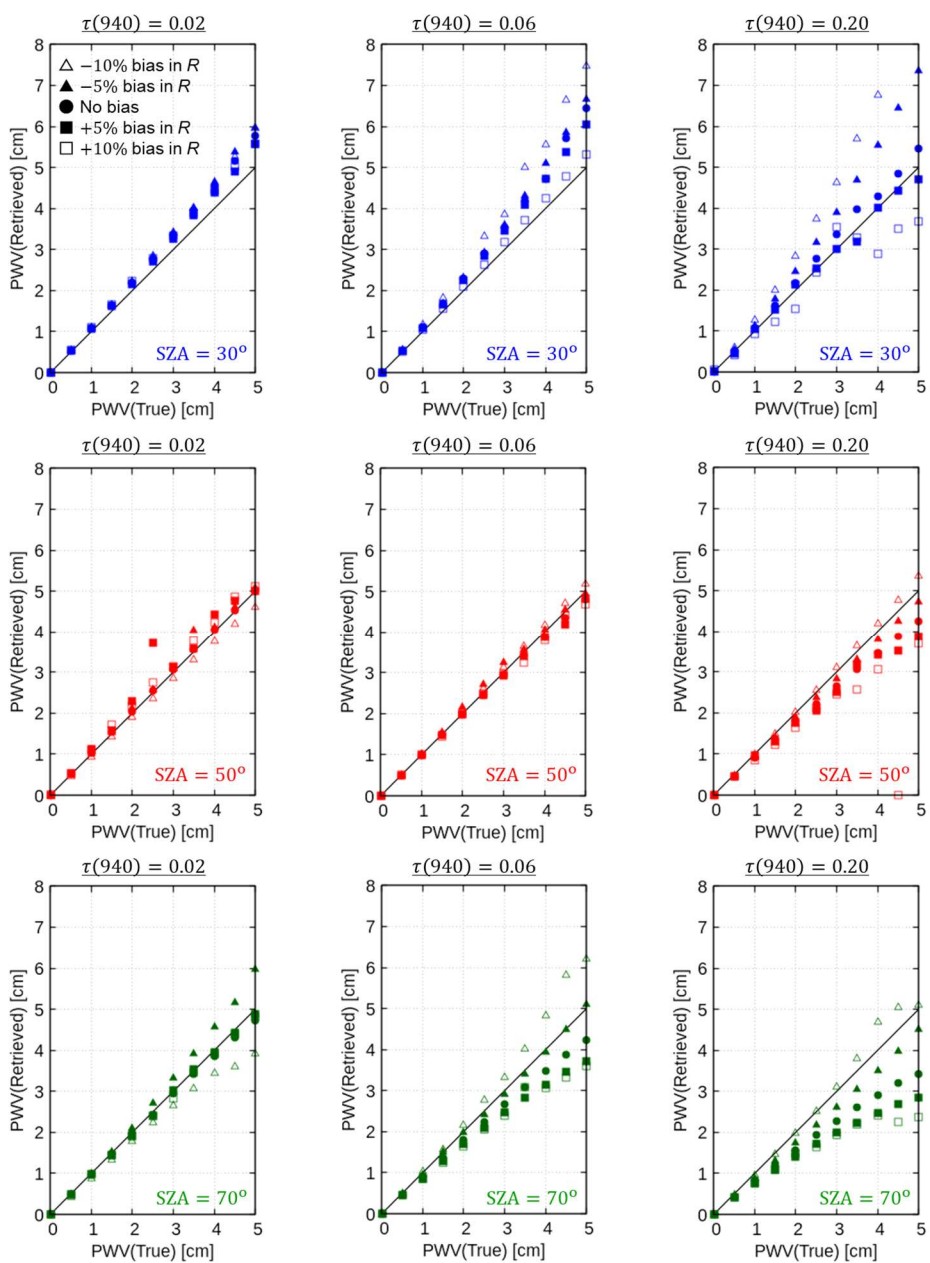

**Figure 14: Similar to Fig. 13 but for transported dust aerosol.**

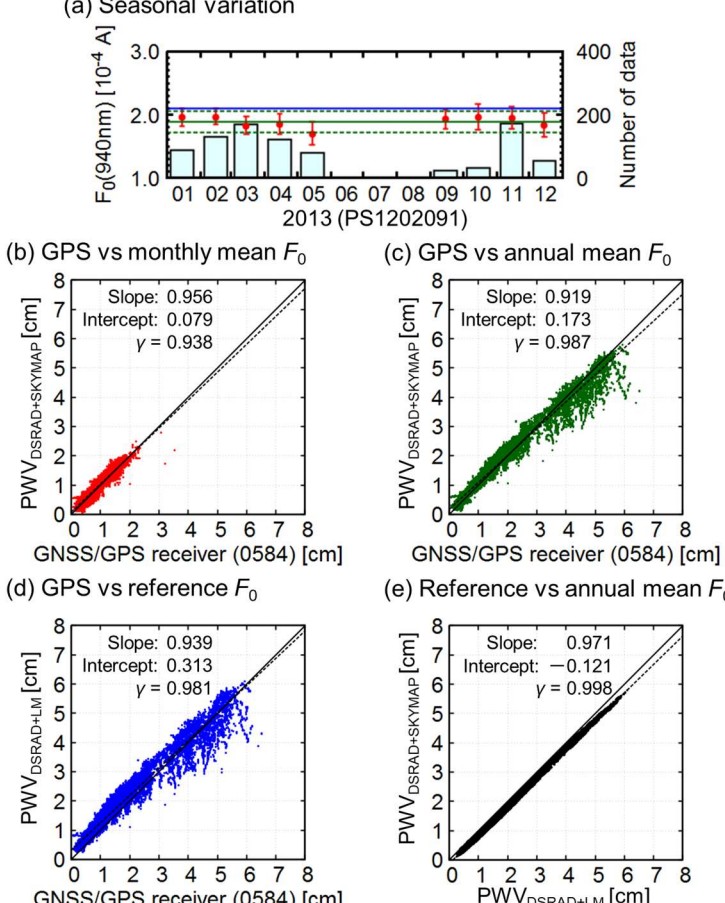

**Figure 15: Application of our methods to observational data from Tsukuba in 2013. (a) Seasonal variation in the calibration constant of the water vapor channel (red circles and error bars are monthly means and standard deviations, respectively; green solid and dotted lines are annual means and standard deviations, respectively; the blue line is the value obtained by a side-by-side comparison with the reference sky-radiometer; boxes indicate the number of data points). (b-d) Comparisons of PWV between the GNSS/GPS receiver and the sky-radiometer with (b) the monthly mean $F_0$, (c) the annual mean $F_0$, and (d) the reference $F_0$. (e) Comparison of PWV from the sky-radiometer with the reference and annual mean $F_0$.**

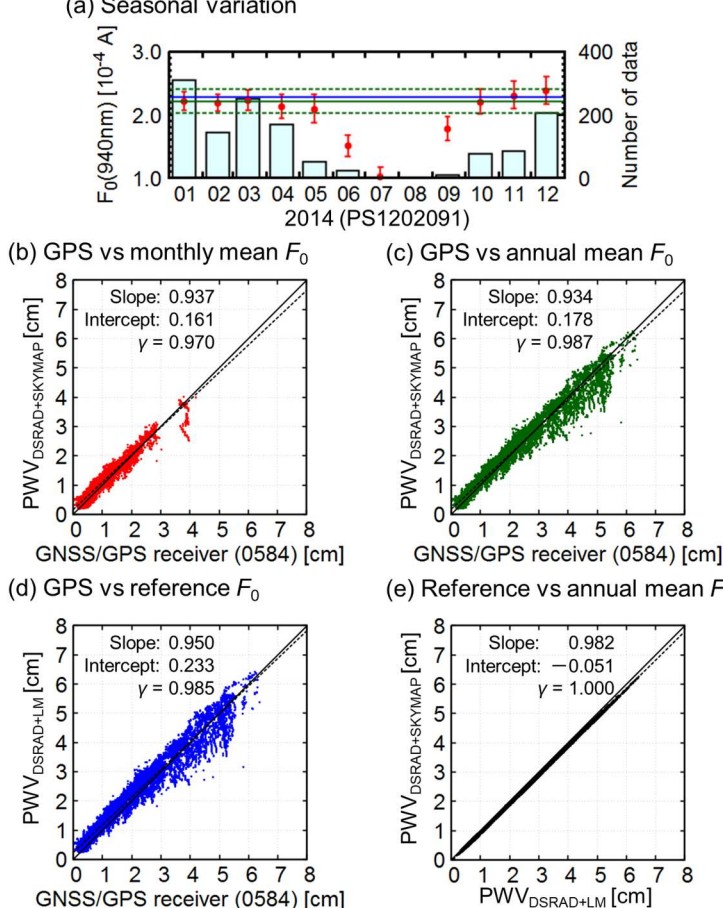

**Figure 16: Similar to Fig. 15 but in 2014.**

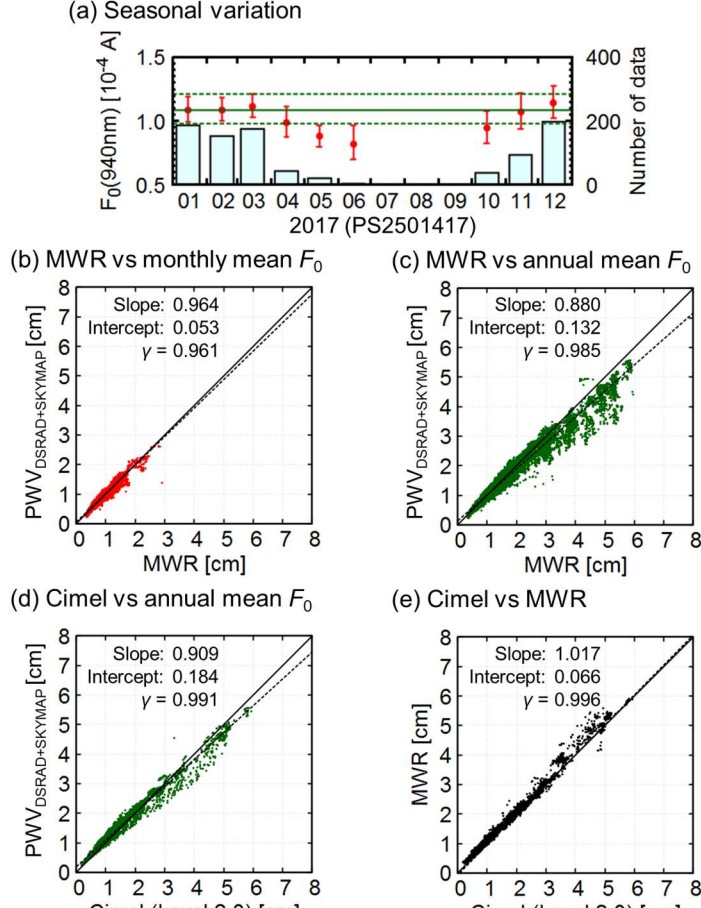

**Figure 17: Application of our methods to observational data from Chiba in 2017. (a) Seasonal variation in the calibration constant of the water vapor channel (red circles and error bars are monthly means and standard deviations, respectively; green solid and dotted lines are annual means and standard deviations, respectively; boxes indicate the number of data points). (b, c) Comparison of PWV between the MWR and the sky-radiometer with (b) the monthly mean $F_0$, and (c) the annual mean $F_0$. (d) Comparison of PWV between the Cimel level 2.0 data and the sky-radiometer with annual mean $F_0$. (e) Comparison of PWV between the Cimel level 2.0 data and the MWR.**

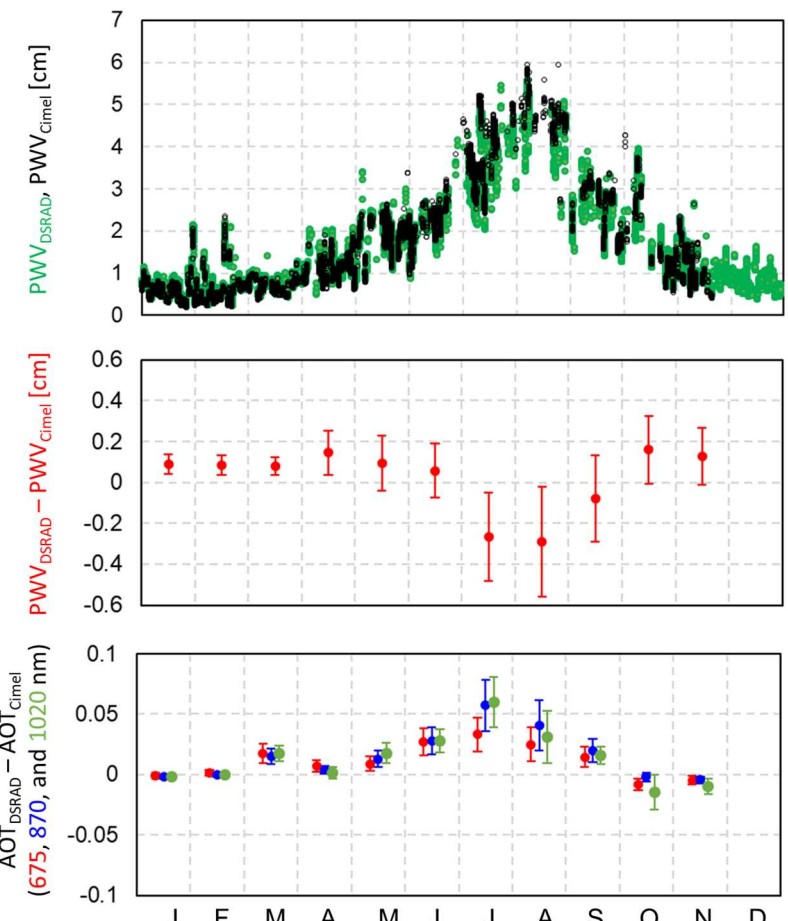

**Figure 18: The top row shows the time series of PWV in 2017 at Chiba (green and black circles are PWV$_{\text{DSRAD+SKYMAP}}$ and PWV$_{\text{Cimel}}$, respectively). The middle row is the difference between PWV$_{\text{DSRAD+SKYMAP}}$ and PWV$_{\text{Cimel}}$. The bottom row is the difference in aerosol optical thicknesses at 675 nm (red), 870 nm (blue), and 1020 nm (green) between the DSRAD algorithm and the AERONET retrieval results. Circles and error bars in the middle and bottom rows are means and standard deviations, respectively.**

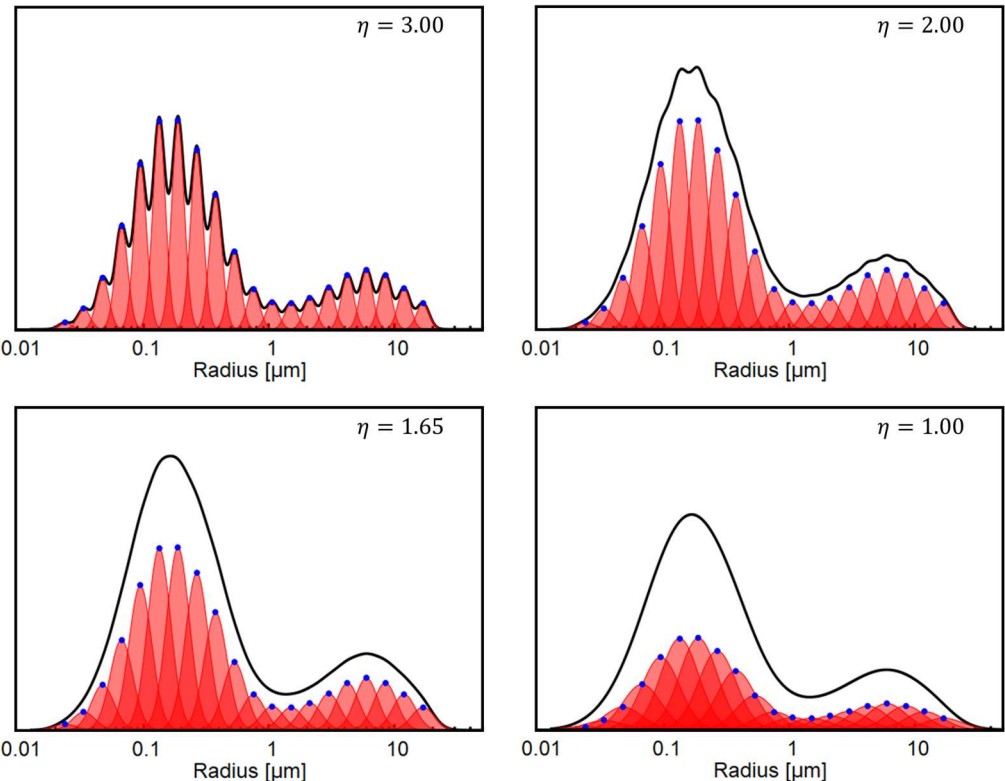

Figure A1: Relationship between the volume size distribution and $\eta$. The black line is the volume size distribution, which is computed by the integration of 20-modal lognormal distribution functions (red lines). Blue circles are the peak volume of lognormal size distribution.

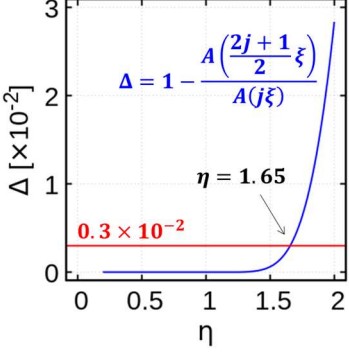

Figure A2: Relationship between the parameter $\eta$ and the difference $\Delta$.

**Table 1: Sky-radiometer specifications. Each sky-radiometer is equipped with a filter indicated by a circle. "Standard" is the standard specification of sky-radiometer models POM-01 and POM-02.**

| Wavelength [nm] | Strong gas absorption | Main target substance | POM-01 Standard | POM-02 Standard | POM-02 PS1202091 | POM-02 PS2501417 |
|---|---|---|---|---|---|---|
| 315 | $O_3$ | Ozone | ○ | ○ | — | ○ |
| 340 | — | Aerosol | — | ○ | ○ | ○ |
| 380 | — | Aerosol | — | ○ | ○ | ○ |
| 400 | — | Aerosol | ○ | ○ | ○ | ○ |
| 500 | — | Aerosol | ○ | ○ | ○ | ○ |
| 675 | — | Aerosol | ○ | ○ | ○ | ○ |
| 870 | — | Aerosol | ○ | ○ | ○ | ○ |
| 940 | $H_2O$ | Water vapor | ○ | ○ | ○ | ○ |
| 1020 | — | Aerosol | ○ | ○ | ○ | ○ |
| 1225 | $O_2$, $CO_2$, $H_2O$ | Cloud | — | — | ○ | — |
| 1627 | $CH_4$, $CO_2$ | Cloud | — | ○ | ○ | ○ |
| 2200 | $CH_4$, $H_2O$ | Cloud | — | ○ | ○ | ○ |

 **Table 2: Microphysical and optical properties and vertical profiles of aerosol used in sensitivity tests.**

| Aerosol | Components | Particle shape | Size distribution | | Refractive index at 940 nm | | Relative weight in total optical thickness at 500 nm | Vertical profile |
|---|---|---|---|---|---|---|---|---|
| | | | Mode radius ($\mu$m) | Mode width | Real | Imaginary | | |
| Continental average | Water-soluble | Sphere | 0.18 | 0.81 | 1.43 | 0.0074 | 0.90 | $\exp(-z/H)$, $H = 8$ km |
| | Soot | Sphere | 0.05 | 0.69 | 1.75 | 0.44 | 0.07 | $\exp(-z/H)$, $H = 4$ km |
| | Insoluble | Spheroid | 5.98 | 0.92 | 1.52 | 0.008 | 0.03 | $\exp(-z/H)$, $H = 2$ km |
| Transported dust | Dust | Spheroid | 3.23 | 0.79 | 1.53 | 0.004 | 0.25 | $\frac{1}{\sqrt{2\pi}\sigma}\exp\left(-\frac{(z-z_c)}{2\sigma^2}\right)$, $z_c = 3.5$ km $\sigma = 0.4$ km |
| | Water-soluble | Sphere | 0.18 | 0.81 | 1.43 | 0.0074 | 0.67 | $\exp(-z/H)$, $H = 8$ km |
| | Soot | Sphere | 0.05 | 0.69 | 1.75 | 0.44 | 0.05 | $\exp(-z/H)$, $H = 4$ km |
| | Insoluble | Spheroid | 5.98 | 0.92 | 1.52 | 0.008 | 0.03 | $\exp(-z/H)$, $H = 2$ km |

**Table 3: Validation of the SCAD method by visual observation from 2013 to 2014 in Tsukuba.**

| Visual observation | Sky-radiometer measuring plane | |
| --- | --- | --- |
| Cloud cover | Best condition | Poor condition |
| Clear, less than 20% | **463 (83.4%)*** | 68 (8.7%) |
| Cloud affected, more than 20% | 92 (16.6%) | **714(91.3%)*** |

*Obviously correct determination.

 **Table 4: References and methodologies of the DSRAD algorithm.**

| | DSRAD |
| --- | --- |
| Solar coordinates | Nagasawa (1999) |
| Refraction correction | Nagasawa (1999) |
| Sun-Earth distance | Nagasawa (1999) |
| Optical mass | Gueymard (2001) |
| Rayleigh scattering | Fröhlich and Shaw (1980); Young(1981) |
| Ozone absorption | Sekiguchi and Nakajima (2008) |
| Water vapor absorption | Sekiguchi and Nakajima (2008) |
| Filter response function | Stepwise function |
| Retrieval of PWV | Newton-Raphson method |

 **Table 5: Comparison of PWV between DSRAD and other instruments.**

| | Slope $C_1$ | Intercept $C_2$ [cm] | $\gamma$ | Bias [cm] | RMSE [cm] |
|---|---|---|---|---|---|
| PS1202091 at Tsukuba, Japan | | | | | |
| Monthly mean $F_0$ vs GNSS/GPS receiver (2013) | 0.956 | 0.079 | 0.938 | -0.049 | 0.138 |
| vs GNSS/GPS receiver (2014) | 0.937 | 0.161 | 0.970 | -0.110 | 0.170 |
| Annual mean $F_0$ vs GNSS/GPS receiver (2013) | 0.919 | 0.173 | 0.987 | -0.061 | 0.226 |
| vs GNSS/GPS receiver (2014) | 0.934 | 0.178 | 0.987 | -0.089 | 0.223 |
| PS2501417 at Chiba, Japan | | | | | |
| Monthly mean $F_0$ vs MWR (2017) | 0.964 | 0.053 | 0.961 | -0.027 | 0.091 |
| vs AERONET (2017) | 0.987 | 0.107 | 0.976 | 0.098 | 0.122 |
| Annual mean $F_0$ vs MWR (2017) | 0.880 | 0.132 | 0.985 | 0.042 | 0.231 |
| vs AERONET (2017) | 0.909 | 0.184 | 0.991 | 0.055 | 0.186 |

$C_1$, $C_2$: $\text{PWV}_{\text{DSRAD}} = C_1 \times \text{PWV}_{\text{Other}} + C_2$

Bias: $\text{PWV}_{\text{DSRAD}} - \text{PWV}_{\text{Other}}$

**Table 6: Difference in PWV between DSRAD with the annual mean calibration constants and other instruments.**

| | $PWV_{Other}$ | | | | |
| | 0 – 1 cm | 1 – 2 cm | 2 – 3 cm | 3 – 4 cm | > 4 cm |
| | Bias [cm] (RMSE [cm]) | Bias [cm] (RMSE [cm]) | Bias [cm] (RMSE [cm]) | Bias [cm] (RMSE [cm]) | Bias [cm] (RMSE [cm]) |
|---|---|---|---|---|---|
| PS1202091 at Tsukuba, Japan | | | | | |
| vs GNSS/GPS receiver (2013) | 0.083 (0. 124) | 0.160 (0.211) | 0.084 (0.236) | -0.098 (0.326) | -0.339 (0.537) |
| vs GNSS/GPS receiver (2014) | 0.110 (0.142) | 0.163 (0.221) | 0.107 (0.251) | -0.055 (0.353) | -0.239 (0.492) |
| PS2501417 at Chiba, Japan | | | | | |
| vs MWR (2017) | 0.017 (0.066) | 0.024 (0.153) | -0.041 (0.212) | -0.356 (0.465) | -0.594 (0.722) |
| vs AERONET (2017) | 0.088 (0.105) | 0.118 (0.192) | 0.017 (0.223) | -0.214 (0.386) | -0.264 (0.306) |

Bias: $PWV_{DSRAD} - PWV_{Other}$