# Peer review of "Development of on-site self-calibration and retrieval methods for sky-radiometer observations of precipitable water vapor"

_Atmospheric Measurement Techniques, 2019_

## Referee Comment (RC1) · Anonymous Referee #1 · 31 Dec 2019

GENERAL COMMENTS

The manuscript by Momoi et al. describes a novel method to self-calibrate the POM sun/sky radiometer for water vapour (WV) retrieval using diffuse sky radiance measurements and to estimate precipitable WV from direct irradiance at 940 nm (using a more "physical" approach than a non-linear empirical parametrisation of the Bouguer-Lambert-Beer law). The method is thoroughly and clearly explained, and the description is supported by sensitivity tests using radiative transfer models. The manuscript is skewed in favour of a theoretical/modelling perspective, with only two ending paragraphs focussed on experimental data, which is however justifiable owing to the main

purpose of presenting a new method rather than studying the retrieved dataset. I recommend the publication of the manuscript after the authors have addressed some minor remarks, mainly aimed at improving readability by the unexperienced reader.

SPECIFIC COMMENTS

1. The new method seems to provide slightly worse results compared to AERONET, although POM and Cimel instruments are similar. It would be good if the authors could elaborate on this, thus enhancing a bit the experimental part of the paper. What is the most likely reason for this result? Is it due to the more physical (less empirical) approach employed in the study, with fewer empirical constraints? Have the authors explored the sensitivity of the retrievals to the accuracy of the instrumental characterisation (e.g., filter response function, field of view, etc.), to the used spectroscopic data (cross sections) or vertical profiles? If so, they could mention some of their results. More generally, on the basis of what criteria can the results of the WV retrieval be considered satisfactory? What are the maximum expected/permissible deviations, using such kind of instrument?

2. At least one plot of the time evolution of the retrieved w should be presented, also in order to understand when the maximum deviations from reference instruments occur;

3. It should be stressed that the sensitivity tests using synthetic data do not include measurement noise. If the authors also made some tests with noise, it would be interesting to present those results in the paper;

4. The present algorithm splits the instrumental characterisation (F0, bandpass, FOV) and the atmospheric parameters (WV profiles), while previous approaches use mixed empirical coefficients (a and b) dependent on both the spectral bandpass and the vertical WV profile. If my understanding is correct, this would permit to use the algorithm in different conditions (place/time) compared to the ones when the instrument was calibrated. Could this be an advantage to be underlined in the conclusions?
TECHNICAL CORRECTIONS

- l. 19, "whose aerosol channels": too abrupt beginning, especially for the readers not experienced in measurements with POM photometers. I would argue that aerosols are seldom the only influencing factor at a specific wavelength, therefore "aerosol channel" sounds more like a colloquial shortcut than a technical term;

- l. 21-22, "by sky-radiometer remain challenge": some articles (a/the) missing;

- l. 22-23, "generally calibrated by the standard Langley method": only for reference instruments, then calibration is transferred to network instruments (as explained later in the text);

- l. 24, "water vapor channel": please mention the channel wavelength here;

- l. 28, "aerosol channels": specify the wavelengths;

- l. 53, "columns of water vapor content": do you mean column concentrations? Unclear, since radiosonde also measure vertical profiles;

- l. 60, "these previous studies": even the studies by Fowle («1992)?

- l. 64: do you really mean "radiometric calibration" (as in F0) or, e.g., "spectral sensitivity"?

- l. 75, "Sky-radiometer": article (the) missing?

- l. 77, "11 wavelengths": maybe a table of the channel wavelengths, together with the main extinction factors, could be useful. See also my first technical comment about the expressions "aerosol channels ... ozone channels" (e.g., even the "ozone channel" is affected by aerosol);

- l. 79, "observation ... self-calibration": a bit confusing, please reformulate to avoid mixing of observation and calibration procedures;

- l. 80, "works in turbid atmospheric conditions": "only" in turbid conditions or "also" in

turbid conditions?

- l. 81-82, "standard ... modified": please, define what a "standard" method and a "modified" one are;

- l. 93, "two SKYNET sites": explain why these two sites were selected. Do they have any particular characteristics, or was this choice oriented by the co-located instrumentation?

- l. 103, "We explain normalized radiance": article missing?

- l. 107-108, "aerosol ... cloud ... water vapor ... ozone channels": cf. previous comments. These approximate expressions could be used only after a short explanation;

- Eq. (1): specify earlier in the text that this holds only for a plane-parallel nonrefractive atmosphere (l. 122, now);

- Eq. (2): if L is defined as sky radiance (l. 106), then it should be already divided by the solid view angle (omega);

- l. 150: the sentence is missing its subject. It is also unclear if this limitation (the real atmosphere not being a single layer) will be addressed in the following text;

- l. 152, "sensitivity of R": ... at a wavelengt of 940 nm;

- l. 154: I guess that these AOD values refer to 940 nm, too?

- l. 155-156, "the aerosol optical thickness does not affect this relationship": unclear, since R decreases, but the values do depend on AOD;

- l. 164: I would not define such a change as "drastic". The variation is only visible in the lower subfigures with a linear y-axis (please, put some letters next to the subplots), and mainly for PWV<=2 cm (as explained later in the text);

- l. 188: is "SKYMAP" an acronym?

- l. 197, "transmittance of the total extinction": isn't just "trasmittance" enough?

- Eq. (8): please, explain where the 1.65 factor comes from;

- l. 218, "local minimum": does this local minimum change at every retrieval, then?

- l. 338: specify that the integral of the filter response function was normalised to 1 (not its maximum);

- Eq. (25)-(26): it could be useful to use a subscript (j?) for the single F0's. Also, please use another letter instead of w (in w_H);

- Eq. (28)-(29): perhaps it would be better to identify the indices with other letters than R (already used for radiances). Also, it should be mentioned that the range of scattering angles for the calculation of index 2 changes during the day;

- l. 390, l. 397 and Table 2, "misjudged"/"incorrect": it is unclear whether the cloud-screening criterium correctly works for the portion of the sky seen by the photometer or not. In the first case, the algorithm does its job, and I think that "misjudged"/"incorrect" are misleading terms, since the conditions of whole sky should not be considered as reference;

- l. 396-397: are <1 and >2 oktas?

- l. 407, "line regression": do you mean linear regression using AOD and wavelength (not logs)?

- Sect. 3: were only "aerosol channels" (l. 107) + "water vapor channel" used in the synthetic retrieval?

- l. 464-465: are the -10% and -3% deviations within the expected uncertainty? If not, can you explain these results? Please, use a proper number of significant digits;

- l. 472: maybe it would be more scientifically correct to plot these values anyway (with another colour/marker) even though they will not be considered;

- l. 478: a more natural choice would be to linearly interpolate the monthly calibrations.

[Figure]

The authors certainly have good reasons for considering the annual mean value, can they elaborate on this?

- l. 518, "much more": "larger"?

- Figs. 1 and 7: the authors should explain the difference between the straight boxes and the rounded ones, and why the latter were not used in Fig. 7;

- Fig. 2: the label "Principal plane" should be put lower, on the principal place circumference;

- Fig. 3: mention somewhere that the plots refer to the principal plane;

- Fig. 10: colours in the second row are hardly distinguishable. Explain that they overlap (in the caption);

- Table 3, "Retrieved the PWV": "PWV retrieval"?

---

## Referee Comment (RC2) · Anonymous Referee #2 · 7 Jan 2020

The study performed by Momoi et al., is a very important evolution in PWV retrieved from radiometers. It exploits findings of established methods and proposes an approach that can be used without prior calibration and could provide reliable PWV from sky-sun measurements. The methodology is described in detail and justified in an appropriate manner. Results are promising and the method could be adopted operationally from Skynet and other networks. I suggest accepting the manuscript for publishing in AMT, after some minor technical corrections and clarifications.

Since it is a novel method, it is crucial to add some preliminary discussion on the uncertainties of the method. It is important for scientists to have an estimation of the

expected uncertainties, before applying the method. Is it more accurate than the well -established methods or the main advantage is the field calibration? Also, when transferring the calibration from the one method to the other, to retrieve PWV from the direct sun data, the error propagation is expected to be very high. I suggest discussing this issue in detail.

It should be clarified in abstract and introduction section, that the calibration constant is referring to the extraterrestrial / Top of the atmosphere value of F. It might be reasonable for people into sunphotometry, but when referring to various equations and approaches, it should be crystal clear the referred quantity.

Method for cloud screening described in 2.2.4, is first time described? If it is not, or if it is based on an existing method, some references should be provided. If it totally new, it should be discussed more. Not just one day as an example for the validity. What are the improvements compared to other approaches? If 17% of cloud cases are contaminating the clear sky data, why don't change to different threshold values? 17% seem a big number which will end in high errors to the data set. At least for the validation of PWV method, stricter criterion is preferable, though it might result in smaller database, because the goal is to estimate the results of algorithm in clear sky conditions.

In step 2 of SKYMAP. Does it retrieves PWV or just the corresponding transmittance (as stated in the abstract?) If it is just the transmittance is should be clarified in the description and change the title of 2.2.2. If it is PWV it is important to plot separately the PWV retrieved with SKYMAP in the comparisons sections. Are these retrievals useful or are just a step in the calculations to obtain the Fo?

Technical comments

Abreviations should be defined also in the abstract.

L56-60 . The two sentences should be separated. It seems that PWV is defined only

at 940nm. 940 is a bandwidth that is selected because of the highest absorption in the shortwave spectral range.

L66 Bruegge approach was also dependable on the altitude of the station, which made it difficult when transporting an instrument.

Figure 10-11. It is not clear what is referred as true values. Please explain in the manuscript.

L124 It seems that something is missing. Which quantity is integrated from BOA to TOA?

L152-153 How R was simulated? An RTM was used? Please describe in detail.

Figure 3. Please use same range in y-axis because it is confusing when it changes all the time.

L167 Some explanation should be provided regarding the selection of dust for the simulation.

L240 There are not 18 boundary layers. As it stated by the term boundary, it is located on the boundaries. Stratosphere is not a boundary layer. I suggest to change to just " 18 layers"/

L440. 0.5cm is too big and not visible in figure 10. Since it is for values <2cm, this is more tha 50% error. Is it a typo or it is estimated from somewhere not shown on the figure?

L476-85. Write in a clear manner that annual values refer to Fo and not PWV.
* * *

---

## Referee Comment (RC3) · Anonymous Referee #3 · 15 Jan 2020

This manuscript written by Momoi et al. presents a new in-situ method to calibrate the so-called water channel in a sky-radiometer Prede POM02, and to derive the precipitable water vapor content in the atmospheric column, by using measurements from the sky-radiometer in the almucantar and principal planes. The method means a further step in the development of methods for the retrieval of aerosols and gases, with very good applicability for the SKYNET international network.

The method description is also complemented with sensitivity analysis, ending with a couple of experimental cases along 1-2 years in two different sites.

The paper is clear, with a good detail about the theoretical considerations. The English

grammar is good, with very few typo errors if any.

I consider the manuscript must be accepted in AMT, after some comments are addressed.

General comments:

- It is very important to include an estimation of the uncertainty of the method, performed by expanding the forward sensitivity analysis or by comparison with other methods, even if preliminary. This information is critical and I miss it in the abstract and more in depth in the text body, if it is currently available.

- The POM02 has been used in this study; however, version POM01 also has a channel in 940nm. Then I understand the method can also be applied to this version. This should be made clear.

- What is the model used for the sensitivity analysis performed in section 2.1.2? RSTAR? State it clearly and give some detail.

- A comparison to other techniques for PWV retrieval developed in SKYNET framework is envisaged (i.e. Campanelli et al), and discussion of pro/cons advised. If this is planned for a future work, please state.

- Errors in aerosol properties interpolated at 940 nm could have an impact on the comparison of PWV between AERONET and this method? Have the authors checked the comparison of aerosol properties from Cimel and this method, during the period studied? If yes, it is not need to include a detailed description, but comment (related to Figure 14).

- Is the cloudscreening also a new method developed for this study? Two different thresholds are used for the two near and far indices. How these thresholds have been selected? Did you performed an statistically study, or it is simply a preliminary proposal?

Specific comments:

- Abstract, line 22: "remains challenging" ?

- Abstract, line 28-30: in the second step, the transmittance of PWV is retrieved; however, both in the text and the plots look like it is the PWV content, not the transmittance, retrieved. Please write consistently.

- page 3, line 73: the calibration method described for AERONET is solely for field instruments, please state.

- page 6, line 150: I'm not sure to understand the need for this sentence here, or it looks like incomplete discussion here.

- Page 11, line 326: In equation 21, the band average transmittance is used, converted from PWV in step 2. Do you mean the conversion is performed in the step 3, from the PWV obtained in step 1?

- Page 12, line 355. Is the Huber's M-estimation method iterative? (the weight to calculate ln<F0> depends on ln<F0>, so any condition applied?)

- Page 14, line 415: what is the time stamp used for any triplet? The central element time stamp or the first element time stamp?

- Page 14, line 440: this error estimation about 0.5cm looks somewhat high to me, in comparison to other techniques. It would be useful to provide literature estimations of the uncertainty for the methods used as referents (AERONET, MWR, GPS...) in the comparison discussion.

- Page 17, summary (conclusions): is there any error estimation for PWV>2cm? Some discussion on the pros and cons of this method and other methods developed for Prede POM instruments would be useful.

- Is figure 1 used in the text?

- Caption figure 3: please explain what S-S refers to (I think it is single scattering) .

- Figures: I think saying "top row" and "bottom row" is more appropiate than "top and bottom line".

---

## Author Comment (AC3) · 25 Mar 2020

**Response to comments of referee 3**

Authors would like to express sincere thanks to the referee 3 for valuable comments. We revised a manuscript carefully based on given comments. The comments of the referee 3 are in blue, our replies are in black, and changes made in the revised manuscript are in red. The English in this document has been checked by at least two professional editors, both native speakers of English. Our replies to the comments are as below.

This manuscript written by Momoi et al. presents a new in-situ method to calibrate the so-called water channel in a sky-radiometer Prede POM02, and to derive the precipitable water vapor content in the atmospheric column, by using measurements from the sky-radiometer in the almucantar and principal planes. The method means a further step in the development of methods for the retrieval of aerosols and gases, with very good applicability for the SKYNET international network.

The method description is also complemented with sensitivity analysis, ending with a couple of experimental cases along 1-2 years in two different sites.

The paper is clear, with a good detail about the theoretical considerations. The English grammar is good, with very few typo errors if any. I consider the manuscript must be accepted in AMT, after some comments are addressed.

General comments:

- It is very important to include an estimation of the uncertainty of the method, performed by expanding the forward sensitivity analysis or by comparison with other methods, even if preliminary. This information is critical and I miss it in the abstract and more in depth in the text body, if it is currently available.

- Errors in aerosol properties interpolated at 940 nm could have an impact on the comparison of PWV between AERONET and this method? Have the authors checked the comparison of aerosol properties from Cimel and this method, during the period studied? If yes, it is not need to include a detailed description, but comment (related to Figure 14).

We discussed the uncertainty of PWV by conducted the sensitivity tests using the simulated data with the bias errors in diffuse radiances in Section 3 and compared with the AERONET sun-sky radiometer in Section 4.2. We also estimate error propagation from aerosol optical thickness to PWV in Appendix B in the revised manuscript as below:

*(Section 3: L510-521)*

We also conducted sensitivity tests using the simulated data with bias errors to investigate uncertainty in the SKYMAP-derived PWV. The bias errors were ± 5% and ± 10% for $R$. The value of 5% was given by following reasons. The SVA bias errors of the diffuse radiances for the sky-radiometer observations were estimated to be less than 5% (Uchiyama *et al*., 2018b). According to Dubovik *et al*. (2000), the uncertainty of the diffuse radiances for the AERONET measurements is ± 5%. Figures 13 and 14 show the results from the simulated data for the continental average and transported dust aerosols with aerosol optical thicknesses of 0.02, 0.06 and 0.20 at 940 nm. PWV was overestimated when − 5% bias was applied to $R$. This corresponds to the relationship between $R$ and PWV, where $R$ decreases with increasing PWV (Section 2.1.2). The bias errors strongly affected the retrieval of PWV at high PWV (> 2 cm), because the sensitivity of high PWV is lower than that of low PWV. The retrieval error of PWV increased with increasing bias errors. The retrieval error of PWV due to ± 5% and ± 10% errors for $R$ was within 10% for PWV < 2 cm and up to 200% for PWV > 2 cm.

[Figure]

**Figure 13: Comparison of the "true" and retrieval values of PWV from simulated data for continental average aerosol with bias errors. The top, middle, and bottom rows are the retrieval results at SZA = 30°, 50°, and 70°, respectively. Closed circles are the results with no bias errors. Closed squares and closed triangles are the results with bias errors of plus and minus 5% in $R$, respectively. Open squares and open triangles are the results with bias errors of plus and minus**

**10% in $R$, respectively.**

[Figure]

**Figure 14: Similar to Fig. 13 but for transported dust aerosol.**

*(Section 4.2: L607-625)*

PWV$_{\text{DSRAD+SKYMAP}}$ using the annual mean calibration constant was 12% and 9.1% smaller than PWV$_{\text{MWR}}$ and PWV$_{\text{Cimel}}$, respectively (Table 5). These results suggest an underestimation of PWV$_{\text{DSRAD+SKYMAP}}$, as the uncertainty of PWV$_{\text{Cimel}}$ compared to the GNSS/GPS receiver is expected to

be less than 10% (Giles *et al.*, 2018). The underestimation of PWV$_{DSRAD+SKYMAP}$ was due to two factors. The first is the retrieval of PWV by the annual mean calibration constant for the water vapor channel. The calibration constant not only is subject to aging but also undergoes seasonal variation due to temperature dependency (Uchiyama *et al.*, 2018a). Thus, it is possible to underestimate the calibration constant in the wet season. Second, uncertainty regarding the aerosol optical thickness affected PWV retrieval. Figure 18 depicts the differences in PWV and aerosol optical thicknesses at 675, 870, and 1020 nm between the DSRAD algorithm and the AERONET retrieval. In the periods from January to May and from October to November, the differences in PWV and aerosol optical thicknesses were less than 0.1 cm and 0.015, respectively. However, the difference in PWV was greater than 0.1 cm from July to September. This corresponds to the difference in aerosol optical thicknesses at 675, 870, and 1020 nm from July to September, which indicates that the transmittance of water vapor was overestimated by the overestimation of aerosol optical thickness. This led to the underestimation of PWV$_{DSRAD+SKYMAP}$ using the annual mean calibration constant when PWV was > 3 cm. In our error estimation, the error of + 0.03 for the aerosol optical thickness at 940 nm resulted in the error of − 0.214 cm for PWV (Appendix B).

[Figure]

**Figure 18: The top row shows the time series of PWV in 2017 at Chiba (green and black circles are PWV$_{DSRAD+SKYMAP}$ and PWV$_{Cimel}$, respectively). The middle row is the difference between**

**PWV**DSRAD+SKYMAP **and PWV**Cimel**. The bottom row is the difference in aerosol optical thicknesses at 675 nm (red), 870 nm (blue), and 1020 nm (green) between the DSRAD algorithm and the AERONET retrieval results. Circles and error bars in the middle and bottom rows are means and standard deviations, respectively.**

**Table 5: Comparison of PWV between DSRAD and other instruments.**

| | | Slope $C_1$ | Intercept $C_2$[cm] | $\gamma$ | Bias [cm] | RMSE [cm] |
|---|---|---|---|---|---|---|
| PS1202091 at Tsukuba, Japan | | | | | | |
| Monthly mean $F_0$ | vs GNSS/GPS receiver (2013) | 0.956 | 0.079 | 0.938 | -0.049 | 0.138 |
| | vs GNSS/GPS receiver (2014) | 0.937 | 0.161 | 0.970 | -0.110 | 0.170 |
| Annual mean $F_0$ | vs GNSS/GPS receiver (2013) | 0.919 | 0.173 | 0.987 | -0.061 | 0.226 |
| | vs GNSS/GPS receiver (2014) | 0.934 | 0.178 | 0.987 | -0.089 | 0.223 |
| PS2501417 at Chiba, Japan | | | | | | |
| Monthly mean $F_0$ | vs MWR (2017) | 0.964 | 0.053 | 0.961 | -0.027 | 0.091 |
| | vs AERONET (2017) | 0.987 | 0.107 | 0.976 | 0.098 | 0.122 |
| Annual mean $F_0$ | vs MWR (2017) | 0.880 | 0.132 | 0.985 | 0.042 | 0.231 |
| | vs AERONET (2017) | 0.909 | 0.184 | 0.991 | 0.055 | 0.186 |

$C_1, C_2$: $\text{PWV}_{\text{DSRAD}} = C_1 \times \text{PWV}_{\text{Other}} + C_2$
Bias: $\text{PWV}_{\text{DSRAD}} - \text{PWV}_{\text{Other}}$

*(Appendix B)*

**Appendix B: Error propagation from aerosol optical thickness to PWV**

We evaluated the influence of the uncertainty of aerosol optical thickness on PWV using the empirical equation of Bruegge *et al*. (1992). PWV is described using the adjustment parameters as follows

$$w = \frac{1}{m_0}\left(-\frac{\ln \bar{T}_{\text{H2O}}}{a}\right)^{\frac{1}{b}} \text{[cm]}. \qquad \text{(B1)}$$

The uncertainty of PWV $\epsilon_{\text{PWV}}$ is given from the partial differentiation of Eq. (B1) with respect to $\ln \bar{T}_{\text{H2O}}$ as follows

$$\epsilon_{\text{PWV}} = \frac{\partial w}{\partial \ln \bar{T}_{\text{H2O}}}\epsilon_{\ln \bar{T}_{\text{H2O}}} = \frac{w}{b \ln \bar{T}_{\text{H2O}}}\epsilon_{\ln \bar{T}_{\text{H2O}}}. \qquad \text{(B2)}$$

where $\epsilon_{\ln \bar{T}_{\text{H2O}}}$ is the uncertainty of $\bar{T}_{\text{H2O}}$. Using Eq. (B1) with the adjusting parameters of the sky-radiometer, with $a = 0.620$ and $b = 0.625$ as the coefficient values for the trapezoidal spectral response function (Uchiyama *et al*., 2018a), we write the uncertainty of PWV as

$$\epsilon_{\text{PWV}} = -\frac{w}{ab}(m_0 w)^{-b}\epsilon_{\ln \bar{T}_{\text{H2O}}} = -\frac{w}{0.388}(m_0 w)^{-0.625}\epsilon_{\ln \bar{T}_{\text{H2O}}}. \qquad \text{(B3)}$$

If the uncertainty of the calibration constant at the water vapor channel is ignored, the uncertainty of $\bar{T}_{H2O}$ is given from Eq. (21) as follows

$$\epsilon_{\ln \bar{\tau}_{H2O}} = m_0 \epsilon_{AOT}. \qquad (B4)$$

where $\epsilon_{AOT}$ is the uncertainty of the aerosol optical thickness at 940 nm. The uncertainty of PWV is written by Eqs. (B3) and (B4) as

$$\epsilon_{PWV} = -\frac{1}{0.388}(m_0 w)^{0.375} \epsilon_{AOT} = -0.214 \text{ [cm]}. \qquad (B5)$$

where $m_0 = 3.0$, $w = 5.0$ cm, and $\epsilon_{AOT} = 0.03$.

- The POM02 has been used in this study; however, version POM01 also has a channel in 940nm. Then I understand the method can also be applied to this version. This should be made clear.

We agree with the reviewer. Our methods (SKYMAP and DSRAD) can be used for sky-radiometer model POM-01. Thus, Section 1 and 2 of the revised manuscript was generally written.

- What is the model used for the sensitivity analysis performed in section 2.1.2? RSTAR? State it clearly and give some detail.

Yes, $R$ is simulated by RTM (RSTAR). We revised it (L193-195) as below:
We examined the sensitivity of $R$ at 940 nm in the two observation planes to PWV, aerosol optical properties, and aerosol vertical profiles by simulating $R$ using the radiative transfer model RSTAR (Nakajima and Tanaka, 1986, 1988).

- A comparison to other techniques for PWV retrieval developed in SKYNET framework is envisaged (i.e. Campanelli et al), and discussion of pro/cons advised. If this is planned for a future work, please state.

We agree with the reviewer. We revised it (L656-658) as below:
In future work, we plan to compare our method with others in the SKYNET framework (Uchiyama *et al*. 2014; Campanelli *et al*., 2014).

- Is the cloud screening also a new method developed for this study? Two different thresholds are used for the two near and far indices. How these thresholds have been selected? Did you performed an statistically study, or it is simply a preliminary proposal?

Yes, the cloud screening method, called SCAD method, is developed for this study. We determined the thresholds by comparing the images of the whole-sky camera and the time series of the surface solar radiation observed by the pyranometer and validated by visual observation. We added the validation results of the SCAD method in the revised manuscript (L455-464) as below:

The results were validated using visual observation of the amount of clouds in the Aerological Observatory of the JMA. Figure 10a shows the histograms of index 1 for cases in which the sun was and was not covered by clouds. Index 1 had a low value when there were no clouds shading the sun but had a wide range of values when clouds were shading the sun. Fig. 10b shows the histograms of index 2 when cloud cover was and was not < 20%. The peak shifted to the right when cloud cover was ≥ 20%, but the effect was not significant. Table 3 shows the validation results of this method. We defined "best condition" as cloud cover < 20% and "poor condition" as cloud cover ≥ 20%. In less than 17% of cases a "poor condition" was judged as a "best condition". The sky-radiometer observes only a part of the whole sky, but our algorithm showed good results.

[Figure]

**Figure 10: Histograms of indexes 1 and 2 of sky-radiometer observations at Tsukuba. (a) Index 1 when the sun is covered by clouds (blue boxes) and not covered by clouds (red boxes). (b) Index 2 when cloud cover is less than to 20% (red boxes) and greater than or equal to 20% (blue boxes).**

**Table 3: Validation of the SCAD method by visual observation from 2013 to 2014 in Tsukuba.**

| Visual observation | Sky-radiometer measuring plane | |
| Cloud cover | Available condition | Poor quality condition |
| --- | --- | --- |
| Clear, less than 20% | **463 (83.4%)\*** | 68 (8.7%) |
| Cloud affected, more than 20% | 92 (16.6%) | **714(91.3%)\*** |

\*Obviously correct determination.

Technical comments

- Abstract, line 22: "remains challenging" ?

Yes, it is. We revised it (L28) as below:

…the sky-radiometer remains challenging…

- Abstract, line 28-30: in the second step, the transmittance of PWV is retrieved; however, both in the text and the plots look like it is the PWV content, not the transmittance, retrieved. Please write consistently.

The step 2 of the SKYMAP retrieves PWV to estimate the transmittance of the PWV, but not directly retrieves the PWV. We revised it (L39).

- page 3, line 73: the calibration method described for AERONET is solely for field instruments, please state.

We agree with the reviewer. We revised it (L97-100) as below:

In the AERONET led by NASA, the field instrument of the AERONET sun-sky radiometer is calibrated every year by lamp calibration and side-by-side comparison with a reference spectroradiometer (Holben *et al.*, 1998).

- page 6, line 150: I'm not sure to understand the need for this sentence here, or it looks like incomplete discussion here.

We agree with the reviewer. It is deleted and mentioned in Sect. 2.1.2 in the revised manuscript.

- Page 11, line 326: In equation 21, the band average transmittance is used, converted from PWV in step 2. Do you mean the conversion is performed in the step 3, from the PWV obtained in step 1?

The band average transmittance in Eq. 21 is converted from the PWV retrieved in step 2 not step 1. Since the diffuse sky irradiances are calculated for each sub-band by the radiative transfer model, in step 2 you can get the PWV instead of the band average transmittance of the PWV and need to convert it to the band average transmittance to get the F0.

- Page 12, line 355. Is the Huber's M-estimation method iterative? (the weight to calculate ln<F0> depends on ln<F0>, so any condition applied?)

Yes, Huber's M-estimation method is iterative. This method has the potential to apply noisy satisfying normal distribution, not white noise.

- Page 14, line 415: what is the time stamp used for any triplet? The central element time stamp or the first element time stamp?

The time stamp used the central element time stamp. We revised it (L485-486) as below:

Second, the triplet of the aerosol optical thickness in Smirnov *et al.* (2000) is built from the pre/post 1 min data instead of 30 s.

- Page 14, line 440: this error estimation about 0.5cm looks somewhat high to me, in comparison to other techniques. It would be useful to provide literature estimations of the uncertainty for the methods used as referents (AERONET, MWR, GPS...) in the comparison discussion.

We estimate the uncertainty from the sensitivity tests in Section 3. "0.5 cm" is a typo. We compared our retrievals with more establish methods, such as GNSS/GPS-derived PWV, and AERONET-derived PWV as below:

*(Section 4.1: L582-587)*

Table 5 summarizes the results of comparisons of DSRAD-derived PWV and GNSS/GPS-derived PWV. The magnitude of the bias error and root mean square error were small, less than 0.11 cm and less than 0.226 cm, during 2013 to 2014. Table 6 shows the errors of the retrieved PWV with the annual mean calibration constants for the rank of PWV. The bias error was larger for high PWV than it was for low PWV. The magnitude of the bias errors of PWV was less than 0.163 cm for PWV < 3 cm and less than 0.339 cm for PWV > 3 cm.

*(Section 4.2: L602-610)*

$PWV_{DSRAD+SKYMAP}$ using the annual mean calibration constant agreed with $PWV_{MWR}$ (Fig. 17c). The error of $PWV_{DSRAD+SKYMAP}$ was $-0.041 < bias < 0.024$ cm and RMSE $< 0.212$ cm for low PWV (<3

235 cm) and bias $< -0.356$ cm and RMSE $> 0.465$ cm for high PWV (Table 6). Figure 17d shows that $PWV_{DSRAD+SKYMAP}$ using the annual mean calibration constant also agreed with $PWV_{Cimel}$ for low PWV ($< 3$ cm) but was smaller than $PWV_{Cimel}$ for high PWV ($> 3$ cm). $PWV_{MWR}$ was larger than $PWV_{Cimel}$ (Fig. 17e). $PWV_{DSRAD+SKYMAP}$ using the annual mean calibration constant was 12% and 9.1% smaller than $PWV_{MWR}$ and $PWV_{Cimel}$, respectively (Table 5).

240

**Table 5: Comparison of PWV between DSRAD and other instruments.**

|  |  | Slope $C_1$ | Intercept $C_2$[cm] | $\gamma$ | Bias [cm] | RMSE [cm] |
|---|---|---|---|---|---|---|
| PS1202091 at Tsukuba, Japan |  |  |  |  |  |  |
| Monthly mean $F_0$ | vs GNSS/GPS receiver (2013) | 0.956 | 0.079 | 0.938 | -0.049 | 0.138 |
|  | vs GNSS/GPS receiver (2014) | 0.937 | 0.161 | 0.970 | -0.110 | 0.170 |
| Annual mean $F_0$ | vs GNSS/GPS receiver (2013) | 0.919 | 0.173 | 0.987 | -0.061 | 0.226 |
|  | vs GNSS/GPS receiver (2014) | 0.934 | 0.178 | 0.987 | -0.089 | 0.223 |
| PS2501417 at Chiba, Japan |  |  |  |  |  |  |
| Monthly mean $F_0$ | vs MWR (2017) | 0.964 | 0.053 | 0.961 | -0.027 | 0.091 |
|  | vs AERONET (2017) | 0.987 | 0.107 | 0.976 | 0.098 | 0.122 |
| Annual mean $F_0$ | vs MWR (2017) | 0.880 | 0.132 | 0.985 | 0.042 | 0.231 |
|  | vs AERONET (2017) | 0.909 | 0.184 | 0.991 | 0.055 | 0.186 |

$C_1, C_2$: $PWV_{DSRAD} = C_1 \times PWV_{Other} + C_2$
Bias: $PWV_{DSRAD} - PWV_{Other}$

245 **Table 6: Difference in PWV between DSRAD with the annual mean calibration constants and other instruments.**

|  | PWV$_{Other}$ | | | | |
|---|---|---|---|---|---|
|  | $0 - 1$ cm | $1 - 2$ cm | $2 - 3$ cm | $3 - 4$ cm | $> 4$ cm |
|  | Bias [cm] (RMSE [cm]) | Bias [cm] (RMSE [cm]) | Bias [cm] (RMSE [cm]) | Bias [cm] (RMSE [cm]) | Bias [cm] (RMSE [cm]) |
| PS1202091 at Tsukuba, Japan |  |  |  |  |  |
| vs GNSS/GPS receiver (2013) | 0.083 (0. 124) | 0.160 (0.211) | 0.084 (0.236) | -0.098 (0.326) | -0.339 (0.537) |
| vs GNSS/GPS receiver (2014) | 0.110 (0.142) | 0.163 (0.221) | 0.107 (0.251) | -0.055 (0.353) | -0.239 (0.492) |
| PS2501417 at Chiba, Japan |  |  |  |  |  |
| vs MWR (2017) | 0.017 (0.066) | 0.024 (0.153) | -0.041 (0.212) | -0.356 (0.465) | -0.594 (0.722) |
| vs AERONET (2017) | 0.088 (0.105) | 0.118 (0.192) | 0.017 (0.223) | -0.214 (0.386) | -0.264 (0.306) |

Bias: $PWV_{DSRAD} - PWV_{Other}$

- Page 17, summary (conclusions): is there any error estimation for PWV>2cm? Some discussion on the
250 pros and cons of this method and other methods developed for Prede POM instruments would be useful.

We added the results of the uncertainty for PWV >2 cm in the revised manuscript (L650-655) as below: The magnitude of the bias error and the root mean square error were $< 0.163$ cm and $< 0.251$ cm, respectively, for low PWV ($< 3$ cm). However, our retrieved PWV was underestimated in the wet

conditions, and the magnitude of the bias error and the root mean square error were less than 0.594 cm and less than 0.722 cm for high PWV. This was due to seasonal variation in the calibration constant and the overestimation of aerosol optical thickness at 940 nm interpolated from those at 870 and 1020 nm.

- Is figure 1 used in the text?

It is using the top of Section 2 and Section 3 in the discussion paper.

- Caption figure 3: please explain what S-S refers to (I think it is single scattering) .

Yes, "S-S approx." means the single scattering approximation. We added the sentence in the caption as below:

S-S Approx. is single scattering approximation.

- Figures: I think saying "top row" and "bottom row" is more appropiate than "top and bottom line".

We agree with the reviewer. We revised it.

[revised manuscript text omitted]
(x) = \frac{1}{2}\big(y^{\text{meas}} - y(x)\big)^T (W^2)^{-1}\big(y^{\text{meas}} - y(x)\big) + \frac{1}{2}\big(y_a(x)\big)^T (W_a^2)^{-1}\big(y_a(x)\big), \quad (13)$$

where vector $y^{\text{meas}}$ describes the measurements (normalized radiances $R^{\text{meas}}$ and transmittances of total extinction $T^{\text{meas}}$ ) at the aerosol channels, vector $x$ describes the aforementioned aerosol parameters —$n(\lambda)$, $k(\lambda)$, $C_i$, and $\delta$— to be estimated, vector $y(x)$ comprises the values corresponding to $y^{\text{meas}}$ calculated from $x$ by the forward model ($R^{\text{ret}}$ and $T^{\text{ret}}$), and matrix $W^2$ is the covariance matrix of $y$ and is assumed to be diagonal. The diagonal elements of $W$ are  standard errors in the measurements. We set their values at 0.02 for $T^{\text{meas}}$, and 10% for $R^{\text{meas}}$.

To reduce the effects of observational error on retrieval and to conduct stable analyses, Dubovik and King (2000) considered restricting the spectral variability of the volume size distribution and limiting the length of  the refractive index derivative with respect to the wavelength . They considered this *a priori* smoothness constraint as being of the same nature as a measurement and incorporated the smoothness constraint into their retrieval scheme. We also consider the smoothness constraints in this study. The second term of Eq. (13) consists of *a priori* information on the wavelength dependencies of the refractive index, aerosol optical thickness, and smoothness of the volume spectrum, which is described as

$$y_a(x) = \big(y_a^{\text{Re}}, y_a^{\text{Im}}, y_a^{\text{Sca}}, y_a^{\text{Abs}}, y_a^{\text{Vol}}\big)^T, \quad (14)$$

where vectors $y_a^{\text{Re}}$, $y_a^{\text{Im}}$, $y_a^{\text{Sca}}$, $y_a^{\text{Abs}}$, and $y_a^{\text{Vol}}$ are *a priori* information on the wavelength dependencies of the refractive index (real and imaginary parts), aerosol optical thickness (scattering and absorption parts), and smoothness of the volume spectrum, respectively. The matrix $W_a^2$ in Eq. (13) is the covariance matrix for determining the strengths of the constraints.

We adapt the smoothness constraints of the second derivatives for the real and imaginary parts of the refractive index. The second derivatives are defined as

$$y_a^{\text{Re}(i)}(x) = \left(\frac{\ln n(\lambda_i) - \ln n(\lambda_{i+1})}{\ln \lambda_i - \ln \lambda_{i+1}} - \frac{\ln n(\lambda_{i+1}) - \ln n(\lambda_{i+2})}{\ln \lambda_{i+1} - \ln \lambda_{i+2}}\right), \quad (15)$$

$$y_a^{\text{Im}(i)}(x) = \left(\frac{\ln k(\lambda_i) - \ln k(\lambda_{i+1})}{\ln {}_i - \ln \lambda_{i+1}} - \frac{\ln k(\lambda_{i+1}) - \ln k(\lambda_{i+2})}{\ln \lambda_{i+1} - \ln \lambda_{i+2}}\right), \quad (16)$$

$$(i = 1, \cdots, N_w - 2),$$

where $y_a^{\text{Re}(i)}$ and $y_a^{\text{Im}(i)}$ are the $i$-th elements of the vectors $y_a^{\text{Re}}$ and $y_a^{\text{Im}}$, respectively. $N_w$ is the number of wavelengths. The values entered into the weight matrix $W_a$ are 0.2 for the real part and 1.25 for the

imaginary part. These values are adopted from Dubovik and King (2000). Furthermore, we introduce the smoothness constraints to the spectral distributions of the scattering and absorption parts of the aerosol optical thickness by

$$y_a^{\text{Sca}(i)}(\boldsymbol{x}) = \left( \frac{\ln \tau_{sca}(\lambda_i) - \ln \tau_{sca}(\lambda_{i+1})}{\ln \lambda_i - \ln \lambda_{i+1}} - \frac{\ln \tau_{sca}(\lambda_{i+1}) - \ln_{sca}(\lambda_{i+2})}{\ln \lambda_{i+1} - \ln_{i+2}} \right), \qquad (17)$$

$$y_a^{\text{Abs}(i)}(\boldsymbol{x}) = \left( \frac{\ln \tau_{abs}(\lambda_i) - \ln \tau_{abs}(\lambda_{i+1})}{\ln \lambda_i - \ln \lambda_{i+1}} - \frac{\ln \tau_{abs}(\lambda_{i+1}) - \ln \tau_{abs}(\lambda_{i+2})}{\ln \lambda_{i+1} - \ln \lambda_{i+2}} \right), \qquad (18)$$

$$(i = 1, \cdots, N_w - 2),$$

where $y_a^{\text{Sca}(i)}$ and $y_a^{\text{Abs}(i)}$ are the $i$-th elements of the vectors $\boldsymbol{y}_a^{\text{Sca}}$ and $\boldsymbol{y}_a^{\text{Abs}}$, respectively. The value entered in the weight matrix $\boldsymbol{W}_a$ is 2.5 for both the scattering and absorption parts of the aerosol optical thickness. To stabilize the estimation of the volume size distribution, we introduce the smoothness constraint for the adjacent volume size spectrum $C_i$, as:

$$y_a^{\text{Vol}(i)}(\boldsymbol{x}) = (\ln C_{i-1} - \ln C_i) - (\ln C_i - \ln C_{i+1}), \quad (19)$$

$$(i = 1, \cdots, 20),$$

$$C_0 = 0.01 \times \min\{C_i | i = 1, \cdots, 20\}, \quad C_{21} = 0.01 \times \min\{C_i | r_i > r_b, i = 1, \cdots, 20\}.$$

where $y_a^{\text{Vol}(i)}$ is the $i$-th element of the vector $\boldsymbol{y}_a^{\text{Vol}}$. The small values of $C_0$ and $C_{21}$ at $r_0$ and $r_{21}$ are given to prevent both ends of the size distribution ($C_1$ and $C_{20}$) from being abnormal values because *F*  and *V*  do not have sufficient information to estimate the size distribution of both small ($r < 0.1$ μm) and large particles ($r > 7$ μm; Dubovik *et al.*, 2000). Note that $r_0$ and $r_{21}$ satisfy Eq. (7). The value entered in the weight matrix $\boldsymbol{W}_a$ is 1.6 for the smoothness constraint of the size distribution.

We minimize $f(\boldsymbol{x})$ of Eq. (13) using the algorithm developed in Kudo *et al.* (2016), which is based on the Gauss-Newton method and the logarithmic transformations of $\boldsymbol{x}$ and $\boldsymbol{y}$. Finally, the aerosol optical properties from aerosol channels are obtained from $\boldsymbol{x}$ using Eqs. (11) and (12).

**2.2.2 Step 2: Retrieval of PWV**

We estimate PWV by the following procedure. The aerosol volume size distribution is obtained from step 1, and the refractive index at 940 nm is calculated from those at 870 and 1020 nm by linear interpolation in the log-log plane. Using the size distribution and the interpolated refractive index, we

can compute the aerosol optical properties and the normalized angular distribution at the water vapor channel using the forward model described in Section 2.2.1. We retrieve PWV by minimizing the following cost function:

$$f(\mathbf{x}) = \frac{1}{2}\left(\mathbf{y}^{\text{meas}} - \mathbf{y}(\mathbf{x})\right)^T (\mathbf{W}^2)^{-1}\left(\mathbf{y}^{\text{meas}} - \mathbf{y}(\mathbf{
[revised manuscript text omitted]

---

## Author Response (AR1)

**Response to comments of referee 1**

Authors would like to express sincere thanks to the referee 1 for valuable comments. We revised a manuscript carefully based on given comments. The comments of the referee 1 are in blue, our replies are in black, and changes made in the revised manuscript are in red. The English in this document has been checked by at least two professional editors, both native speakers of English. Our replies to the comments are as below.

**General comments**

The manuscript by Momoi et al. describes a novel method to self-calibrate the POM sun/sky radiometer for water vapour (WV) retrieval using diffuse sky radiance measurements and to estimate precipitable WV from direct irradiance at 940 nm (using a more "physical" approach than a non-linear empirical parametrisation of the Bouguer-Lambert-Beer law). The method is thoroughly and clearly explained, and the description is supported by sensitivity tests using radiative transfer models. The manuscript is skewed in favour of a theoretical/modelling perspective, with only two ending paragraphs focussed on experimental data, which is however justifiable owing to the main purpose of presenting a new method rather than studying the retrieved dataset. I recommend the publication of the manuscript after the authors have addressed some minor remarks, mainly aimed at improving readability by the unexperienced reader.

**Specific comments**

1. The new method seems to provide slightly worse results compared to AERONET, although POM and Cimel instruments are similar. It would be good if the authors could elaborate on this, thus enhancing a bit the experimental part of the paper. What is the most likely reason for this result? Is it due to the more physical (less empirical) approach employed in the study, with fewer empirical constraints? Have the authors explored the sensitivity of the retrievals to the accuracy of the instrumental characterisation (e.g., filter response function, field of view, etc.), to the used spectroscopic data (cross sections) or vertical profiles? If so, they could mention some of their results. More generally, on the basis of what criteria can the results of the WV retrieval be considered satisfactory? What are the maximum expected/permissible deviations, using such kind of instrument?

2. At least one plot of the time evolution of the retrieved w should be presented, also in order to understand when the maximum deviations from reference instruments occur;

The DSRAD algorithm retrieved the precipitable water vapor by a physical approach although the previous study of the sky-radiometer used an empirical approach (Uchiyma et al., 2014; Campanelli et al., 2014, 2018). The AERONET also uses the empirical method (Holben *et al*., 1998). Our retrieved value was slightly worse compared with AERONET retrievals and others. The underestimation of

40   $PWV_{DSRAD+SKYMAP}$ was due to two factors. The first is the retrieval of PWV by the annual mean calibration constant for the water vapor channel. The calibration constant not only is subject to aging but also undergoes seasonal variation due to temperature dependency (Uchiyama *et al*., 2018a). Thus, it is possible to underestimate the calibration constant in the wet season. Second, uncertainty regarding the aerosol optical thickness affected PWV retrieval. Figure 18 depicts the differences in PWV and aerosol

45   optical thicknesses at 675, 870, and 1020 nm between the DSRAD algorithm and the AERONET retrieval. In the periods from January to May and from October to November, the differences in PWV and aerosol optical thicknesses were less than 0.1 cm and 0.015, respectively. However, the difference in PWV was greater than 0.1 cm from July to September. This corresponds to the difference in aerosol optical thicknesses at 675, 870, and 1020 nm from July to September, which indicates that the

50   transmittance of water vapor was overestimated by the overestimation of aerosol optical thickness. This led to the underestimation of $PWV_{DSRAD+SKYMAP}$ using the annual mean calibration constant when PWV was > 3 cm. The above description is added in the revised manuscript (L612-625) and we also added the time series of PWV in the revised manuscript (Figure 18).

[Figure]

**Figure 18: The top row shows the time series of PWV in 2017 at Chiba (green and black circles are PWV_{DSRAD+SKYMAP} and PWV_{Cimel}, respectively). The middle row is the difference between PWV_{DSRAD+SKYMAP} and PWV_{Cimel}. The bottom row is the difference in aerosol optical thicknesses at 675 nm (red), 870 nm (blue), and 1020 nm (green) between the DSRAD algorithm and the AERONET retrieval results. Circles and error bars in the middle and bottom rows are means and standard deviations, respectively.**

3. It should be stressed that the sensitivity tests using synthetic data do not include measurement noise. If the authors also made some tests with noise, it would be interesting to present those results in the paper;

We conducted the sensitivity tests using the simulated data with the bias errors in the diffuse radiances. It is added in the revised manuscript (L510-521) as below:

[revised manuscript text omitted]

Bias: $PWV_{DSRAD} - PWV_{Other}$

4. The present algorithm splits the instrumental characterisation (F0, bandpass, FOV) and the atmospheric parameters (WV profiles), while previous approaches use mixed empirical coefficients (a and b) dependent on both the spectral bandpass and the vertical WV profile. If my understanding is correct, this would permit to use the algorithm in different conditions (place/time) compared to the ones when the instrument was calibrated. Could this be an advantage to be underlined in the conclusions?

Yes, exactly. We add the sentence in the conclusion (L631-636) as below:

Our DSRAD algorithm retrieves PWV from the direct solar irradiance. This method does not require adjustment parameters used in the empirical methods of previous studies (e.g., Holben *et al.*, 1998; Uchiyama *et al.*, 2014; Campanelli *et al*., 2014, 2018). Instead, the filter response function and the vertical profiles of water vapor, temperature, and pressure are required as input parameters. Thus, our physics-based algorithm has the potential to be applied to sky-radiometers all over the world. This is the greatest advantage of the present study.

Technical corrections

- l. 19, "whose aerosol channels": too abrupt beginning, especially for the readers not experienced in measurements with POM photometers. I would argue that aerosols are seldom the only influencing factor at a specific wavelength, therefore "aerosol channel" sounds more like a colloquial shortcut than a technical term;
- l. 28, "aerosol channels": specify the wavelengths;

We agree with the reviewer. We revised it (L19-26) as below:

The Prede sky-radiometer measures direct solar irradiance and the angular distribution of diffuse radiances at the ultraviolet, visible, and near-infrared wavelengths. These data are utilized for remote sensing of aerosols, water vapor, ozone, and clouds, but the calibration constant which is the sensor output current of the extraterrestrial solar irradiance at the mean distance between the Earth and the sun, is needed. The aerosol channels, which are the weak gas absorption wavelengths of 340, 380, 400, 500, 675, 870, and 1020 nm, can be calibrated by an on-site self-calibration method, the Improved Langley method.

- l. 21-22, "by sky-radiometer remain challenge": some articles (a/the) missing;

We agree with the reviewer. We revised it (L28) as below:

by the sky-radiometer remains challenging

- l. 22-23, "generally calibrated by the standard Langley method": only for reference instruments, then calibration is transferred to network instruments (as explained later in the text);

- l. 24, "water vapor channel": please mention the channel wavelength here;

We agree with the reviewer. We revised it (L27-31) as below:

However, the continuous long-term observation of precipitable water vapor (PWV) by the sky-radiometer remains challenging, because calibrating the water vapor absorption channel of 940 nm generally relies on the standard Langley method (SL) at limited observation sites (*e.g.*, the Mauna Loa Observatory) and the transfer of the calibration constant by side-by-side comparison with the reference sky-radiometer calibrated by the SL method.

- l. 53, "columns of water vapor content": do you mean column concentrations? Unclear, since radiosonde also measure vertical profiles;

It means "precipitable water vapor". We revised it (L76).

- l. 60, "these previous studies": even the studies by Fowle («1992)?

We intended "these previous studies" points "SKYNET sky-radiometer (Campanelli *et al.*, 2014, 2018; Uchiyama *et al.*, 2014, 2018a), and AERONET sun-sky photometer (Holben *et al.*, 1998)". Therefore, we revised it (L84-87) as below:

Previous studies of SKYNET and AERONET derived PWV from the observed transmittance of water vapor ($\overline{T}_{H2O}$), assuming $\overline{T}_{H2O} = e^{-a(m \cdot w)^b}$ (Bruegge *et al.*, 1992), where *a* and *b* are adjustment parameters, *m* is the optical air mass, and *w* is PWV.

- l. 64: do you really mean "radiometric calibration" (as in F0) or, e.g., "spectral sensitivity"?

We intend "spectral sensitivity of spectroradiometer". We revised it (L87-89) as below:

However, there is a known noticeable uncertainty in the estimate of PWV because the adjustment parameters depend on the spectral sensitivity of the spectroradiometer as well as the vertical profiles of water vapor and temperature.

We revised it (L102) as below:

The sky-radiometer models POM-01 and POM-02 (Prede, Tokyo, Japan), which are …

- l. 77, "11 wavelengths": maybe a table of the channel wavelengths, together with the main extinction factors, could be useful. See also my first technical comment about the expressions "aerosol channels ... ozone channels" (e.g., even the "ozone channel" is affected by aerosol)

195 - l. 107-108, "aerosol ... cloud ... water vapor ... ozone channels": cf. previous comments. These approximate expressions could be used only after a short explanation;

We agree with reviewer. We revised it (L102-109) and added the table of sky-radiometer specifications as below:

200 The sky-radiometer models POM-01 and POM-02 (Prede, Tokyo, Japan), which are deployed in the international radiation observation network SKYNET, measure solar direct irradiances and diffuse irradiances at the ultraviolet, visible, and near-infrared wavelengths. These measurements are used for the remote sensing of aerosol, cloud, water vapor, and ozone (Table 1; Takamura and Nakajima, 2004; Nakajima *et al.*, 2007). Table 1 shows the relationship between the wavelengths and the main target of

205 the remote sensing. The aerosol channels are 340, 380, 400, 500, 675, 870, and 1020 nm; the water vapor channel is 940 nm; the ozone channel is 315 nm; and the cloud channels are 1225, 1627, and 2200 nm.

**Table 1: Sky-radiometer specifications. Each sky-radiometer is equipped with a filter indicated by**
210 **a circle. "Standard" is the standard specification of sky-radiometer models POM-01 and POM-02.**

| Wavelength [nm] | Strong gas absorption | Main target substance | POM-01 Standard | POM-02 Standard | POM-02 PS1202091 | POM-02 PS2501417 |
|---|---|---|---|---|---|---|
| 315 | $O_3$ | Ozone | ○ | ○ | — | ○ |
| 340 | — | Aerosol | — | ○ | ○ | ○ |
| 380 | — | Aerosol | — | ○ | ○ | ○ |
| 400 | — | Aerosol | ○ | ○ | ○ | ○ |
| 500 | — | Aerosol | ○ | ○ | ○ | ○ |
| 675 | — | Aerosol | ○ | ○ | ○ | ○ |
| 870 | — | Aerosol | ○ | ○ | ○ | ○ |
| 940 | $H_2O$ | Water vapor | ○ | ○ | ○ | ○ |
| 1020 | — | Aerosol | ○ | ○ | ○ | ○ |
| 1225 | $O_2, CO_2, H_2O$ | Cloud | — | — | ○ | — |
| 1627 | $CH_4, CO_2$ | Cloud | — | ○ | ○ | ○ |
| 2200 | $CH_4, H_2O$ | Cloud | — | ○ | ○ | ○ |

- l. 79, "observation ... self-calibration": a bit confusing, please reformulate to avoid mixing of observation and calibration procedures;

215 - l. 80, "works in turbid atmospheric conditions": "only" in turbid conditions or "also" in turbid conditions?

We agree with the reviewer. We revised it (L109-114) as blow:

Through on-site self-calibration of the aerosol channels by the Improved Langley (IL) method (Tanaka *et al.,* 1986; Nakajima *et al.,* 1996; Campanelli *et al.*, 2004, 2007), the SKYNET system is capable of long-term and continuous aerosol observation. The IL method works not only in clean atmospheric conditions, but also in turbid atmospheric conditions.

- l. 81-82, "standard ... modified": please, define what a "standard" method and a "modified" one are;

The standard and modified Langley methods are developed by Uchiyama et al., (2014) and Campanelli et al. (2014), respectively. We revised it (L115-116) as below:

However, no improved calibration method has replaced the standard (Uchiyama *et al.*, 2014) or modified (Campanelli *et al.*, 2014, 2018) Langley methods for the water vapor channel.

- l. 93, "two SKYNET sites": explain why these two sites were selected. Do they have any particular characteristics, or was this choice oriented by the co-located instrumentation?

We chose sites which had been installed not only the sky-radiometer, but also AERONET sun-sky radiometer, GPS/GNSS receiver, and/or MWR. We added the sentence at the end of Section 1 (L129-130) as below:

At these two sites, PWV is observed by the GNSS/GPS receiver, MWR, or AERONET sun-sky radiometer other than the sky-radiometer.

- l. 103, "We explain normalized radiance": article missing?

We agree with the reviewer. We revised it (L142) as below:

We explain the normalized radiance…

- Eq. (1): specify earlier in the text that this holds only for a plane-parallel nonrefractive atmosphere (l. 122, now);

We agree with the reviewer. We revised it (L154-162) as below:

In the plane-parallel non-refractive atmosphere, *F* at the bottom of the atmosphere (BOA) at the solar zenith angle (SZA) $\theta_0$ and the solar azimuth angle $\phi_0$ is derived from

$$F(\lambda) = \frac{F_0}{d^2}\exp\left(-m_0\tau(\lambda)\right), \quad (1)$$

where $F_0$ is the calibration constant; $d$ is the distance between Earth and the sun (AU); $\lambda$ is the wavelength; $\tau$ is the total optical thickness; and $m_0$ is optical air mass, represented as $m_0 = 1/\cos\theta_0$.

- Eq. (2): if L is defined as sky radiance (l. 106), then it should be already divided by the solid view angle (omega);

We used the sky irradiance instead of the sky radiance in the revised manuscript.

- l. 150: the sentence is missing its subject. It is also unclear if this limitation (the real atmosphere not being a single layer) will be addressed in the following text;

We agree with the reviewer. It is deleted and mentioned in Sect. 2.1.2 in the revised manuscript.

- l. 152, "sensitivity of R": ... at a wavelength of 940 nm;

Yes, it is at a wavelength of 940 nm. We revised it (L193) as below:

We examined the sensitivity of $R$ at 940 nm …

- l. 154: I guess that these AOD values refer to 940 nm, too?

Yes exactly. We revised it (L199-201) as below:

Figure 3 shows the dependencies of $R$ in the almucantar plane on PWV for continental average aerosol with aerosol optical thicknesses of 0.02 and 0.20 at 940 nm.

- l. 155-156, "the aerosol optical thickness does not affect this relationship": unclear, since R decreases, but the values do depend on AOD;

We agree with the reviewer. The relationship that $R$ decreases with an increase of PWV was seen in both low and high AOT cases. We revised it (L201-203) as below:

$R$ decreases with increasing PWV regardless of the aerosol optical thickness.

- l. 164: I would not define such a change as "drastic". The variation is only visible in the lower subfigures with a linear y-axis (please, put some letters next to the subplots), and mainly for PWV<=2 cm (as explained later in the text);

It is deleted the word "drastic" in the revised manuscript (L212).

290

No, it isn't.

295 - l. 197, "transmittance of the total extinction": isn't just "trasmittance" enough?

We agree with the reviewer. We revised it (L248).

- Eq. (8): please, explain where the 1.65 factor comes from;

300

This factor, which is defined as $\eta$ in the revised manuscript, was determined in consideration of the smoothness of the VSD. It is written in the Appendix A in the revised manuscript.

**Appendix A: Width of the volume size distribution**

Because $\frac{dV(r)}{d\ln r}$ is expressed by the superposition of 20-modal lognormal size distributions (Eq. [6]), the

305 width of $\frac{dV(r)}{d\ln r}$ is larger than that of each lognormal size distribution. The width of the lognormal size distribution should be small to deal with the complicated and step variations in $\frac{dV(r)}{d\ln r}$. However, $\frac{dV(r)}{d\ln r}$ cannot represent a natural curve if $\eta$ is large and $s$ is small (Fig. A1). Hence, we have to find the maximum value of $\eta$ for making $\frac{dV(r)}{d\ln r}$ a natural curve. When $C_i$ is constant, such value of $\eta$ minimizes the roughness of $\frac{dV(r)}{d\ln r}$, and $\frac{dV(r)}{d\ln r}$ approaches to a flat shape. For a simple formulation, we consider the

310 function $A(x)$ which consists of the multimodal normal distribution function $B_i$ with a constant height. $A(x)$ and $B_i$ are expressed as

$$A(x) = \sum_{i=-\infty}^{\infty} B_i(x) = \sum_{i=-\infty}^{\infty} \exp\left[-\frac{\eta^2}{2}\left(\frac{x-i\xi}{\xi}\right)^2\right], \quad (A1)$$

315 where $i\xi$ and $\frac{\xi}{\eta}$ are the mean and standard deviation, respectively. Its differential is written as

$$\frac{dA}{dx} = \sum_{i=-\infty}^{\infty} \frac{dB_i}{dx} = \sum_{i=-\infty}^{\infty} -\eta^2\left(\frac{x-i\xi}{\xi}\right)\exp\left[-\frac{\eta^2}{2}\left(\frac{x-i\xi}{\xi}\right)^2\right]. \quad (A2)$$

When the shape of $A(x)$ approaches to be flat, the difference between local maximum and minimum values of $A(x)$ is approximately 0. Because $\frac{dB_i}{dx}$ equals 0 at $x = j\xi$ $(j \in \mathbb{Z})$, $A(x)$ has the local maximum and minimum at $x = j\xi$ and $\left(j + \frac{1}{2}\right)\xi$ in $j \le \frac{x}{\xi} < j + 1$. The difference $\Delta$ between the local maximum and minimum values is obtained as

$$\Delta = 1 - \frac{A\left(\frac{2j+1}{2}\xi\right)}{A(j\xi)}. \quad (A3)$$

Figure A2 shows the relation between $\eta$ and $\Delta$. The value of $\Delta$ increases drastically at around $\eta = 1.5$. in addition, the shape of $\frac{dV(r)}{d\ln r}$ is unnatural when $\eta = 2.0$ (Fig. A1). Therefore, the value of $\eta$ should be selected from the values around $\eta = 1.5$. In this study, we fixed $\eta$ at 1.65. This value represents the natural curve of $\frac{dV(r)}{d\ln r}$ and satisfies that the value of $\Delta$ is small enough, $\Delta = 3.0 \times 10^{-3}$.

[Figure]

**Figure A1: Relationship between the volume size distribution and $\eta$. The black line is the volume size distribution, which is computed by the integration of 20-modal lognormal distribution functions (red lines). Blue circles are the peak volume of lognormal size distribution.**

[Figure]

**Figure A2: Relationship between the parameter $\eta$ and the difference $\Delta$.**

- l. 218, "local minimum": does this local minimum change at every retrieval, then?

Yes, exactly. The local minimum of the VSD is calculated at every retrieval.

- l. 338: specify that the integral of the filter response function was normalised to 1 (not its maximum);

The response function is not normalized to 1. We revised it (L386) as below:

$$\bar{T}_{H2O} = \frac{\int_{\Delta\lambda} \Phi(\lambda) T_{H2O}(\lambda) d\lambda}{\int_{\Delta\lambda} \Phi(\lambda) d\lambda} = \frac{\int_{\Delta\lambda} \Phi(\lambda) \exp\left(-m_{H2O}(\theta) \int_0^z \alpha_{H2O}(g_w(z), K(z), \lambda) dz\right) d\lambda}{\int_{\Delta\lambda} \Phi(\lambda) d\lambda} \quad (22),$$

- Eq. (25)-(26): it could be useful to use a subscript (j?) for the single F0's. Also, please use another letter instead of w (in w_H);

We agree with the reviewer. We changed it (L408-417) as below:

The mean value of the calibration constant at the water vapor channel is determined by the robust statistical and iterative method with Huber's M-estimation:

$$\ln \bar{F}_0 = \sum_i \beta_H(t_i) \cdot \ln F_0(t_i), \quad (25)$$

$$\beta_H(t_i) = \begin{cases} 1 & (|\ln \bar{F}_0 - \ln F_0(t_i)| \leq 0.03) \\ \dfrac{0.03}{|\ln \bar{F}_0 - \ln F_0(t_i)|} & (|\ln \bar{F}_0 - \ln F_0(t_i)| > 0.03) \end{cases}, \quad (26)$$

where $\bar{F}_0$ is the mean calibration constant and is calculated at each iterative step, $F_0(t_i)$ is the calibration constant at a specific time $t$, and $\beta_H$ is Huber's weight function.

- Eq. (28)-(29): perhaps it would be better to identify the indices with other letters than R (already used for radiances). Also, it should be mentioned that the range of scattering angles for the calculation of index 2 changes during the day;

We agree with the reviewer. We revised it (L430-442) as below:

Next, the running mean of the time series of $\bar{R}_{\text{near}}(t)$ with a window of three consecutive data points is calculated as $< \bar{R}_{\text{near}}(t) >$. Index 1 is defined as the deviation $\tilde{R}_{\text{near}}(t)$ of $\bar{R}_{\text{near}}(t)$ from $< \bar{R}_{\text{near}}(t) >$,

$$\tilde{R}_{\text{near}}(t) = |\bar{R}_{\text{near}}(t) - < \bar{R}_{\text{near}}(t) >| / < \bar{R}_{\text{near}}(t) >. \qquad (28)$$

Index 2 is the deviation $\tilde{R}_{\text{far}}$ of normalized angular distributions far from the sun and is defined as

$$\tilde{R}_{\text{far}}(t) = \sigma \left( \frac{R(\Theta,t) - <R_{\text{far}}(\Theta,t)>}{<R_{\text{far}}(\Theta,t)>} \right), \Theta > 10^{\text{o}}, \qquad (29)$$

where $< R_{\text{far}}(\Theta, t) >$ is the running mean of $R(\Theta_i, t)$ with a window of three consecutive data points, and $\sigma(\mathbf{X})$ is the standard deviation of data set $\mathbf{X}$. Note that the data for calculating $\tilde{R}_{\text{far}}$ varies depending on SZA, which limits available scattering angles.

- l. 390, l. 397 and Table 2, "misjudged"/"incorrect": it is unclear whether the cloud screening criterium correctly works for the portion of the sky seen by the photometer or not. In the first case, the algorithm does its job, and I think that "misjudged"/"incorrect" are misleading terms, since the conditions of whole sky should not be considered as reference;

"misjudged" and "incorrect" were revised. And "clear-sky" and "cloud affected" were changed to "best condition" and "poor condition" in the revised manuscript (L461-463).

- l. 396-397: are <1 and >2 oktas?

It means cloud cover the range from 0 (no cloud) to 10 (cloud). We used the percentage of cloud cover for whole sky instead of previous "cloud cover the range from 0 (no cloud) to 10 (cloud)". We revised it.

- l. 407, "line regression": do you mean linear regression using AOD and wavelength (not logs)?

"line regression" means "linear interpolation in the log-log plane". We revised it (L473).

- Sect. 3: were only "aerosol channels" (l. 107) + "water vapor channel" used in the synthetic retrieval?

Yes, Sect. 3 conducted intensive sensitivity tests using aerosol channels and the water vapor channel.

- l. 464-465: are the -10% and -3% deviations within the expected uncertainty? If not, can you explain these results? Please, use a proper number of significant digits;

We consider that the difference in the value of the calibration constant between the SKYMAP algorithm and the side-by-side comparison with the reference sky-radiometer was attributable mainly to the calibration period. The calibration constant of the sky-radiometer has seasonal variation due to the temperature dependency of the sensor output (Uchiyama *et al.*, 2018a). Calibration by side-by-side comparison with the reference sky-radiometer was performed only in the winter. However, the calibration constant of the SKYMAP algorithm was the annual mean. The above description is added in the revised manuscript (L551-556).

- l. 472: maybe it would be more scientifically correct to plot these values anyway (with another colour/marker) even though they will not be considered;

We calculated the monthly mean calibration constant in all of observation periods (Fig. 15a, 16a, 17a). However, we did not retrieve PWV using the monthly mean calibration constants for June and July 2014 because their values were obviously small, and because little data were successfully retrieved due to the wet and cloudy conditions in the summer. In addition, it is possible that the measurements were contaminated by clouds. The above description is added in the revised manuscript (L565-568).

- l. 478: a more natural choice would be to linearly interpolate the monthly calibrations. The authors certainly have good reasons for considering the annual mean value, can they elaborate on this?

In this study, we used the annual mean calibration constant due to two factors. First, the monthly mean calibration constants were not significant changed in the dry seasons. Second, the accuracy of the calibration constant decreases at the transition between the wet season and the dry season because little data were successfully retrieved due to the wet and cloudy conditions.

- l. 518, "much more": "larger"?

Yes. We revised it (L639-640) as below:

Larger retrieval errors occurred in the cases when PWV was >2 cm because PWV became less sensitive to the normalized angular distribution.

- Figs. 1 and 7: the authors should explain the difference between the straight boxes and the rounded ones, and why the latter were not used in Fig. 7;

We intend square boxes show the operation of the calculation and input/output parameters and rounded boxes show the operation of the algorithm. We wrote the explanation in the caption of Figures.

- Fig. 2: the label "Principal plane" should be put lower, on the principal place circumference;

We agree with the reviewer. We revised it as below:

[Figure]

**Figure 2: Observation planes (almucantar and principal planes) of the sky-radiometer.**

- Fig. 3: mention somewhere that the plots refer to the principal plane;

Figure 3 in the discussion paper show the normalized radiance in the almucantar plane. We revised it.

[Figure]

$\tau(940) = 0.02$ $\tau(940) = 0.20$

**Figure 3: Normalized angular distributions simulated for continental average aerosol (Table 2) in the almucantar plane with aerosol optical thicknesses of 0.02 and 0.20 at 940 nm. Simulations were conducted for SZA = 70° and PWV (w) = 0, 1, 2, 3, 4, and 5 cm. The top row is the normalized radiance $R(w, \Theta)$, and the bottom row is the ratio of $R(w, \Theta)$ to $R(0, \Theta)$. S-S Approx. is single scattering approximation.**

- Fig. 10: colours in the second row are hardly distinguishable. Explain that they overlap (in the caption);

We agree with the reviewer. We added it in the caption of Figs. 11 and 12 in the revised manuscript as below:

Note that the blue, red, green, and black lines in the middle row overlap.

- Table 3, "Retrieved the PWV": "PWV retrieval"?

We agree with the reviewer. We revised it as below:

**Table 4: References and methodologies of the DSRAD algorithm.**

|  | DSRAD |
|---|---|
| Solar coordinates | Nagasawa (1999) |
| Refraction correction | Nagasawa (1999) |
| Sun-Earth distance | Nagasawa (1999) |
| Optical mass | Gueymard (2001) |
| Rayleigh scattering | Fröhlich and Shaw (1980); Young(1981) |
| Ozone absorption | Sekiguchi and Nakajima (2008) |
| Water vapor absorption | Sekiguchi and Nakajima (2008) |
| Filter response function | Stepwise function |
| Retrieval of PWV | Newton-Raphson method |

**Response to comments of referee 2**

Authors would like to express sincere thanks to the referee 2 for valuable comments. We revised a manuscript carefully based on given comments. The comments of the referee 2 are in blue, our replies are in black, and changes made in the revised manuscript are in red. The English in this document has been checked by at least two professional editors, both native speakers of English. Our replies to the comments are as below.

The study performed by Momoi et al., is a very important evolution in PWV retrieved from radiometers. It exploits findings of established methods and proposes an approach that can be used without prior calibration and could provide reliable PWV from sky-sun measurements. The methodology is described in detail and justified in an appropriate manner. Results are promising and the method could be adopted operationally from Skynet and other networks. I suggest accepting the manuscript for publishing in AMT, after some minor technical corrections and clarifications.

Since it is a novel method, it is crucial to add some preliminary discussion on the uncertainties of the method. It is important for scientists to have an estimation of the expected uncertainties, before applying the method. Is it more accurate than the well-established methods or the main advantage is the field calibration? Also, when transferring the calibration from the one method to the other, to retrieve PWV from the direct sun data, the error propagation is expected to be very high. I suggest discussing this issue in detail.

The main advantage of this manuscript is the field calibration of the water vapor channel (around 940 nm) of the sky-radiometer. In general, the transferring by side-by-side comparison increase the error, but our SKYMAP algorithm is self-calibrating and does not require transfer. The uncertainty of retrieved PWV is discussed by the simulated data with the bias errors in the diffused radiances in Section 3 (L510-521) in revised manuscript as below:

We also conducted sensitivity tests using the simulated data with bias errors to investigate uncertainty in the SKYMAP-derived PWV. The bias errors were ± 5% and ± 10% for $R$. The value of 5% was given by following reasons. The SVA bias errors of the diffuse radiances for the sky-radiometer observations were estimated to be less than 5% (Uchiyama *et al*., 2018b). According to Dubovik *et al*. (2000), the uncertainty of the diffuse radiances for the AERONET measurements is ± 5%. Figures 13 and 14 show the results from the simulated data for the continental average and transported dust aerosols with aerosol optical thicknesses of 0.02, 0.06 and 0.20 at 940 nm. PWV was overestimated when − 5% bias was applied to $R$. This corresponds to the relationship between $R$ and PWV, where $R$ decreases with

increasing PWV (Section 2.1.2). The bias errors strongly affected the retrieval of PWV at high PWV (> 2 cm), because the sensitivity of high PWV is lower than that of low PWV. The retrieval error of PWV increased with increasing bias errors. The retrieval error of PWV due to ± 5% and ± 10% errors for $R$ was within 10% for PWV < 2 cm and up to 200% for PWV > 2 cm.

[Figure]

**Figure 13: Comparison of the "true" and retrieval values of PWV from simulated data for continental average aerosol with bias errors. The top, middle, and bottom rows are the retrieval results at SZA = 30°, 50°, and 70°, respectively. Closed circles are the results with no bias errors. Closed squares and closed triangles are the results with bias errors of plus and minus 5% in $R$, respectively. Open squares and open triangles are the results with bias errors of plus and minus 10% in $R$, respectively.**

[Figure]

**Figure 14: Similar to Fig. 13 but for transported dust aerosol.**

It should be clarified in abstract and introduction section, that the calibration constant is referring to the extraterrestrial / Top of the atmosphere value of F. It might be reasonable for people into sunphotometry, but when referring to various equations and approaches, it should be crystal clear the referred quantity.

We agree with the reviewer. We added the explanation of the calibration constant in abstract and Section 1 in revised manuscript as below:

*(Abstract: L20-23)*

These data are utilized for remote sensing of aerosols, water vapor, ozone, and clouds, but the calibration constant which is the sensor output current of the extraterrestrial solar irradiance at the mean distance between the Earth and the sun, is needed.

*(Section 1: L93-96)*

To estimate PWV using a spectroradiometer, it is necessary to calibrate the water vapor channel. The calibration constant, which is the sensor output current of the extraterrestrial solar irradiance at the mean distance between the Earth and the sun, at the water vapor channel can be determined by the Langley method.

Method for cloud screening described in 2.2.4, is first time described? If it is not, or if it is based on an existing method, some references should be provided. If it totally new, it should be discussed more. Not just one day as an example for the validity. What are the improvements compared to other approaches? If 17% of cloud cases are contaminating the clear sky data, why don't change to different threshold values? 17% seem a big number which will end in high errors to the data set. At least for the validation of PWV method, stricter criterion is preferable, though it might result in smaller database, because the goal is to estimate the results of algorithm in clear sky conditions.

The SCAD method is newly developed in this study for the application to the observational site where only the sky-radiometer in installed. We add the further discussion about determining thresholds in the revised manuscript (L455-464) as below:

The results were validated using visual observation of the amount of clouds in the Aerological Observatory of the JMA. Figure 10a shows the histograms of index 1 for cases in which the sun was and was not covered by clouds. Index 1 had a low value when there were no clouds shading the sun but had a wide range of values when clouds were shading the sun. Fig. 10b shows the histograms of index 2 when cloud cover was and was not $< 20\%$. The peak shifted to the right when cloud cover was $\geq 20\%$, but the effect was not significant. Table 3 shows the validation results of this method. We defined "best condition" as cloud cover $< 20\%$ and "poor condition" as cloud cover $\geq 20\%$. In less than 17% of cases a "poor condition" was judged as a "best condition". The sky-radiometer observes only a part of the whole sky, but our algorithm showed good results.

[Figure]

[Figure]

**Figure 10: Histograms of indexes 1 and 2 of sky-radiometer observations at Tsukuba. (a) Index 1 when the sun is covered by clouds (blue boxes) and not covered by clouds (red boxes). (b) Index 2 when cloud cover is less than to 20% (red boxes) and greater than or equal to 20% (blue boxes).**

**Table 3: Validation of the SCAD method by visual observation from 2013 to 2014 in Tsukuba.**

| Visual observation | Sky-radiometer measuring plane | |
|---|---|---|
| Cloud cover | Available condition | Poor quality condition |
| Clear, less than 20% | **463 (83.4%)*** | 68 (8.7%) |
| Cloud affected, more than 20% | 92 (16.6%) | **714(91.3%)*** |

\*Obviously correct determination.

In step 2 of SKYMAP. Does it retrieves PWV or just the corresponding transmittance (as stated in the abstract?) If it is just the transmittance is should be clarified in the description and change the title of 2.2.2. If it is PWV it is important to plot separately the PWV retrieved with SKYMAP in the comparisons sections. Are these retrievals useful or are just a step in the calculations to obtain the Fo?

Step 2 in SKYMAP algorithm retrieves the PWV, not the transmittance of the PWV. It is just a step in the calculations to obtain the F0. Instead of plotting SKYMAP-derived PWV, DSRAD-derived PWV with the calibration constant determined by the SKYMAP algorithm is compared in section 4.

Technical comments

- Abreviations should be defined also in the abstract.

We agree with the reviewer. We revised it.

- L56-60. The two sentences should be separated. It seems that PWV is defined only at 940nm. 940 is a bandwidth that is selected because of the highest absorption in the shortwave spectral range.

We agree with the reviewer. We revised it (L79-84) as below:

Precipitable water vapor (PWV), which is the total atmospheric water vapor contained in a vertical column, has been estimated from the measurement of direct solar irradiance at the water vapor absorption bands. One of the strong water vapor absorption bands is around 940 nm and can be measured by sun photometer (Fowle, 1912, 1915; Bruegge *et al.,* 1992; Schmid *et al.,* 1996, 2001; Halthore *et al.,* 1997), SKYNET sky-radiometer (Campanelli *et al.,* 2014, 2018; Uchiyama *et al.,* 2014, 2018a), and AERONET sun-sky photometer (Holben *et al.,* 1998).

L66 Bruegge approach was also dependable on the altitude of the station, which made it difficult when transporting an instrument.

Yes, it is written in L.63-65 in the discussion paper and added the sentence in the revised manuscript (L87-90) as below:

However, there is a known noticeable uncertainty in the estimate of PWV because the adjustment parameters depend on the spectral sensitivity of the spectroradiometer as well as the vertical profiles of water vapor and temperature. Therefore, the adjustment parameters should be determined for each observation site.

Figure 10-11. It is not clear what is referred as true values. Please explain in the manuscript.

We agree with the reviewer. We revised it (L499-501) as below:

The retrievals of the volume size distribution, aerosol optical thickness, and PWV corresponded with their input values ("true" values in Fig. 11) when the input of PWV was <2 cm.

L124 It seems that something is missing. Which quantity is integrated from BOA to TOA?

We agree with the reviewer. We revised it (L162-165) as below:

In clear-sky conditions, the total optical thickness is the integrated value of aerosol scattering + absorption, Rayleigh scattering, and gas absorption coefficients in the column.

L152-153 How R was simulated? An RTM was used? Please describe in detail.

Yes, $R$ is simulated by RTM (RSTAR). We revised it (L193-195) as below:

We examined the sensitivity of $R$ at 940 nm in the two observation planes to PWV, aerosol optical properties, and aerosol vertical profiles by simulating $R$ using the radiative transfer model RSTAR (Nakajima and Tanaka, 1986, 1988).

Figure 3. Please use same range in y-axis because it is confusing when it changes all the time.

We agree with the reviewer. We revised it as below:

[Figure]

**Figure 3: Normalized angular distributions simulated for continental average aerosol (Table 2) in the almucantar plane with aerosol optical thicknesses of 0.02 and 0.20 at 940 nm. Simulations were conducted for SZA = 70° and PWV (w) = 0, 1, 2, 3, 4, and 5 cm. The top row is the normalized radiance $R(w, \Theta)$, and the bottom row is the ratio of $R(w, \Theta)$ to $R(0, \Theta)$. S-S Approx. is single scattering approximation.**

L167 Some explanation should be provided regarding the selection of dust for the simulation.

We agree with the reviewer. We revised it (L216-218) as below:

The transported dust aerosol is composed of coarse particles, which have larger impacts on the angular distribution of $R$ at the near-infrared wavelength than fine particles.

L240 There are not 18 boundary layers. As it stated by the term boundary, it is located on the boundaries. Stratosphere is not a boundary layer. I suggest to change to just " 18 layers"/

The atmosphere consists 17 layers which boundary altitude is 18. Therefore, we revised it (L291-292) as below:

175 The model atmosphere is divided by 18 altitudes of 0, 1, 2, 3, 4, 5, 6, 7, 8, 9, 10, 15, 20, 30, 40, 50, 70, and 120 km.

L440. 0.5cm is too big and not visible in figure 10. Since it is for values <2cm, this is more tha 50% error. Is it a typo or it is estimated from somewhere not shown on the figure?

180

We estimate the uncertainty from the sensitivity tests in Section 3. "0.5 cm" is a typo. We revised it (L522-524) as below:

When the input of PWV was < 2 cm, the SKYMAP algorithm retrieved PWV very well, within an error of 10% regardless of the aerosol optical thickness or aerosol type. This was also observed when the bias

185 errors were added for $R$.

L476-85. Write in a clear manner that annual values refer to Fo and not PWV.

Yes, it refers to F0. We wrote "the annual mean calibration constant" instead of "the annual mean value"

190 in the revised manuscript.

**Response to comments of referee 3**

Authors would like to express sincere thanks to the referee 3 for valuable comments. We revised a manuscript carefully based on given comments. The comments of the referee 3 are in blue, our replies are in black, and changes made in the revised manuscript are in red. The English in this document has been checked by at least two professional editors, both native speakers of English. Our replies to the comments are as below.

This manuscript written by Momoi et al. presents a new in-situ method to calibrate the so-called water channel in a sky-radiometer Prede POM02, and to derive the precipitable water vapor content in the atmospheric column, by using measurements from the sky-radiometer in the almucantar and principal planes. The method means a further step in the development of methods for the retrieval of aerosols and gases, with very good applicability for the SKYNET international network.

The method description is also complemented with sensitivity analysis, ending with a couple of experimental cases along 1-2 years in two different sites.

The paper is clear, with a good detail about the theoretical considerations. The English grammar is good, with very few typo errors if any. I consider the manuscript must be accepted in AMT, after some comments are addressed.

General comments:

- It is very important to include an estimation of the uncertainty of the method, performed by expanding the forward sensitivity analysis or by comparison with other methods, even if preliminary. This information is critical and I miss it in the abstract and more in depth in the text body, if it is currently available.

- Errors in aerosol properties interpolated at 940 nm could have an impact on the comparison of PWV between AERONET and this method? Have the authors checked the comparison of aerosol properties from Cimel and this method, during the period studied? If yes, it is not need to include a detailed description, but comment (related to Figure 14).

We discussed the uncertainty of PWV by conducted the sensitivity tests using the simulated data with the bias errors in diffuse radiances in Section 3 and compared with the AERONET sun-sky radiometer in Section 4.2. We also estimate error propagation from aerosol optical thickness to PWV in Appendix B in the revised manuscript as below:

*(Section 3: L510-521)*

We also conducted sensitivity tests using the simulated data with bias errors to investigate uncertainty in the SKYMAP-derived PWV. The bias errors were ± 5% and ± 10% for $R$. The value of 5% was given by following reasons. The SVA bias errors of the diffuse radiances for the sky-radiometer observations were estimated to be less than 5% (Uchiyama *et al*., 2018b). According to Dubovik *et al*. (2000), the uncertainty of the diffuse radiances for the AERONET measurements is ± 5%. Figures 13 and 14 show the results from the simulated data for the continental average and transported dust aerosols with aerosol optical thicknesses of 0.02, 0.06 and 0.20 at 940 nm. PWV was overestimated when − 5% bias was applied to $R$. This corresponds to the relationship between $R$ and PWV, where $R$ decreases with increasing PWV (Section 2.1.2). The bias errors strongly affected the retrieval of PWV at high PWV (> 2 cm), because the sensitivity of high PWV is lower than that of low PWV. The retrieval error of PWV increased with increasing bias errors. The retrieval error of PWV due to ± 5% and ± 10% errors for $R$ was within 10% for PWV < 2 cm and up to 200% for PWV > 2 cm.

[Figure]

 **Figure 13: Comparison of the "true" and retrieval values of PWV from simulated data for continental average aerosol with bias errors. The top, middle, and bottom rows are the retrieval results at SZA = 30°, 50°, and 70°, respectively. Closed circles are the results with no bias errors. Closed squares and closed triangles are the results with bias errors of plus and minus 5% in *R*, respectively. Open squares and open triangles are the results with bias errors of plus and minus**

 **10% in *R*, respectively.**

[Figure]

**Figure 14: Similar to Fig. 13 but for transported dust aerosol.**

 *(Section 4.2: L607-625)*

PWV$_{DSRAD+SKYMAP}$ using the annual mean calibration constant was 12% and 9.1% smaller than PWV$_{MWR}$ and PWV$_{Cimel}$, respectively (Table 5). These results suggest an underestimation of PWV$_{DSRAD+SKYMAP}$, as the uncertainty of PWV$_{Cimel}$ compared to the GNSS/GPS receiver is expected to

be less than 10% (Giles *et al.*, 2018). The underestimation of PWV$_{DSRAD+SKYMAP}$ was due to two factors. The first is the retrieval of PWV by the annual mean calibration constant for the water vapor channel. The calibration constant not only is subject to aging but also undergoes seasonal variation due to temperature dependency (Uchiyama *et al.*, 2018a). Thus, it is possible to underestimate the calibration constant in the wet season. Second, uncertainty regarding the aerosol optical thickness affected PWV retrieval. Figure 18 depicts the differences in PWV and aerosol optical thicknesses at 675, 870, and 1020 nm between the DSRAD algorithm and the AERONET retrieval. In the periods from January to May and from October to November, the differences in PWV and aerosol optical thicknesses were less than 0.1 cm and 0.015, respectively. However, the difference in PWV was greater than 0.1 cm from July to September. This corresponds to the difference in aerosol optical thicknesses at 675, 870, and 1020 nm from July to September, which indicates that the transmittance of water vapor was overestimated by the overestimation of aerosol optical thickness. This led to the underestimation of PWV$_{DSRAD+SKYMAP}$ using the annual mean calibration constant when PWV was > 3 cm. In our error estimation, the error of + 0.03 for the aerosol optical thickness at 940 nm resulted in the error of − 0.214 cm for PWV (Appendix B).

[Figure]

**Figure 18: The top row shows the time series of PWV in 2017 at Chiba (green and black circles are PWV$_{DSRAD+SKYMAP}$ and PWV$_{Cimel}$, respectively). The middle row is the difference between**

**Table 5: Comparison of PWV between DSRAD and other instruments.**

|  |  | Slope $C_1$ | Intercept $C_2$[cm] | $\gamma$ | Bias [cm] | RMSE [cm] |
|---|---|---|---|---|---|---|
| PS1202091 at Tsukuba, Japan |  |  |  |  |  |  |
| Monthly mean $F_0$ | vs GNSS/GPS receiver (2013) | 0.956 | 0.079 | 0.938 | -0.049 | 0.138 |
|  | vs GNSS/GPS receiver (2014) | 0.937 | 0.161 | 0.970 | -0.110 | 0.170 |
| Annual mean $F_0$ | vs GNSS/GPS receiver (2013) | 0.919 | 0.173 | 0.987 | -0.061 | 0.226 |
|  | vs GNSS/GPS receiver (2014) | 0.934 | 0.178 | 0.987 | -0.089 | 0.223 |
| PS2501417 at Chiba, Japan |  |  |  |  |  |  |
| Monthly mean $F_0$ | vs MWR (2017) | 0.964 | 0.053 | 0.961 | -0.027 | 0.091 |
|  | vs AERONET (2017) | 0.987 | 0.107 | 0.976 | 0.098 | 0.122 |
| Annual mean $F_0$ | vs MWR (2017) | 0.880 | 0.132 | 0.985 | 0.042 | 0.231 |
|  | vs AERONET (2017) | 0.909 | 0.184 | 0.991 | 0.055 | 0.186 |

$C_1, C_2$: $PWV_{DSRAD} = C_1 \times PWV_{Other} + C_2$
Bias: $PWV_{DSRAD} - PWV_{Other}$

90

*(Appendix B)*

**Appendix B: Error propagation from aerosol optical thickness to PWV**

We evaluated the influence of the uncertainty of aerosol optical thickness on PWV using the empirical equation of Bruegge *et al*. (1992). PWV is described using the adjustment parameters as follows

95

$$w = \frac{1}{m_0}\left(-\frac{\ln \bar{T}_{H2O}}{a}\right)^{\frac{1}{b}} \text{[cm]}. \qquad \text{(B1)}$$

The uncertainty of PWV $\epsilon_{PWV}$ is given from the partial differentiation of Eq. (B1) with respect to $\ln \bar{T}_{H2O}$ as follows

100

$$\epsilon_{PWV} = \frac{\partial w}{\partial \ln \bar{T}_{H2O}}\epsilon_{\ln \bar{T}_{H2O}} = \frac{w}{b \ln \bar{T}_{H2O}}\epsilon_{\ln \bar{T}_{H2O}}. \qquad \text{(B2)}$$

where $\epsilon_{\ln \bar{T}_{H2O}}$ is the uncertainty of $\bar{T}_{H2O}$. Using Eq. (B1) with the adjusting parameters of the sky-radiometer, with $a = 0.620$ and $b = 0.625$ as the coefficient values for the trapezoidal spectral response function (Uchiyama *et al*., 2018a), we write the uncertainty of PWV as

105

$$\epsilon_{PWV} = -\frac{w}{ab}(m_0 w)^{-b}\epsilon_{\ln \bar{T}_{H2O}} = -\frac{w}{0.388}(m_0 w)^{-0.625}\epsilon_{\ln \bar{T}_{H2O}}. \qquad \text{(B3)}$$

If the uncertainty of the calibration constant at the water vapor channel is ignored, the uncertainty of

$\bar{T}_{H2O}$ is given from Eq. (21) as follows

$$\epsilon_{\ln \bar{\tau}_{H2O}} = m_0 \epsilon_{AOT}. \qquad (B4)$$

where $\epsilon_{AOT}$ is the uncertainty of the aerosol optical thickness at 940 nm. The uncertainty of PWV is written by Eqs. (B3) and (B4) as

$$\epsilon_{PWV} = -\frac{1}{0.388}(m_0 w)^{0.375}\epsilon_{AOT} = -0.214 \text{ [cm]}. \qquad (B5)$$

where $m_0 = 3.0$, $w = 5.0$ cm, and $\epsilon_{AOT} = 0.03$.

- The POM02 has been used in this study; however, version POM01 also has a channel in 940nm. Then I understand the method can also be applied to this version. This should be made clear.

We agree with the reviewer. Our methods (SKYMAP and DSRAD) can be used for sky-radiometer model POM-01. Thus, Section 1 and 2 of the revised manuscript was generally written.

- What is the model used for the sensitivity analysis performed in section 2.1.2? RSTAR? State it clearly and give some detail.

Yes, $R$ is simulated by RTM (RSTAR). We revised it (L193-195) as below:

We examined the sensitivity of $R$ at 940 nm in the two observation planes to PWV, aerosol optical properties, and aerosol vertical profiles by simulating $R$ using the radiative transfer model RSTAR (Nakajima and Tanaka, 1986, 1988).

- A comparison to other techniques for PWV retrieval developed in SKYNET framework is envisaged (i.e. Campanelli et al), and discussion of pro/cons advised. If this is planned for a future work, please state.

We agree with the reviewer. We revised it (L656-658) as below:

In future work, we plan to compare our method with others in the SKYNET framework (Uchiyama *et al*. 2014; Campanelli *et al*., 2014).

- Is the cloud screening also a new method developed for this study? Two different thresholds are used for the two near and far indices. How these thresholds have been selected? Did you performed an statistically study, or it is simply a preliminary proposal?

Yes, the cloud screening method, called SCAD method, is developed for this study. We determined the thresholds by comparing the images of the whole-sky camera and the time series of the surface solar radiation observed by the pyranometer and validated by visual observation. We added the validation results of the SCAD method in the revised manuscript (L455-464) as below:

The results were validated using visual observation of the amount of clouds in the Aerological Observatory of the JMA. Figure 10a shows the histograms of index 1 for cases in which the sun was and was not covered by clouds. Index 1 had a low value when there were no clouds shading the sun but had a wide range of values when clouds were shading the sun. Fig. 10b shows the histograms of index 2 when cloud cover was and was not < 20%. The peak shifted to the right when cloud cover was ≥ 20%, but the effect was not significant. Table 3 shows the validation results of this method. We defined "best condition" as cloud cover < 20% and "poor condition" as cloud cover ≥ 20%. In less than 17% of cases a "poor condition" was judged as a "best condition". The sky-radiometer observes only a part of the whole sky, but our algorithm showed good results.

[Figure]

**Figure 10: Histograms of indexes 1 and 2 of sky-radiometer observations at Tsukuba. (a) Index 1 when the sun is covered by clouds (blue boxes) and not covered by clouds (red boxes). (b) Index 2 when cloud cover is less than to 20% (red boxes) and greater than or equal to 20% (blue boxes).**

**Table 3: Validation of the SCAD method by visual observation from 2013 to 2014 in Tsukuba.**

| Visual observation | Sky-radiometer measuring plane | |
| --- | --- | --- |
| Cloud cover | Available condition | Poor quality condition |
| Clear, less than 20% | **463 (83.4%)\*** | 68 (8.7%) |
| Cloud affected, more than 20% | 92 (16.6%) | **714(91.3%)\*** |

\*Obviously correct determination.

Technical comments

170

- Abstract, line 22: "remains challenging" ?

Yes, it is. We revised it (L28) as below:

…the sky-radiometer remains challenging…

175

- Abstract, line 28-30: in the second step, the transmittance of PWV is retrieved; however, both in the text and the plots look like it is the PWV content, not the transmittance, retrieved. Please write consistently.

180  The step 2 of the SKYMAP retrieves PWV to estimate the transmittance of the PWV, but not directly retrieves the PWV. We revised it (L39).

- page 3, line 73: the calibration method described for AERONET is solely for field instruments, please state.

185

We agree with the reviewer. We revised it (L97-100) as below:

In the AERONET led by NASA, the field instrument of the AERONET sun-sky radiometer is calibrated every year by lamp calibration and side-by-side comparison with a reference spectroradiometer (Holben *et al.*, 1998).

190

- page 6, line 150: I'm not sure to understand the need for this sentence here, or it looks like incomplete discussion here.

We agree with the reviewer. It is deleted and mentioned in Sect. 2.1.2 in the revised manuscript.

195

- Page 11, line 326: In equation 21, the band average transmittance is used, converted from PWV in step 2. Do you mean the conversion is performed in the step 3, from the PWV obtained in step 1?

The band average transmittance in Eq. 21 is converted from the PWV retrieved in step 2 not step 1.
Since the diffuse sky irradiances are calculated for each sub-band by the radiative transfer model, in step 2 you can get the PWV instead of the band average transmittance of the PWV and need to convert it to the band average transmittance to get the F0.

- Page 12, line 355. Is the Huber's M-estimation method iterative? (the weight to calculate ln<F0> depends on ln<F0>, so any condition applied?)

Yes, Huber's M-estimation method is iterative. This method has the potential to apply noisy satisfying normal distribution, not white noise.

- Page 14, line 415: what is the time stamp used for any triplet? The central element time stamp or the first element time stamp?

The time stamp used the central element time stamp. We revised it (L485-486) as below:
Second, the triplet of the aerosol optical thickness in Smirnov *et al*. (2000) is built from the pre/post 1 min data instead of 30 s.

- Page 14, line 440: this error estimation about 0.5cm looks somewhat high to me, in comparison to other techniques. It would be useful to provide literature estimations of the uncertainty for the methods used as referents (AERONET, MWR, GPS...) in the comparison discussion.

We estimate the uncertainty from the sensitivity tests in Section 3. "0.5 cm" is a typo. We compared our retrievals with more establish methods, such as GNSS/GPS-derived PWV, and AERONET-derived PWV as below:
*(Section 4.1: L582-587)*
Table 5 summarizes the results of comparisons of DSRAD-derived PWV and GNSS/GPS-derived PWV. The magnitude of the bias error and root mean square error were small, less than 0.11 cm and less than 0.226 cm, during 2013 to 2014. Table 6 shows the errors of the retrieved PWV with the annual mean calibration constants for the rank of PWV. The bias error was larger for high PWV than it was for low PWV. The magnitude of the bias errors of PWV was less than 0.163 cm for PWV < 3 cm and less than 0.339 cm for PWV > 3 cm.

*(Section 4.2: L602-610)*
$PWV_{DSRAD+SKYMAP}$ using the annual mean calibration constant agreed with $PWV_{MWR}$ (Fig. 17c). The error of $PWV_{DSRAD+SKYMAP}$ was $-0.041 < bias < 0.024$ cm and RMSE $< 0.212$ cm for low PWV (<3

235 cm) and bias $< -0.356$ cm and RMSE $> 0.465$ cm for high PWV (Table 6). Figure 17d shows that $PWV_{DSRAD+SKYMAP}$ using the annual mean calibration constant also agreed with $PWV_{Cimel}$ for low PWV ($< 3$ cm) but was smaller than $PWV_{Cimel}$ for high PWV ($> 3$ cm). $PWV_{MWR}$ was larger than $PWV_{Cimel}$ (Fig. 17e). $PWV_{DSRAD+SKYMAP}$ using the annual mean calibration constant was 12% and 9.1% smaller than $PWV_{MWR}$ and $PWV_{Cimel}$, respectively (Table 5).

240

**Table 5: Comparison of PWV between DSRAD and other instruments.**

| | Slope $C_1$ | Intercept $C_2$[cm] | $\gamma$ | Bias [cm] | RMSE [cm] |
|---|---|---|---|---|---|
| PS1202091 at Tsukuba, Japan | | | | | |
| Monthly mean $F_0$  vs GNSS/GPS receiver (2013) | 0.956 | 0.079 | 0.938 | -0.049 | 0.138 |
| vs GNSS/GPS receiver (2014) | 0.937 | 0.161 | 0.970 | -0.110 | 0.170 |
| Annual mean $F_0$  vs GNSS/GPS receiver (2013) | 0.919 | 0.173 | 0.987 | -0.061 | 0.226 |
| vs GNSS/GPS receiver (2014) | 0.934 | 0.178 | 0.987 | -0.089 | 0.223 |
| PS2501417 at Chiba, Japan | | | | | |
| Monthly mean $F_0$  vs MWR (2017) | 0.964 | 0.053 | 0.961 | -0.027 | 0.091 |
| vs AERONET (2017) | 0.987 | 0.107 | 0.976 | 0.098 | 0.122 |
| Annual mean $F_0$  vs MWR (2017) | 0.880 | 0.132 | 0.985 | 0.042 | 0.231 |
| vs AERONET (2017) | 0.909 | 0.184 | 0.991 | 0.055 | 0.186 |

$C_1, C_2$: $PWV_{DSRAD} = C_1 \times PWV_{Other} + C_2$
Bias: $PWV_{DSRAD} - PWV_{Other}$

245 **Table 6: Difference in PWV between DSRAD with the annual mean calibration constants and other instruments.**

| | $PWV_{Other}$ | | | | |
|---|---|---|---|---|---|
| | $0-1$ cm | $1-2$ cm | $2-3$ cm | $3-4$ cm | $> 4$ cm |
| | Bias [cm] (RMSE [cm]) | Bias [cm] (RMSE [cm]) | Bias [cm] (RMSE [cm]) | Bias [cm] (RMSE [cm]) | Bias [cm] (RMSE [cm]) |
| PS1202091 at Tsukuba, Japan | | | | | |
| vs GNSS/GPS receiver (2013) | 0.083 (0. 124) | 0.160 (0.211) | 0.084 (0.236) | -0.098 (0.326) | -0.339 (0.537) |
| vs GNSS/GPS receiver (2014) | 0.110 (0.142) | 0.163 (0.221) | 0.107 (0.251) | -0.055 (0.353) | -0.239 (0.492) |
| PS2501417 at Chiba, Japan | | | | | |
| vs MWR (2017) | 0.017 (0.066) | 0.024 (0.153) | -0.041 (0.212) | -0.356 (0.465) | -0.594 (0.722) |
| vs AERONET (2017) | 0.088 (0.105) | 0.118 (0.192) | 0.017 (0.223) | -0.214 (0.386) | -0.264 (0.306) |

Bias: $PWV_{DSRAD} - PWV_{Other}$

- Page 17, summary (conclusions): is there any error estimation for PWV>2cm? Some discussion on the
250 pros and cons of this method and other methods developed for Prede POM instruments would be useful.

We added the results of the uncertainty for PWV >2 cm in the revised manuscript (L650-655) as below: The magnitude of the bias error and the root mean square error were $< 0.163$ cm and $< 0.251$ cm, respectively, for low PWV ($< 3$ cm). However, our retrieved PWV was underestimated in the wet

255 conditions, and the magnitude of the bias error and the root mean square error were less than 0.594 cm and less than 0.722 cm for high PWV. This was due to seasonal variation in the calibration constant and the overestimation of aerosol optical thickness at 940 nm interpolated from those at 870 and 1020 nm.

- Is figure 1 used in the text?

260

It is using the top of Section 2 and Section 3 in the discussion paper.

- Caption figure 3: please explain what S-S refers to (I think it is single scattering) .

265 Yes, "S-S approx." means the single scattering approximation. We added the sentence in the caption as below:

S-S Approx. is single scattering approximation.

- Figures: I think saying "top row" and "bottom row" is more appropiate than "top and bottom line".

270

We agree with the reviewer. We revised it.

[revised manuscript text omitted]

$$\qquad f(x) = \frac{1}{2}\big(y^{\text{meas}} - y(x)\big)^T (W^2)^{-1}\big(y^{\text{meas}} - y(x)\big) + \frac{1}{2}\big(y_a(x)\big)^T (W_a^2)^{-1}\big(y_a(x)\big), \quad (13)$$

where vector $y^{\text{meas}}$ describes the measurements (normalized radiances $R^{\text{meas}}$ and transmittances of total extinction $T^{\text{meas}}$ ) at the aerosol channels, vector $x$ describes the aforementioned
305 aerosol parameters —$n(\lambda)$, $k(\lambda)$, $C_i$, and $\delta$— to be estimated, vector $y(x)$ comprises the values corresponding to $y^{\text{meas}}$ calculated from $x$ by the forward model ($R^{\text{ret}}$ and $T^{\text{ret}}$), and matrix $W^2$ is the covariance matrix of $y$ and is assumed to be diagonal. The diagonal elements of $W$ are  standard errors in the measurements. We set their values at 0.02 for $T^{\text{meas}}$, and 10% for $R^{\text{meas}}$.

To reduce the effects of observational error on retrieval and to conduct stable analyses, Dubovik and King (2000) considered restricting the spectral variability of the volume size
310 distribution and  limiting the length of  the refractive index derivative with respect to the wavelength . They considered this *a priori* smoothness constraint as being of the same nature as a measurement and incorporated the smoothness constraint into their retrieval scheme. We also consider the smoothness constraints in this study. The second term of Eq. (13) consists of *a priori* information on the wavelength dependencies of the refractive index, aerosol
315 optical thickness, and smoothness of the volume spectrum, which is described as

$$y_a(x) = \big(y_a^{\text{Re}}, y_a^{\text{Im}}, y_a^{\text{Sca}}, y_a^{\text{Abs}}, y_a^{\text{Vol}}\big)^T, \qquad (14)$$

where vectors $y_a^{\text{Re}}$, $y_a^{\text{Im}}$, $y_a^{\text{Sca}}$, $y_a^{\text{Abs}}$, and $y_a^{\text{Vol}}$ are *a priori* information on the wavelength dependencies
320 of the refractive index (real and imaginary parts), aerosol optical thickness (scattering and absorption parts), and smoothness of the volume spectrum, respectively. The matrix $W_a^2$ in Eq. (13) is the covariance matrix for determining the strengths of the constraints.

[revised manuscript text omitted]

$$\bar{T}_{\text{H2O}} = \frac{\int_{\Delta\lambda}\Phi(\lambda)T_{\text{H2O}}(\lambda)d\lambda}{\int_{\Delta\lambda}\Phi(\lambda)d\lambda} =$$

$$\frac{\int_{\Delta\lambda}\Phi(\lambda)\exp\left(-m_{\text{H2O}}(\theta)\int_0^z \alpha_{\text{H2O}}(g_w(z),K(z),\lambda)dz\right)d\lambda}{\int_{\Delta\lambda}\Phi(\lambda)d\lambda}$$

$$(22)$$

$$w = \int_0^z g_w(z)dz, \quad (23)$$

where $\Phi(\lambda)$ is the filter response function, $\Delta\lambda$ is the bandwidth of the filter response function, $T_{\text{H2O}}$ is the transmittance of water vapor at wavelength $\lambda$, $m_{\text{H2O}}(\theta)$ is the optical air mass, $g_w$ is the mass mixing ratio, $K$ is temperature, $\alpha_{\text{H2O}}$ is the absorption coefficient at altitude $z$, and $w$ is PWV. Eq. (22) is discretized by

$$\bar{T}_{\text{H2O}} =$$

$$\frac{\sum_i^{N_s} \Phi_i \int_{\Delta\lambda_i} \exp\left(-m_{\text{H2O}}(\theta)\int_0^Z \alpha_{\text{H2O}}(g_w(z),K(z),\lambda)dz\right)d\lambda}{\sum_i^{N_s}\Phi_i\Delta\lambda_i} \quad \frac{1}{\Delta\lambda}\sum_i^{N_s}\Phi_i\int_{\Delta\lambda_i}\exp\left(-m_{\text{H2O}}(\theta)\int_0^z\sigma_{\text{H2O}}(g_w(z),K(z),\lambda)dz\right)d\lambda$$

$$\tag{24}$$

where $\Phi_i$ is the stepwise filter response function, $\Delta\lambda_i$ is the sub-bandwidth of the filter response function, and $N_s$ is the number of sub-bands. We calculate the absorption coefficients at each wavelength by the correlated $k$-distribution (Sekiguchi and Nakajima, 2008) using the vertical profiles of temperature, pressure, and specific humidity in the NCEP/NCAR Reanalysis 1 data.

We can calculate a value for $F_0$ from a data set of the normalized angular distribution. Therefore, for example,  a_ time series of $F_0$ in a day is obtained from the daily measurements of the sky-radiometer. The mean value of the calibration constant at the water vapor channel is determined by the robust statistical_ and iterative_ method with Huber's M-estimation:

$$\ln \bar{F}_0 = \sum_i \beta_H(t_i) \cdot \ln F_0(t_i) \; \cancel{\textstyle\sum w_H \cdot \ln F_0} \tag{25}$$

$$\cancel{w}\beta_H(t_i) = \begin{cases} 1 & (|\ln \bar{F}_0 - \ln F_0(t_i)\cancel{F_0}| \le 0.03) \\ \dfrac{0.03}{|\ln \bar{F}_0 - \ln F_0(t_i)\cancel{F_0}|} & (|\ln \bar{F}_0 - \ln F_0(t_i)\cancel{F_0}| > 0.03) \end{cases} \tag{26}$$

where $\bar{F}_0$ is the mean calibration constant and is calculated at each iterative step, $F_0(t_i)$ is the calibration constant at a specific time_ t_, and $\cancel{w}\beta_H$ is Huber's weight function.

**2.2.4 Cloud screening using the smoothness criteria of the angular distributions (SCAD method)**

The SKYMAP algorithm can only be applied to measurements under clear-sky conditions. We estimated clear-sky conditions from two indexes calculated from sky-radiometer measurements. Index 1 is a value for  the normalized radiances_ near the sun. If clouds pass over the sun, index 1 has large temporal variation. Index 2 is a value for the normalized angular distribution. If clouds are detected on the scanning plane of the sky-radiometer, the normalized angular distribution has large variation. Index 1 is defined as follows. First, the mean normalized radiance near the sun_ $\bar{R}_{\text{near}}$ is calculated by

$$\bar{R}_{\text{near}}(t) = \frac{1}{N}\sum_{i=1}^{N} R(\Theta_i, t), \; \Theta \le 10^{\circ} \tag{27}$$

where $N$ is the number of measurements, and $R$ is the normalized radiance at a time $t$, scattering angle $\Theta$, and wavelength 500 nm. Next, the running mean of the time series of $\bar{R}_{\text{near}}$$(t)$ with a window of three consecutive data points is calculated as $< \bar{R}_{\text{near,mean}}(t) >$. Index 1 is defined as the deviation $\tilde{R}_{\text{near}}(t)$ of $\bar{R}_{\text{near}}(t)$ from $< \bar{R}_{\text{near}}(t) >$,

$$\tilde{R}_{\text{near,}\sout{\text{dev}}}(t) = \left| \bar{R}_{\text{near}}(t) - < \bar{R}_{\text{near}}(t) > \sout{R_{\text{near,mean}}(t)} \right| / < \bar{R}_{\text{near}}(t) > \sout{R_{\text{near,mean}}(t)}. \quad (28)\sout{.}$$

[revised manuscript text omitted]